# Erase to Improve: Erasable Reinforcement Learning for Search-Augmented LLMs

**Kang An**[1,2*†], **Ziliang Wang**[1*‡], **Xhui Zheng**[1,3*†],
**Faqiang Qian**[1], **Weikun Zhang**[1], **Yuhang Wang**[1‡], **Yichao Wu**[1§]
[1]SenseTime [2]Shenzhen University [3]Nanjing University
{wangziliang1, qianfaqiang, zhangweikun,
wangyuhang, wuyichao}@sensetime.com
zhengxuhui@smail.nju.edu.cn, ankang@gml.ac.cn

## Abstract

While search-augmented large language models (LLMs) exhibit impressive capabilities, their reliability in complex multi-hop reasoning remains limited. This limitation arises from three fundamental challenges: decomposition errors, where tasks are incorrectly broken down; retrieval missing, where key evidence fails to be retrieved; and reasoning errors, where flawed logic propagates through the reasoning chain. A single failure in any of these stages can derail the final answer. We propose Erasable Reinforcement Learning (ERL), a novel framework that transforms fragile reasoning into a robust process. ERL explicitly identifies faulty steps, erases them, and regenerates reasoning in place. This targeted correction mechanism turns brittle reasoning into a more resilient process. Models trained with ERL, termed ESearch, achieve substantial improvements on HotpotQA, MuSiQue, 2Wiki, and Bamboogle, with the 3B model achieving +8.48% EM and +11.56% F1, and the 7B model achieving +5.38% EM and +7.22% F1 over previous state-of-the-art(SOTA) results. These findings suggest that erasable reinforcement learning provides a powerful paradigm shift for robust multi-step reasoning in LLMs.

## 1 Introduction

Large language models (LLMs) have produced remarkable advances across a broad spectrum of natural language processing tasks, including question answering, reasoning, and code generation (Singh et al., 2025; Yang et al., 2025). Notwithstanding these advances, inherent limitations in their static pretraining corpora leave them susceptible to hallucination and factual error, especially in knowledge-intensive domains and in tasks that require reasoning over multiple steps (Huang et al., 2025a; 2024). Even the most advanced models tailored for rigorous reasoning, such as OpenAI o1 (Jaech et al., 2024), DeepSeek R1 (Guo et al., 2025) and Kimi k2 (Team et al., 2025), still face substantial difficulty in reliably solving complex multi-hop problems that demand precise decomposition, dependable retrieval, and long-term logical consistency (Xi et al., 2025). To address these difficulties, retrieval-augmented generation (RAG) has emerged as a dominant paradigm, enriching large language models with external knowledge sources (Lewis et al., 2020). Over time, RAG has evolved into sophisticated research agents that integrate search and reasoning within an autonomous loop. Systems such as OpenAI Deep Research (OpenAI, 2025), Gemini Deep Research (DeepMind, 2025), and Perplexity Deep Research (AI, 2025) mark significant milestones in this trajectory. Reinforcement learning (RL) (Li, 2017) has emerged as a central force driving recent breakthroughs in the field of search-augmented agents. An increasing number of studies explore leveraging RL to guide decomposition, retrieval, and reasoning (Jin et al., 2025b; Song et al., 2025; Zhao et al., 2025b). These approaches employ reward signals to improve sub-query generation, evidence retrieval, and reasoning chains, achieving substantial gains on challenging benchmarks such

---

[*]Equal contribution
[†]Work done during internship at SenseTime
[‡]Project leader
[§]Corresponding author

as HotpotQA (Yang et al., 2018), MuSiQue (Trivedi et al., 2022), and 2WikiMultiHopQA (Ho et al., 2020). Despite these impressive advances, current systems remain highly brittle. They can reliably answer simple factual queries, yet even minor errors in decomposition, retrieval, or reasoning can compromise an entire multi-hop trajectory (Huang et al., 2025b; Li et al., 2025). In contrast, humans rarely fail so catastrophically. When we recognize a flaw in a reasoning step, we pause, correct the mistake, and continue from the corrected point. This stark contrast highlights a critical limitation of current search-augmented RL systems: they lack the robust self-correction mechanisms that underlie human reasoning.

Through extensive empirical analysis of current RL-based search agents, we uncover three critical failure modes that fundamentally limit their capabilities:

- **Decomposition Errors**: Incorrect subqueries derail the retrieval process entirely, preventing downstream steps from ever accessing the crucial evidence needed to answer the question.
- **Retrieval missing**: Retrieved documents that are irrelevant even with appropriate subqueries due to noise, ambiguity, or incomplete coverage, causing subsequent reasoning to fail.
- **Reasoning Errors**: LLMs may make mistakes when integrating retrieved information, and these errors accumulate across steps, systematically undermining the reliability of the final answer.

Deeper structural flaws significantly exacerbate these issues. Existing reinforcement learning agents typically treat the entire search and reasoning trajectory as a single Markov Decision Process (MDP) (Kaelbling et al., 1996), optimizing only via sparse terminal rewards (Jin et al., 2025b). This monolithic design is fundamentally brittle: a single misstep can compromise the entire trajectory. As reasoning chains extend (Zhang et al., 2025b; Jin et al., 2025a), this fragility intensifies, causing performance to degrade precipitously beyond ten steps (Gao et al., 2025).

Overcoming these limitations requires a radical paradigm shift: agents must emulate human-like self-correction by detecting errors, discarding flawed steps, and resuming reasoning from the most recently corrected state. Analogous to a skilled writer using an eraser to remove a single mistaken word without discarding the entire manuscript. We introduce Erasable Reinforcement Learning (ERL), a novel framework that embodies this principle. ERL enables search-augmented LLM agents to identify errors in decomposition, retrieval, or reasoning precisely, selectively erase the faulty segments, and regenerate from the last correct state. This fine-grained corrective mechanism transforms brittle trajectories into resilient ones, allowing agents to recover gracefully from mistakes rather than collapsing entirely. We conduct extensive evaluations on HotpotQA (Yang et al., 2018), MuSiQue (Trivedi et al., 2022), 2WikiMultiHopQA (Ho et al., 2020), and Bamboogle (Press et al., 2023). The results show that models trained with ERL not only surpass strong baselines and state-of-the-art (SOTA) methods but also consistently improve performance; the 3B model achieves gains of +8.48% EM and +11.56% F1, while the 7B model achieves +5.38% EM and +7.22% F1.

The main contributions are summarized as follows:

- Systematic identification of three critical failure modes in search-augmented LLMs for complex multi-hop reasoning.
- Introduction of ERL, a framework for fine-grained error detection, erasure, and regeneration that substantially improves reasoning robustness.
- Establishment of new SOTA results on multiple multi-hop QA benchmarks, validating the effectiveness and generality of ERL.

## 2 PRELIMINARY

Previous work often models complex multi-hop question answering, combining search and reasoning, as a MDP characterized by (Chen et al., 2024):

$$(\mathcal{S}, \mathcal{A}, \mathcal{P}, \mathcal{R}, \gamma). \tag{1}$$

In this framework, the state $s_t$ represents the reasoning trajectory up to time $t$, providing context for subsequent actions (Broekens et al., 2010). We define the state $s_t$ as:

$$s_t = (Q, H_t) = (Q, (a_0, e_0), (a_1, e_1), \dots, (a_{t-1}, e_{t-1})), \tag{2}$$

where $Q$ is the original question and $H_t$ is the sequence of interactions up to time $t$. Each action $a_i \in \mathcal{A}$ represents reasoning or retrieval, and each environment $e_i$ corresponds to the evidence information resulting from $a_i$ by calling the tools. The agent's action space $\mathcal{A} = \{o, r, q\}$ includes atomic operations governing reasoning, Searching and Answering, while the tool corresponding to the environment will provide searched documents:

- Search Query ($q_t$): Produces a query $q_t$ to retrieve relevant evidence $e_t$.

- Observation ($o_t$): Reasoning an observation $o_t$ of the evidence $e_t$ and previous status $s_{t-1}$

- Sub Answer ($r_t$): Yield an intermediate phased conclusion $r_t$ after observation $o_t$.

- Finish(answer): Produces the final answer $A_{\text{final}}$ when sufficient evidence is gathered.

The state transition function $\mathcal{P}(s_{t+1} \mid s_t, a_t)$ (Huang, 2022) is driven by two mechanisms: the stochastic generation by the LLM and the search results from external search engines. For the convenience of representation and to fit the complex answering mechanism of multi-hop QA, each action $a_t$ is defined as a fixed and ordered sequence of unit actions of $\langle (o_t, r_t), q_t \rangle$ (no observation or response to previous evidence in the first round.) for intermediate solving process, or $\langle A_{final} \rangle$ for the answer to finish. All the action and action sequence is sampled from the LLM's conditional distribution:

$$a_t = Act(s_t) \sim P_\theta(\cdot \mid s_t). \tag{3}$$

For the Search Query ($q_t$) issued by the agent, the environment $e_t$ is the information evidence retrieved from the search tools:

$$e_t = \text{Search}(q_t). \tag{4}$$

The next state $s_{t+1}$ is formed by appending the combination of the new action sequence and environment to the trajectory $H_t$:

$$s_{t+1} = (Q, H_t \oplus (a_t, e_t)), \quad a_t \in \begin{cases} \langle (o_t, r_t), q_t \rangle, & \text{intermediate step,} \\ \langle A_{\text{final}} \rangle, & \text{final answer.} \end{cases} \tag{5}$$

In multi-hop question answering, the reward function $\mathcal{R}$ (Jin et al., 2025b; Song et al., 2025)is typically sparse, rewarding the agent only upon completing the reasoning trajectory and producing the final answer $A_{\text{final}}$. The reward is computed by comparing $A_{\text{final}}$ with the reference $A_{\text{gold}}$ using metrics like exact match (EM) or F1 score:

$$\mathcal{R}(s_t, a_t) = \begin{cases} \text{Eval}(A_{\text{final}}, A_{\text{gold}}) & \text{if } a_t \text{ is } \langle A_{final} \rangle, \\ 0 & \text{otherwise.} \end{cases} \tag{6}$$

The agent optimizes the expected terminal reward via policy gradient methods:

$$J(\phi) = \mathbb{E}_{\tau \sim \pi_\phi} \big[ \mathcal{R}(\tau) \big], \tag{7}$$

where $\tau = (s_0, a_0, e_0, \ldots, s_T)$ is the reasoning trajectory. However, treating the entire reasoning trajectory as a monolithic sequence for optimization introduces a structural vulnerability, known as catastrophic fragility. In a reasoning trajectory $\tau = (s_0, a_0, e_o, s_1, \ldots, a_{T-1}, e_{T-1}, s_T)$, any failure at a single step can disrupt the entire process, leading to an erroneous final outcome. For instance, an error at step $t < T$ can result from:

- **Decomposition Errors**: The GenerateQuery action generates a deviated sub-query $q_t$.

- **Retrieval Omissions**: The Search action fails to retrieve the relevant evidence $e_t$.

- **Reasoning Errors**: The Synthesize action produces an incorrect intermediate conclusion $r_t$.

The error at $s_{t+1}$ contaminates all subsequent states $(s_{t+2}, \ldots, s_T)$, as each following action $(a_{t+1}, \ldots, a_{T-1})$ depends on this contaminated history. This resembles a domino effect, where the failure of a single link leads to an entire system collapse. This structural fragility is the core reason for the unreliability of current search-augmented LLMs when tackling complex, multi-hop problems.

## 3 METHOD

### 3.1 REINFORCEMENT LEARNING WITH A SEARCH ENGINE

We extend reinforcement learning to incorporate search engines into policy optimization. The objective is

$$\max_{\pi_\theta} \mathbb{E}_{x \sim \mathcal{D},\, y \sim \pi_\theta(\cdot|x;\mathcal{R})} \big[ r_\phi(x,y) \big] - \beta D_{\mathrm{KL}} \big[ \pi_\theta(y \mid x; \mathcal{R}) \,\|\, \pi_{\mathrm{ref}}(y \mid x; \mathcal{R}) \big],$$

where $\pi_\theta$ is the policy, $\pi_{\mathrm{ref}}$ the reference model, and $r_\phi$ the reward (Jin et al., 2025b). Inputs $x$ contain both natural language and retrieved results, enabling $\pi_\theta$ to learn retrieval–reasoning integration beyond prompt-based methods. For training we adopt Proximal Policy Optimization (PPO) (Schulman et al., 2017), yielding

$$J_{\mathrm{PPO}}(\theta) = \mathbb{E}_{x \sim \mathcal{D},\, y \sim \pi_\theta(\cdot|x;\mathcal{R})} \Bigg[ \frac{1}{L} \sum_{t=1}^{L} I(y_t) \min \Big( \frac{\pi_\theta(y_t \mid y_{<t}, x; \mathcal{R})}{\pi_{\mathrm{old}}(y_t \mid y_{<t}, x; \mathcal{R})} A_t, \\ \mathrm{clip}\Big( \tfrac{\pi_\theta(y_t | y_{<t}, x; \mathcal{R})}{\pi_{\mathrm{old}}(y_t | y_{<t}, x; \mathcal{R})}, \, 1 - \epsilon, \, 1 + \epsilon \Big) A_t \Big) \Bigg] \tag{8}$$

with $\pi_{\mathrm{old}}$ the previous policy, $I(y_t)$ masking retrieved tokens, and $A_t$ the advantage from GAE (Schulman et al., 2015).

### 3.2 ROUND-BASED REASONING

We model reasoning as a sequence of $T$ structured rounds. Each round $t$ produces an interaction pair $\langle a_t, e_t \rangle$, where $a_t$ denotes the action and $e_t$ the retrieved evidence. If $a_t = \langle (o_t, r_t), q_t \rangle$, the agent executes the sequence `<observation> `$o_t$` </observation>` $\rightarrow$ `<sub_answer> `$r_t$` </sub_answer>` $\rightarrow$ `<search> `$q_t$` </search>`, and the query $q_t$ is submitted to a search tool to obtain evidence $e_t$. If $a_t = \langle A_{\mathrm{final}} \rangle$, the process terminates with the final answer. The policy action units can be defined on the original question $Q$ and previous action unit in the dialogue history $h_t = \{ \langle (o_i, r_i), q_i, e_i \rangle \}_{i=1}^{t-1}$:

$$o_t \sim \pi_\theta(\cdot \mid Q, h_t) \rightarrow r_t = \pi_\theta(\cdot \mid Q, \langle h_t, o_t \rangle) \rightarrow q_t = \pi_\theta(\cdot \mid Q, \langle h_t, (o_t, r_t) \rangle) \rightarrow e_t = \mathrm{Search}(q_t). \tag{9}$$

This structured format allows the agent to alternate between querying and reasoning, tightly coupling retrieval with generation. The episode terminates when the policy outputs $\langle A_{final} \rangle$ like `<answer> `$A_{\mathrm{final}}$` </answer>`.

### 3.3 REWARD DESIGN

Dense stepwise rewards are critical to prevent sparse supervision. ERL introduces two intermediate rewards, $R_t^{\mathrm{search}}$ for sub-queries and $R_t^{\mathrm{sub\_answer}}$ for intermediate reasoning, in addition to the final reward $R^{\mathrm{answer}}$.

**Search reward.** Let gold evidence $\mathcal{D}^\star = \{d_i^\star\}_{i=1}^n$ and retrieved set $D^{(t)} = \{d_j^{(t)}\}_{j=1}^k$. Define TF–IDF cosine similarity $s(d_i^\star, d_j^{(t)})$. Maintain coverage vector $m_i^t$:

$$c_i^t = \max_j s(d_i^\star, d_j^{(t)}), \quad \Delta_i^t = \max\{c_i^t - m_i^{t-1}, 0\}, \quad G^t = \frac{1}{n} \sum_{i=1}^n \Delta_i^t, \quad m_i^t = \max\{m_i^{t-1}, c_i^t\}. \tag{10}$$

Redundancy penalty is defined as Eq. (11), and the final search reward is Eq. (12) as below. This design encourages novel evidence retrieval while suppressing repeated queries.

$$P^t = \frac{1}{k} \sum_{j=1}^k \mathbf{1}(d_j^{(t)} \in H^{t-1}), \quad H^t = H^{t-1} \cup D^{(t)}. \tag{11}$$

$$R_t^{\mathrm{search}} = G^t - P^t. \tag{12}$$

**Sub-answer reward.** Let gold sub-answers $\mathcal{A}^\star = \{a_i^\star\}_{i=1}^m$. With F1 overlap:

$$f_{i,t} = \mathrm{F1}(r_t, a_i^\star), \ u_i^t = \max\{u_i^{t-1}, f_{i,t}\}, \ \delta_i^t = \max\{u_i^t - u_i^{t-1}, 0\}, \ S^t = \max_i \delta_i^t. \quad (13)$$

Reward:

$$R_t^{\mathrm{sub\_answer}} = \frac{S^t}{\max\{m, 1\}}. \quad (14)$$

This ensures that only genuine improvements to intermediate reasoning are rewarded.

**Final reward.**

$$R^{\mathrm{answer}} = \tfrac{1}{2}\,\mathrm{EM}(A_{\mathrm{final}}, A_{\mathrm{gold}}) + \tfrac{1}{2}\,\mathrm{F1}(A_{\mathrm{final}}, A_{\mathrm{gold}}). \quad (15)$$

The token-level attribution aligns $R_t^{\mathrm{search}}$ with `</search>`, $R_t^{\mathrm{sub\_answer}}$ with `</observation>` and `</sub_answer>`, and $R^{\mathrm{answer}}$ with `</answer>`.

## 3.4 ERASABLE REINFORCEMENT LEARNING

Rewards alone cannot prevent compounding errors. ERL introduces erasure operators that surgically remove faulty parts of the trajectory, enabling as shown in Figure 1. We define a trajectory as $\tau = (s_0, s_1, \ldots, s_T)$. For any $t \le T$, we denote the truncated prefix of the trajectory up to step $t$ by

$$\tau_{0:t} = (s_0, s_1, \ldots, s_t). \quad (16)$$

We further introduce an erasure operator $\mathcal{E}$, which modifies the action sequence according to different conditions in each round. Formally,

$$\mathcal{E}[a_t, e_t] \in \begin{cases} \langle \texttt{None} \rangle, & \text{if the sub-answer } r_t \text{ is incorrect,} \\ \langle (o_t, r_t) \rangle, & \text{if the initial or subsequent search results are incorrect,} \\ \langle (o_t, r_t), q_t \rangle, e_t, & \text{if the action sequence is valid.} \end{cases} \quad (17)$$

$$s_{t+1} = \tau_{0:t} \oplus \mathcal{E}[Act(s_t), Search(q_t)] = \tau_{0:t} \oplus \mathcal{E}[a_t, e_t].$$

Different erasure conditions can be explained with two thresholds are introduced: $\alpha$ for local errors and $\beta$ for plan-level errors. And here goes the details:

**Sub-Answer Erasure.** If $R_t^{\mathrm{sub\_answer}} \le \alpha$, erase `<observation>`, `<sub_answer>` and all subsequent actions of round $t$, meaning any actions in the current round are discarded:

$$s_{t+1} \leftarrow \tau_{0:t} \oplus \langle \texttt{None} \rangle = s_t. \quad (18)$$

**Subsequent Search Erasure.** If $R_t^{\mathrm{search}} \le \alpha$ and $t > 1$, erase the query behavior `<search>` issued at round $t$ and keep the correct `<observation>` with `<sub_answer>`:

$$s_t \leftarrow \tau_{0:t} \oplus \langle o_t, r_t \rangle. \quad (19)$$

**Initial Search/Plan Erasure.** If $R_1^{\mathrm{search}} \le \beta$ in the first action round of $t = 0$, no observation or sub-answer action unit and erase the query behavior `<search>` (reset the trajectory):

$$\tau \leftarrow \tau_{0:0} \oplus \langle \texttt{None} \rangle = s_0. \quad (20)$$

## 4 EXPERIMENT

### 4.1 EXPERIMENTAL SETUP

**Datasets and Evallution Metrics** We evaluate on four multi-hop QA benchmarks: HotpotQA (Yang et al., 2018), 2WikiMultihopQA (Ho et al., 2020), MuSiQue (Trivedi et al., 2022), and Bamboogle (Press et al., 2023), which span diverse domains and reasoning complexities. We report performance using canonical word-level F1 and Exact Match (EM) metrics, while refraining from the use of third-party LLM evaluators owing to concerns regarding reproducibility and stability.

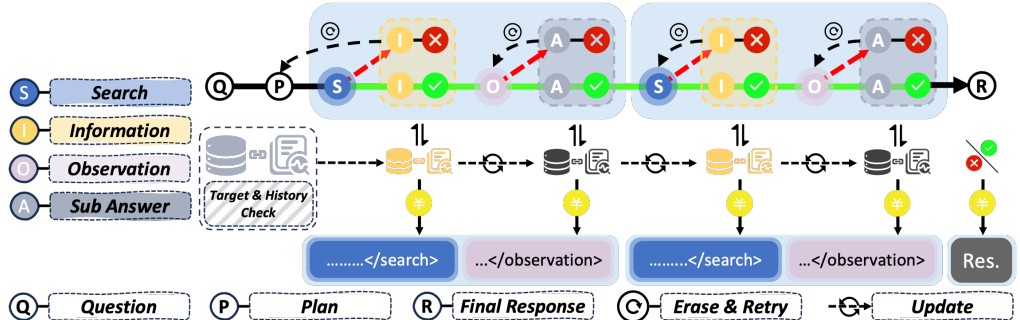

Figure 1: Overview of ESearch. Different colors and symbols are used to represent the interactive behaviors S (Search), I (Information), O (Observation), and A (Sub Answer). In the answering process, there are three types of erasure and retry behaviors: (1) incorrect initial search results trigger initialization plan erasure; (2) incorrect subsequent search results trigger search erasure; (3) incorrect sub-answer triggers observation erasure. In addition, history checking and target set are built for searches and sub-answers respectively to evaluate value gains, which serve as the basis for erasure triggers and reward calculation.

| Method | HotpotQA[†] | | 2Wiki[†] | | MuSiQue[†] | | Bamboogle[†] | | HotpotQA[*] | | 2Wiki[*] | | MuSiQue[*] | | Bamboogle[*] | |
|---|---|---|---|---|---|---|---|---|---|---|---|---|---|---|---|---|
| | EM↑ | F1↑ | EM↑ | F1↑ | EM↑ | F1↑ | EM↑ | F1↑ | EM↑ | F1↑ | EM↑ | F1↑ | EM↑ | F1↑ | EM↑ | F1↑ |
| **Qwen2.5-3b-Base/Instruct** | | | | | | | | | | | | | | | | |
| Search-R1-base | 0.272 | 0.361 | 0.248 | 0.296 | 0.081 | 0.146 | 0.176 | 0.270 | 0.348 | 0.431 | 0.381 | 0.445 | 0.120 | 0.184 | 0.280 | 0.400 |
| Search-R1-instruct | 0.304 | 0.401 | 0.293 | 0.352 | 0.120 | 0.188 | 0.240 | 0.344 | 0.350 | 0.442 | 0.371 | 0.452 | 0.128 | 0.195 | 0.392 | 0.513 |
| ZeroSearch-base | 0.281 | 0.377 | 0.253 | 0.311 | 0.096 | 0.164 | 0.165 | 0.256 | 0.324 | 0.414 | 0.392 | 0.473 | 0.152 | 0.237 | 0.361 | 0.522 |
| ZeroSearch-instruct | 0.267 | 0353 | 0.239 | 0.288 | 0.088 | 0.145 | 0.193 | 0.299 | 0.357 | 0.453 | 0.355 | 0.441 | 0.114 | 0.176 | 0.421 | 0.543 |
| R-Search-instruct-GRPO | 0.329 | 0.427 | 0.307 | 0.351 | 0.131 | 0.208 | 0.228 | 0.327 | 0.374 | 0.460 | 0.457 | 0.519 | 0.142 | 0.227 | 0.504 | 0.644 |
| R-Search-instruct-PPO | 0.289 | 0.381 | 0.277 | 0.328 | 0.124 | 0.187 | 0.260 | 0.355 | 0.398 | 0.495 | 0.496 | 0.558 | 0.152 | 0.234 | 0.496 | 0.656 |
| SSRL-instruct | 0.314 | 0.408 | 0.290 | 0.348 | 0.093 | 0.156 | 0.216 | 0.287 | 0.346 | 0.424 | 0.365 | 0.461 | 0.114 | 0.195 | 0.344 | 0.453 |
| StepSearch-base | 0.329 | 0.434 | 0.339 | 0.395 | 0.181 | 0.273 | 0.328 | 0.419 | 0.345 | 0.464 | 0.434 | 0.542 | 0.196 | 0.291 | 0.502 | 0.631 |
| StepSearch-instruct | 0.345 | 0.452 | 0.320 | 0.385 | 0.174 | 0.261 | 0.344 | 0.452 | 0.394 | 0.470 | 0.402 | 0.496 | 0.150 | 0.240 | 0.520 | 0.626 |
| ESearch-base | 0.415 | 0.548 | **0.428** | 0.499 | **0.236** | **0.345** | 0.414 | 0.529 | 0.435 | 0.586 | **0.581** | **0.684** | **0.247** | **0.367** | 0.633 | 0.797 |
| ESearch-instruct | **0.447** | **0.587** | 0.415 | **0.500** | 0.232 | 0.339 | **0.446** | **0.587** | **0.513** | **0.612** | 0.521 | 0.644 | 0.211 | 0.311 | **0.674** | **0.813** |
| **Qwen2.5-7b-Base/Instruct** | | | | | | | | | | | | | | | | |
| Search-R1-base | 0.432 | 0.547 | 0.350 | 0.411 | 0.206 | 0.290 | 0.430 | 0.545 | 0.508 | 0.610 | 0.533 | 0.607 | 0.219 | 0.310 | 0.577 | 0.692 |
| Search-R1-instruct | 0.394 | 0.502 | 0.312 | 0.376 | 0.181 | 0.262 | 0.384 | 0.501 | 0.464 | 0.570 | 0.475 | 0.561 | 0.182 | 0.268 | 0.536 | 0.660 |
| Research-base | 0.294 | 0.388 | 0.264 | 0.313 | 0.143 | 0.230 | 0.373 | 0.449 | 0.386 | 0.486 | 0.457 | 0.534 | 0.176 | 0.275 | 0.488 | 0.582 |
| Research-instruct | 0.362 | 0.471 | 0.354 | 0.416 | 0.184 | 0.271 | 0.424 | 0.544 | 0.494 | 0.608 | 0.539 | 0.628 | 0.220 | 0.321 | 0.544 | 0.666 |
| ZeroSearch-base | 0.375 | 0.481 | 0.297 | 0.356 | 0.201 | 0.286 | 0.417 | 0.532 | 0.431 | 0.529 | 0.525 | 0.593 | 0.211 | 0.297 | 0.505 | 0.634 |
| ZeroSearch-instruct | 0.388 | 0.497 | 0.360 | 0.422 | 0.219 | 0.320 | 0.433 | 0.540 | 0.394 | 0.483 | 0.431 | 0.534 | 0.136 | 0.225 | 0.368 | 0.492 |
| R-Search-instruct-GRPO | 0.391 | 0.500 | 0.346 | 0.401 | 0.179 | 0.260 | 0.400 | 0.517 | 0.376 | 0.468 | 0.470 | 0.535 | 0.134 | 0.225 | 0.464 | 0.601 |
| R-Search-instruct-PPO | 0.338 | 0.439 | 0.274 | 0.339 | 0.133 | 0.209 | 0.384 | 0.491 | 0.358 | 0.453 | 0.462 | 0.527 | 0.158 | 0.240 | 0.464 | 0.593 |
| SSRL-instruct | 0.380 | 0.489 | 0.332 | 0.399 | 0.153 | 0.238 | 0.344 | 0.466 | 0.388 | 0.465 | 0.358 | 0.442 | 0.106 | 0.184 | 0.336 | 0.438 |
| StepSearch-base | 0.380 | 0.493 | 0.385 | 0.450 | 0.216 | 0.324 | 0.467 | 0.573 | 0.446 | 0.552 | 0.561 | 0.638 | 0.232 | 0.325 | 0.544 | 0.698 |
| StepSearch-instruct | 0.386 | 0.502 | 0.366 | 0.431 | 0.226 | 0.312 | 0.400 | 0.534 | 0.462 | 0.560 | 0.485 | 0.570 | 0.222 | 0.327 | 0.600 | 0.718 |
| ESearch-base | 0.434 | 0.564 | **0.436** | **0.513** | **0.244** | **0.371** | **0.534** | **0.656** | 0.510 | 0.632 | **0.635** | **0.730** | **0.265** | 0.372 | 0.622 | 0.799 |
| ESearch-instruct | **0.442** | **0.576** | 0.419 | 0.494 | 0.241 | 0.358 | 0.458 | 0.612 | 0.507 | **0.642** | 0.550 | 0.654 | 0.254 | **0.375** | **0.687** | **0.823** |

Table 1: The main results. "†" indicates offline retrieval, and "∗" indicates online retrieval.

**Baseline Method** We employ various baselines to evaluate our proposed ESearch, including Search-R1 (Jin et al., 2025b), Research (Chen et al., 2025), ZeroSearch (Sun et al., 2025), R-Search (Zhao et al., 2025a), SSRL (Fan et al., 2025), StepSearch (Zheng et al., 2025a).

**Implementation Details** We conduct experiments using two model scales: Qwen2.5-3B-base/instruct and Qwen2.5-7B-base/instruct (Yang et al., 2024). During training, we adopt E5 (Wang et al., 2022) as the retriever, with the document corpus built from the Wikipedia 2018 dump (Wiki-18) (Karpukhin et al., 2020). For offline evaluation, we maintain the same Wikipedia dump as the retrieval corpus to ensure consistency with the training setup. For online evaluation, we employ the Google Search API as the retrieval source.

| Method | HotpotQA[†] | | 2Wiki[†] | | MuSiQue[†] | | Bamboogle[†] | | HotpotQA[*] | | 2Wiki[*] | | MuSiQue[*] | | Bamboogle[*] | |
|---|---|---|---|---|---|---|---|---|---|---|---|---|---|---|---|---|
| | EM↑ | F1↑ | EM↑ | F1↑ | EM↑ | F1↑ | EM↑ | F1↑ | EM↑ | F1↑ | EM↑ | F1↑ | EM↑ | F1↑ | EM↑ | F1↑ |
| **Qwen2.5-7b-Base** | | | | | | | | | | | | | | | | |
| ERL | **0.434** | **0.564** | **0.436** | **0.513** | **0.244** | **0.371** | **0.534** | **0.656** | **0.510** | **0.632** | **0.635** | **0.730** | **0.265** | **0.372** | **0.622** | **0.799** |
| PPO | 0.371 | 0.475 | 0.279 | 0.326 | 0.196 | 0.278 | 0.428 | 0.545 | 0.382 | 0.422 | 0.475 | 0.547 | 0.198 | 0.277 | 0.475 | 0.603 |
| GRPO | 0.350 | 0.462 | 0.267 | 0.344 | 0.203 | 0.292 | 0.398 | 0.514 | 0.401 | 0.497 | 0.499 | 0.568 | 0.208 | 0.292 | 0.489 | 0.623 |
| **Qwen2.5-3b-Base** | | | | | | | | | | | | | | | | |
| ERL | **0.415** | **0.548** | **0.428** | **0.499** | **0.236** | **0.345** | **0.414** | **0.529** | **0.435** | **0.586** | **0.581** | **0.684** | **0.247** | **0.367** | **0.633** | **0.797** |
| PPO | 0.264 | 0.372 | 0.265 | 0.322 | 0.106 | 0.192 | 0.206 | 0.313 | 0.242 | 0.326 | 0.322 | 0.380 | 0.137 | 0.204 | 0.352 | 0.443 |
| GRPO | 0.258 | 0.367 | 0.254 | 0.321 | 0.113 | 0.188 | 0.223 | 0.312 | 0.237 | 0.319 | 0.326 | 0.382 | 0.134 | 0.199 | 0.345 | 0.437 |

Table 2: Accuracy performance of models trained by different RL algorithms."†" indicates offline retrieval using the wiki-18 knowledge base, and "∗" indicates online retrieval using Google Search.

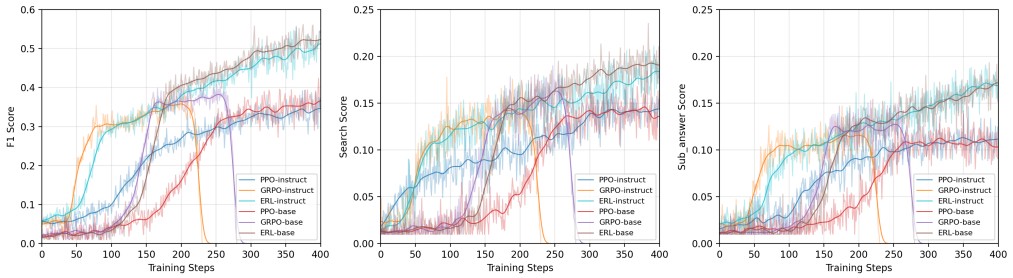

Figure 2: Training dynamics of different RL strategies.

## 4.2 MAIN RESULTS

**Offline evaluation**   Table 1 shows that ESearch sets a new SOTA on four multi-hop QA datasets, consistently surpassing strong baselines with Qwen2.5; with three billion parameters, it gains +6.06% EM and +9.94% F1 on average, rising to +4.78% EM and +6.56% F1 with seven billion parameters.

**Online evaluation**   We evaluate models using a continuously updated search engine instead of a static knowledge base. Online retrieval improves nearly all methods by providing fresher, more comprehensive information. ESearch consistently surpasses baselines, achieving +10.90% EM and +13.18% F1 for Qwen2.5-3B, and +5.98% EM and +7.88% F1 for Qwen2.5-7B. Across scales, ESearch delivers the largest relative gains, demonstrating superior adaptability and robustness in dynamic environments.

**Comparison with classical reinforcement learning algorithms**   To further assess the effectiveness of ERL, we conduct a direct comparison with two classical reinforcement learning algorithms: PPO and GRPO, both of which rely solely on task-level success as the reward signal. Experimental results, summarized in Table 2 and Figure 2, demonstrate that ERL consistently and substantially outperforms PPO and GRPO on both the 3B and 7B models. Compared to PPO, GRPO demonstrates faster learning speed and reward acquisition during training but tends to suffer from instability and potential collapse in the later stages of Search Agent tasks. In contrast, ERL achieves higher learning efficiency than PPO while maintaining training stability.

## 5 ANALYSIS

To quantify the relative contributions of each component in the ERL framework, we conducted a systematic ablation study. Table 3 presents the performance of different component combinations. The full ERL framework achieves the best performance across all datasets, confirming our design principle that the three erasure mechanisms are complementary. Plan-triggered erasure proves essential on highly structured datasets, with disabling it leading to a -2.05% F1 on 2Wiki, yet it remains ineffective in addressing missing retrieval. Search-triggered erasure shows clear advantages for retrieval-intensive tasks, where disabling it results in a -2.40% F1 on Bamboogle, but it fails to remedy global reasoning errors. Sub-answer-triggered erasure benefits reasoning-intensive tasks,

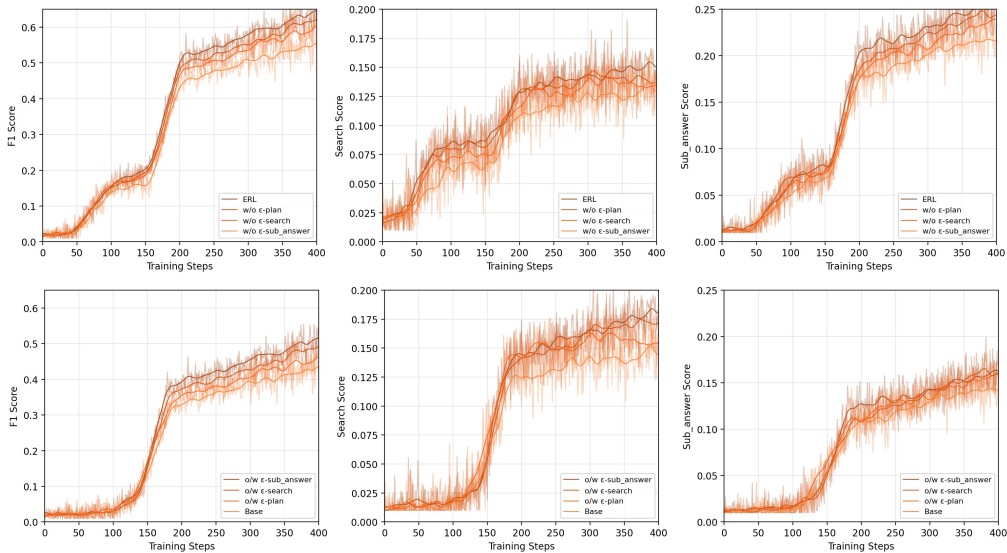

Figure 3: Training dynamics in ablation experiments.

| Method | HotpotQA† | | 2Wiki† | | MuSiQue† | | Bamboogle† | | HotpotQA* | | 2Wiki* | | MuSiQue* | | Bamboogle* | |
|---|---|---|---|---|---|---|---|---|---|---|---|---|---|---|---|---|
| | EM↑ | F1↑ | EM↑ | F1↑ | EM↑ | F1↑ | EM↑ | F1↑ | EM↑ | F1↑ | EM↑ | F1↑ | EM↑ | F1↑ | EM↑ | F1↑ |
| **Qwen2.5-7b-Base** | | | | | | | | | | | | | | | | |
| ERL | **0.434** | **0.564** | **0.436** | **0.513** | **0.244** | **0.371** | **0.534** | **0.656** | **0.510** | **0.632** | **0.635** | **0.730** | **0.265** | **0.372** | **0.622** | **0.799** |
| w/o $\varepsilon - plan$ | 0.420 | 0.545 | 0.421 | 0.496 | 0.236 | 0.359 | 0.517 | 0.634 | 0.494 | 0.611 | 0.620 | 0.706 | 0.257 | 0.361 | 0.602 | 0.773 |
| w/o $\varepsilon - search$ | 0.410 | 0.533 | 0.412 | 0.485 | 0.231 | 0.351 | 0.505 | 0.620 | 0.483 | 0.598 | 0.607 | 0.691 | 0.256 | 0.354 | 0.588 | 0.755 |
| w/o $\varepsilon - sub\_answer$ | 0.392 | 0.509 | 0.393 | 0.463 | 0.220 | 0.335 | 0.482 | 0.592 | 0.461 | 0.570 | 0.579 | 0.659 | 0.241 | 0.336 | 0.563 | 0.721 |
| **Qwen2.5-3b-Base** | | | | | | | | | | | | | | | | |
| Base | 0.322 | 0.418 | 0.323 | 0.380 | 0.181 | 0.275 | 0.396 | 0.486 | 0.378 | 0.468 | 0.476 | 0.541 | 0.195 | 0.273 | 0.461 | 0.592 |
| o/w $\varepsilon - plan$ | 0.348 | 0.451 | 0.349 | 0.410 | 0.195 | 0.297 | 0.428 | 0.525 | 0.408 | 0.506 | 0.514 | 0.584 | 0.215 | 0.301 | 0.498 | 0.639 |
| o/w $\varepsilon - search$ | 0.362 | 0.468 | 0.361 | 0.426 | 0.203 | 0.308 | 0.443 | 0.544 | 0.423 | 0.524 | 0.532 | 0.606 | 0.218 | 0.307 | 0.517 | 0.663 |
| o/w $\varepsilon - sub\_answer$ | **0.377** | **0.489** | **0.378** | **0.445** | **0.213** | **0.322** | **0.463** | **0.569** | **0.443** | **0.548** | **0.557** | **0.633** | **0.229** | **0.323** | **0.543** | **0.693** |

Table 3: Accuracy on 7b and 3b models. '*w/o*' represent 'without' while '*o/w*' for 'only with'."†" indicates offline retrieval, and "∗" indicates online retrieval.

with disabling it yielding a -1.80% F1 on Musique. Although it alleviates error propagation, it cannot fundamentally prevent erroneous reasoning from emerging.

Table 4 and Figure 4 present the performance of individual erasure mechanisms across different iteration numbers on the Qwen2.5-3b-Instruct model. Plan-triggered erasure shows modest gains with increasing iterations, indicating that planning can reduce localized structural mistakes but is insufficient for errors in longer reasoning chains. Notably, even with an imperfect initial plan, the model can still identify the next required information through further interaction with the external environment. Search-triggered erasure yields more pronounced improvements, especially on retrieval-intensive datasets, highlighting the importance of accurate search queries for maintaining reasoning

| Method | HotpotQA† | | 2Wiki† | | MuSiQue† | | Bamboogle† | | HotpotQA* | | 2Wiki* | | MuSiQue* | | Bamboogle* | |
|---|---|---|---|---|---|---|---|---|---|---|---|---|---|---|---|---|
| | EM | F1 | EM | F1 | EM | F1 | EM | F1 | EM | F1 | EM | F1 | EM | F1 | EM | F1 |
| Base | 0.348 | 0.457 | 0.323 | 0.389 | 0.181 | 0.264 | 0.347 | 0.457 | 0.398 | 0.476 | 0.406 | 0.501 | 0.151 | 0.242 | 0.525 | 0.633 |
| o/w $\varepsilon - plan1$ | 0.354 | 0.464 | 0.328 | 0.395 | 0.183 | 0.268 | 0.353 | 0.464 | 0.405 | 0.483 | 0.412 | 0.509 | 0.153 | 0.245 | 0.534 | 0.642 |
| o/w $\varepsilon - plan3$ | 0.358 | 0.470 | 0.333 | 0.401 | 0.256 | 0.272 | 0.358 | 0.473 | 0.413 | 0.491 | 0.417 | 0.516 | 0.156 | 0.250 | 0.541 | 0.652 |
| o/w $\varepsilon - plan5$ | 0.365 | 0.477 | 0.338 | 0.407 | 0.189 | 0.276 | 0.363 | 0.478 | 0.417 | 0.498 | 0.425 | 0.524 | 0.158 | 0.253 | 0.549 | 0.661 |
| o/w $\varepsilon - search1$ | 0.363 | 0.478 | 0.337 | 0.405 | 0.188 | 0.275 | 0.361 | 0.478 | 0.421 | 0.499 | 0.423 | 0.522 | 0.156 | 0.252 | 0.537 | 0.655 |
| o/w $\varepsilon - search3$ | 0.383 | 0.502 | 0.355 | 0.427 | 0.198 | 0.290 | 0.382 | 0.502 | 0.438 | 0.523 | 0.448 | 0.550 | 0.190 | 0.266 | 0.576 | 0.698 |
| o/w $\varepsilon - search5$ | 0.404 | 0.529 | 0.375 | 0.451 | 0.201 | 0.306 | 0.402 | 0.527 | 0.462 | 0.552 | 0.471 | 0.581 | 0.174 | 0.280 | 0.611 | 0.732 |
| o/w $\varepsilon - sub\_answer1$ | 0.371 | 0.486 | 0.344 | 0.414 | 0.183 | 0.281 | 0.369 | 0.486 | 0.410 | 0.507 | 0.432 | 0.533 | 0.162 | 0.257 | 0.559 | 0.673 |
| o/w $\varepsilon - sub\_answer3$ | 0.399 | 0.521 | 0.371 | 0.445 | 0.207 | 0.302 | 0.398 | 0.523 | 0.456 | 0.545 | 0.465 | 0.573 | 0.175 | 0.277 | 0.597 | 0.724 |
| o/w $\varepsilon - sub\_answer5$ | 0.425 | 0.561 | 0.396 | 0.479 | 0.221 | 0.324 | 0.426 | 0.561 | 0.492 | 0.584 | 0.502 | 0.615 | 0.186 | 0.297 | 0.641 | 0.776 |
| ERL | **0.447** | **0.587** | **0.415** | **0.500** | **0.232** | **0.339** | **0.446** | **0.587** | **0.513** | **0.612** | **0.521** | **0.644** | **0.211** | **0.311** | **0.674** | **0.813** |

Table 4: Qwen2.5-3b-Instruct. '*o/w*' for 'only with', 'sub-answer' represents a process supervision rewards based on intermediate sub-answers."†" indicates offline retrieval using the Wiki-18 knowledge base, and "∗" indicates online retrieval using Google Search.

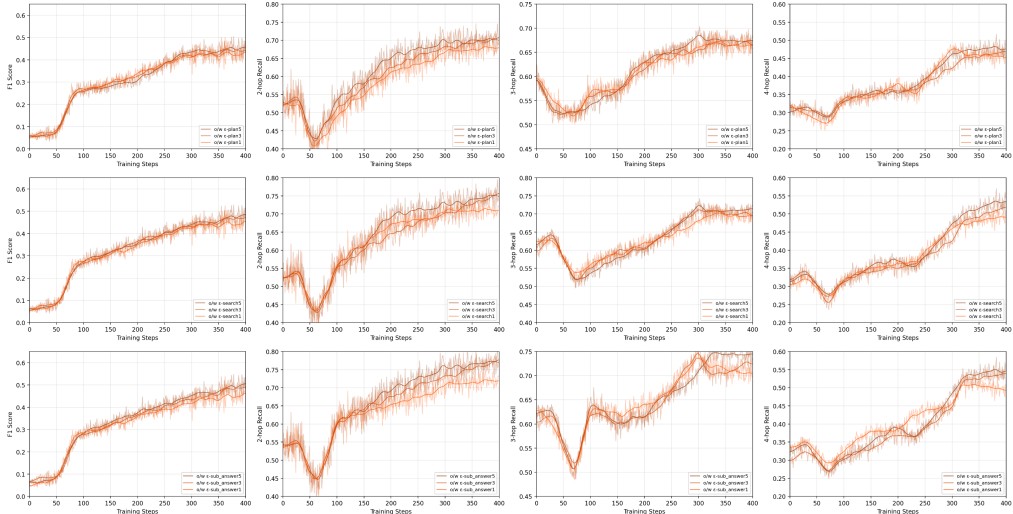

Figure 4: Training dynamics of individual erasure mechanisms across different iteration numbers. In the figure, 'o/w' (only with) indicates that only the current mechanism is added to the base method.

fidelity. Sub-answer-triggered erasure is the most effective, providing consistent gains that approach the full ERL framework's performance as iterations increase, demonstrating that revising intermediate sub-answers significantly mitigates error propagation. Overall, the mechanisms follow a clear hierarchy: sub-answer erasure > search > plan, emphasizing that error correction during reasoning has greater impact than error prevention. Regarding correction rates, ERL exhibits varied effectiveness across error types, correcting 2.01% of decomposition errors, 6.53% of retrieval failures, and 9.6% of reasoning errors.

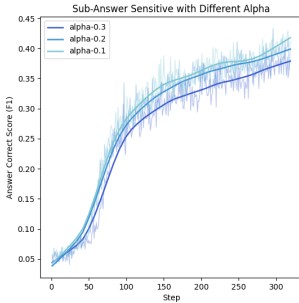 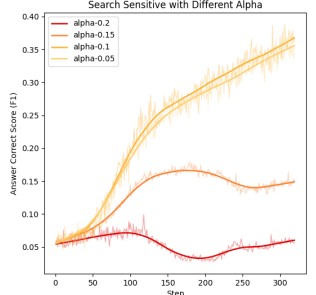 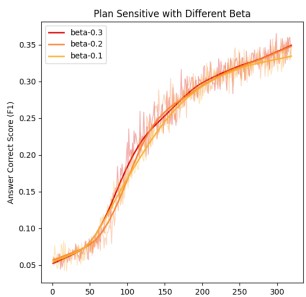

(a) Model performance over training steps. The thresholds $\alpha$ to trigger the *Sub-Answer* erasure are compared between $\{0.1, 0.2, 0.3\}$

(b) Model performance over training steps. The thresholds $\alpha$ to trigger the *Search* erasure are compared between $\{0.05, 0.10, 0.15, 0.20\}$

(c) Model performance over training steps. The thresholds $\beta$ to trigger the *Plan* erasure are compared between $\{0.1, 0.2, 0.3\}$

Figure 5: Model performance over training steps under different erasure trigger thresholds.

To evaluate the stability of ERL, we investigated the sensitivity of three core erasure mechanisms, namely search, planning, and sub-answer modules, to their respective trigger thresholds. The threshold, denoted as $\alpha$ or $\beta$, determines the aggressiveness of the erasure process. A higher threshold requires the module to maintain a higher level of confidence in the retained information, which consequently increases the frequency of erasure. We isolated the impact of each threshold by modifying the values of the target module while disabling the other erasure mechanisms. The resulting variations in Answer F1 scores during training are illustrated in Figure 5. Our analysis indicates that sensitivity is not uniform across different modules, reflecting their distinct roles within the cognitive chain.

The search module exhibits the highest sensitivity. Lower thresholds, such as $\alpha \in 0.05, 0.10$, facilitate rapid convergence and lead to optimal performance. However, when the threshold is increased to $\alpha = 0.2$, a sharp decline in performance is observed. This phenomenon suggests that the search module functions as an information bottleneck for the entire system. An overly aggressive deletion strategy during the retrieval phase leads to information starvation, in which the reasoning engine is deprived of the raw contextual information required to generate accurate sub-answers.

In contrast, the planning module demonstrates strong robustness. Varying $\beta$ within the range of $0.1, 0.2, 0.3$ yields nearly identical learning trajectories. This indicates that downstream search and reasoning processes possess sufficient flexibility to compensate for imperfections in high-level planning, thereby ensuring stable end-to-end performance.

The sub-answer-based erasure mechanism shows moderate and regular sensitivity. As the threshold $\alpha$ increases within $0.1, 0.2, 0.3$, the model maintains a stable convergence trend, although the final Answer F1 scores exhibit a stepwise decrease. This reflects the role of sub-answers as intermediate reasoning nodes, where excessive erasure slightly impedes the effective accumulation of logical information.

## 6 RELATED WORK

Reinforcement learning has been widely applied to enhance retrieval-augmented reasoning in large language models. Search-R1 (Jin et al., 2025b) and ReSearch (Chen et al., 2025) optimize multi-round query generation, R1-Searcher (Song et al., 2025) adopts a two-stage reward, and StepSearch (Zheng et al., 2025a)shapes trajectories with stepwise rewards, while DeepResearcher (Zheng et al., 2025b), $O^2$-Searcher (Mei et al., 2025), and ZeroSearch (Sun et al., 2025) target real webpages, localized environments, and retrieval simulation. MaskSearch (Wu et al., 2025)and EvolveSearch (Zhang et al., 2025a) improve multi-hop reasoning through pretraining or iterative self-evolution, and R-Search (Zhao et al., 2025a) and DynaSearcher (Hao et al., 2025) integrate multi-reward signals with dynamic knowledge graphs. ParallelSearch (Zhao et al., 2025b), HybridDeepSearcher (Ko et al., 2025), and SSRL (Fan et al., 2025) further advance retrieval via parallelization, adaptive strategies, or internal knowledge search.

## 7 LIMITATION & FUTURE DISCUSSION

The strength of the ERL framework lies in its structured cycle of identification, erasure, and regeneration, which enables targeted correction of reasoning errors and significantly improves reliability. This sequential design inherently increases computational overhead and may struggle when multiple heterogeneous errors occur simultaneously within a reasoning trajectory. In such cases, the framework often requires repeated iterations to separately repair failures in retrieval, reasoning, and subsequent retrieval stages, which limits scalability and efficiency. Addressing this challenge calls for strategies that can recognize and resolve multiple concurrent errors in a single corrective pass. Such an advance would require moving beyond localized error signals toward a global understanding of the entire reasoning trajectory, enabling coordinated error mitigation rather than piecemeal correction. Developing this global perspective is not only crucial for enhancing the robustness and efficiency of ERL, but also represents a broader step toward equipping search-augmented language models with genuinely resilient reasoning capabilities.

## 8 CONCLUSION

This paper introduces erasable reinforcement learning algorithm designed to automatically detect and correct decomposition, retrieval, and reasoning errors in complex multi-hop question answering. The method leverages joint signals from the quality of sub-search and sub-answer processes to identify error types, and erases the corresponding segments for regeneration when errors occur, thereby maximizing the utility of both the model and external knowledge. Experimental results demonstrate that our approach surpasses the current state of the art on multi-hop QA benchmarks including HotpotQA, MuSiQue, 2WikiMultiHopQA, and Bamboogle, validating its effectiveness. Future work may explore extending this mechanism to a broader range of generative tasks, or integrating it with online learning to further enhance the model's adaptive error-correction capability.

ETHICS STATEMENT

This study uses only publicly available benchmark datasets (HotpotQA, MuSiQue, 2WikiMulti-HopQA, and Bamboogle), with knowledge sources limited to Wikipedia and the Google Search API. No private or sensitive data are involved. The proposed Erasable Reinforcement Learning (ERL) substantially enhances multi-hop reasoning capabilities, offering positive value for applications such as information retrieval and educational question answering. However, we also recognize that stronger reasoning ability could be misused to generate deceptive or misleading content. We recommend that future research integrate alignment and bias detection mechanisms prior to deployment to mitigate such risks. Overall, this work adheres to established academic ethical standards, balancing capability advancement with responsible use, and aims to contribute to the development of trustworthy artificial intelligence.

REPRODUCIBILITY

We have taken extensive measures to ensure the reproducibility of our work. All datasets used in this study are publicly available benchmarks, including HotpotQA, MuSiQue, 2WikiMultiHopQA, and Bamboogle. For retrieval, we employ both a fixed Wikipedia dump and the Google Search API, and we describe the retrieval setup in detail to enable consistent replication. Implementation details, such as model architectures (Qwen2.5-3B/7B base and instruct), retriever backbone (E5), training hyperparameters, and evaluation metrics (Exact Match and F1), are fully documented. To further facilitate reproducibility, we will release the training scripts, evaluation pipelines, and configuration files required to replicate all results reported in this paper. Random seeds and hardware specifications will also be provided to minimize variance across runs. Our methodology does not rely on proprietary or undisclosed components, ensuring that independent researchers can fully verify and extend our findings.

LLM USAGE

We partially used large language models (LLMs) exclusively for non-scientific writing assistance, specifically for language polishing, clarity improvement, and suggestions. No parts of the core methodology, experiments, or results were generated by LLMs.

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

# A COST–EFFECTIVENESS OF INCREASING THE ERASURE–RETRY BUDGET

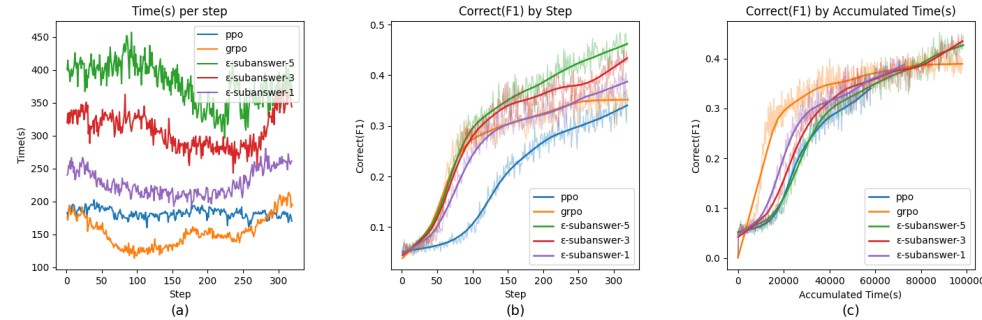

Figure 6: Training efficiency under different Sub-Answer retry budgets. (a) Per-step wall-clock time across PPO, GRPO and Sub-Answer Erasure under different retry budgets. (b) Model performance over training steps. (c) Model performance over cumulative training time.

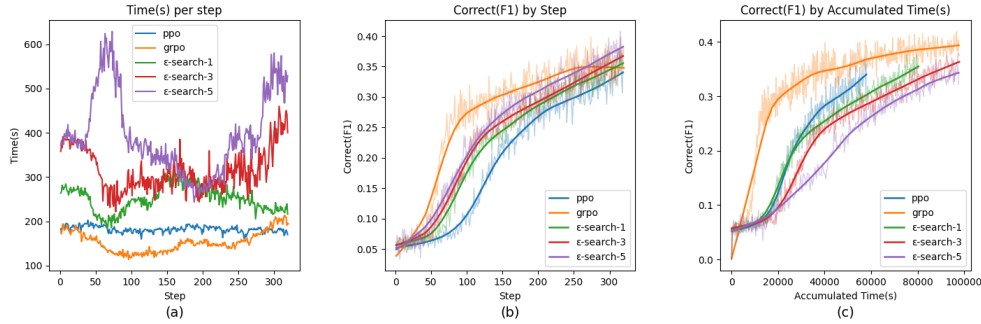

Figure 7: Training efficiency under different Search retry budgets. (a) Per-step wall-clock time across PPO, GRPO and Search Erasure under different retry budgets. (b) Model performance over training steps. (c) Model performance over cumulative training time.

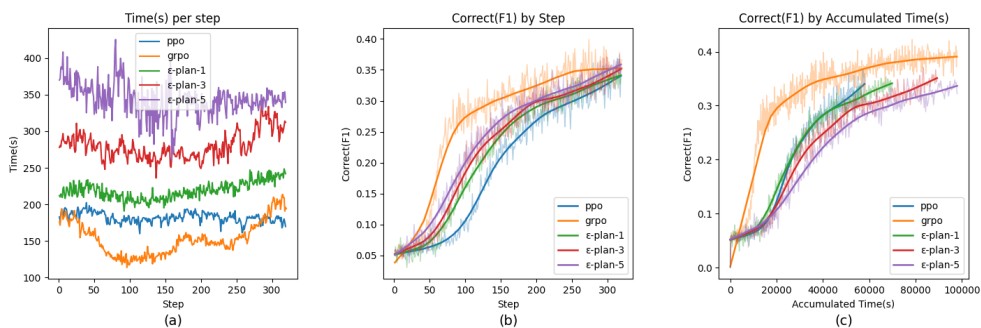

Figure 8: Training efficiency under different Plan retry budgets. (a) Per-step wall-clock time across PPO, GRPO and Plan Erasure under different retry budgets. (b) Model performance over training steps. (c) Model performance over cumulative training time.

To address the concern regarding the computational overhead introduced by longer rollout horizons and more frequent research queries, we conduct an additional experiment that systematically varies the maximum number of allowed erasure retries for each module (*subanswer*, *plan*, and *search*). Since

the erasure mechanism is implemented sequentially, the wall-clock time measured under identical hardware directly reflects both GPU computation and external querying volume. This provides an interpretable and implementation-faithful estimate of the cost of expanding the search-and-repair budget. For each erasure module—SubAnswer-Erase, Plan-Erase, and Search-Erase—we train the model while capping the erasure-retry budget at

$$k \in \{1, 3, 5\}. \tag{21}$$

All other training hyper-parameters, random seeds, dataset order, and hardware remain strictly identical across runs. For each setting we record:

- Per-step wall-clock time (s/step) — directly reflecting computation + querying cost.
- Accuracy-per-step curve — training progress with respect to optimization steps.
- Accuracy-per-time curve — training progress normalized by cumulative runtime, measuring practical training efficiency.

This design allows us to isolate the marginal benefit of higher erasure budgets while quantifying their real-world cost. The experimental results show that applying erasure at different stages produces significantly different effects, primarily reflecting the importance of each stage itself in chain-of-thought reasoning. Applying the erasure mechanism to stages such as "Sub-Answer," which has the greatest impact on the final answer, leads to the most substantial and noticeable improvement in training efficiency. Although the computation time per step becomes longer due to the need for rethinking and responding, the rate of improvement in training metrics per unit of computation time actually increases compared to the original PPO.

Another noteworthy point is that, although GRPO is far more efficient per unit time than the original PPO and PPO optimization algorithms enhanced with the erasure mechanism, its performance ceiling is clearly lower.

## B  DYNAMICS OF ERASURE EVENTS ACROSS MULTI-ROUND REASONING

To further understand how the ERL framework behaves during training, we conduct an additional experiment that analyzes when and how often erasure events occur within a multi-hop reasoning trajectory. Specifically, for each training step, we record the average number of retries triggered by the *Sub-Answer*, *Search* and *Plan* modules under different maximum reasoning depths (*i.e.*, number of rounds). We evaluate four settings with round limits

$$R \in \{1, 2, 3, 4\}, \tag{22}$$

corresponding to increasingly deeper multi-step decomposition strategies. We find that the average frequency of erasing subanswers becomes more pronounced as reasoning deepens. This indicates

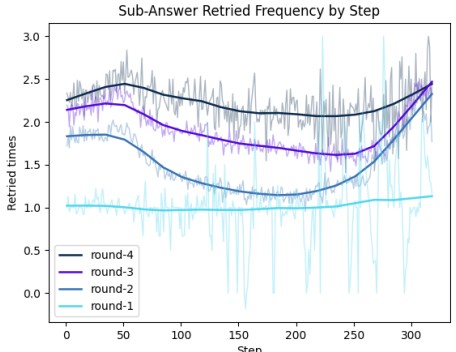 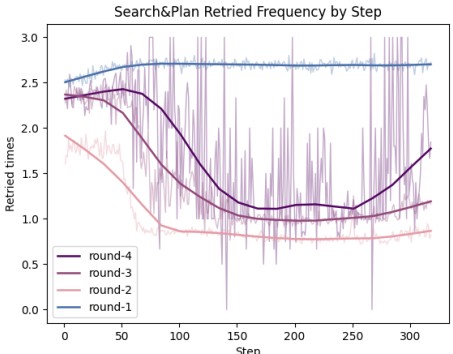

(a) Average frequency of Sub-Answer Erasures occurrence per round with max retry set by 3.

(b) Average frequency of Search&Plan Erasures occurrence per round with max retry set by 3.

Figure 9: Dynamics of erasure events across multi-hop reasoning rounds during training.

that the later stages of the reasoning chain are more likely to produce incorrect subanswers and thus require erasure and retry. Difficult and multistep problems further demonstrate that the accumulation of earlier errors makes subsequent reasoning more challenging. The increased number of retries triggered by such cases fully illustrates the value of the ERL mechanism. A similar phenomenon is observed in the rewriting step during rounds 2, 3, and 4 of the Search process. In contrast, for Plan which specifically includes an erasure mechanism only in the first round, the erasure phenomenon tends to remain stable.

## C   MAIN RESULT WITH CONFIDENCE INTERVALS

| Method | HotpotQA† | | 2Wiki† | | MuSiQue† | | Bamboogle† | | HotpotQA* | | 2Wiki* | | MuSiQue* | | Bamboogle* | |
|---|---|---|---|---|---|---|---|---|---|---|---|---|---|---|---|---|
| | EM↑ | F1↑ | EM↑ | F1↑ | EM↑ | F1↑ | EM↑ | F1↑ | EM↑ | F1↑ | EM↑ | F1↑ | EM↑ | F1↑ | EM↑ | F1↑ |
| **Qwen2.5-3b-Base/Instruct** | | | | | | | | | | | | | | | | |
| Search-R1-base | 0.268±0.004 | 0.355±0.005 | 0.250±0.002 | 0.298±0.002 | 0.078±0.002 | 0.138±0.006 | 0.188±0.011 | 0.281±0.012 | 0.255±0.005 | 0.429±0.007 | 0.377±0.003 | 0.447±0.002 | 0.118±0.006 | 0.186±0.006 | 0.274±0.011 | 0.397±0.013 |
| Search-R1-instruct | 0.304±0.001 | 0.402±0.001 | 0.291±0.002 | 0.351±0.001 | 0.116±0.004 | 0.184±0.005 | 0.240±0.004 | 0.347±0.003 | 0.354±0.002 | 0.436±0.004 | 0.370±0.002 | 0.449±0.002 | 0.129±0.005 | 0.197±0.004 | 0.392±0.007 | 0.513±0.008 |
| ZeroSearch-base | 0.276±0.002 | 0.374±0.003 | 0.255±0.002 | 0.311±0.002 | 0.091±0.005 | 0.166±0.004 | 0.168±0.009 | 0.252±0.010 | 0.320±0.007 | 0.410±0.006 | 0.381±0.009 | 0.465±0.011 | 0.131±0.014 | 0.231±0.015 | 0.355±0.014 | 0.513±0.016 |
| ZeroSearch-instruct | 0.275±0.003 | 0.363±0.001 | 0.250±0.003 | 0.291±0.001 | 0.089±0.003 | 0.146±0.002 | 0.190±0.006 | 0.303±0.005 | 0.357±0.004 | 0.453±0.006 | 0.351±0.007 | 0.446±0.009 | 0.111±0.004 | 0.167±0.003 | 0.427±0.008 | 0.532±0.011 |
| R-Search-instruct-GRPO | 0.313±0.006 | 0.413±0.004 | 0.324±0.017 | 0.366±0.015 | 0.125±0.007 | 0.190±0.011 | 0.246±0.013 | 0.354±0.015 | 0.371±0.008 | 0.465±0.011 | 0.460±0.011 | 0.521±0.014 | 0.139±0.008 | 0.230±0.010 | 0.497±0.015 | 0.627±0.018 |
| R-Search-instruct-PPO | 0.273±0.014 | 0.362±0.015 | 0.278±0.002 | 0.329±0.002 | 0.119±0.006 | 0.177±0.007 | 0.252±0.008 | 0.351±0.011 | 0.392±0.006 | 0.492±0.008 | 0.498±0.006 | 0.560±0.009 | 0.142±0.006 | 0.227±0.007 | 0.493±0.012 | 0.652±0.016 |
| SSRL-instruct | 0.321±0.009 | 0.419±0.010 | 0.298±0.008 | 0.358±0.009 | 0.109±0.009 | 0.171±0.013 | 0.247±0.026 | 0.333±0.040 | 0.366±0.017 | 0.444±0.019 | 0.394±0.010 | 0.496±0.023 | 0.122±0.014 | 0.204±0.016 | 0.389±0.033 | 0.503±0.041 |
| StepSearch-base | 0.327±0.003 | 0.436±0.002 | 0.341±0.004 | 0.393±0.002 | 0.182±0.003 | 0.271±0.004 | 0.322±0.006 | 0.417±0.007 | 0.351±0.006 | 0.473±0.008 | 0.441±0.008 | 0.564±0.010 | 0.201±0.004 | 0.299±0.007 | 0.512±0.011 | 0.652±0.017 |
| StepSearch-instruct | 0.346±0.002 | 0.451±0.001 | 0.320±0.003 | 0.386±0.001 | 0.180±0.003 | 0.260±0.005 | 0.341±0.009 | 0.449±0.010 | 0.396±0.007 | 0.478±0.007 | 0.406±0.010 | 0.501±0.009 | 0.155±0.006 | 0.243±0.009 | 0.520±0.012 | 0.644±0.011 |
| ESearch-base | 0.412±0.005 | 0.546±0.007 | 0.428±0.002 | 0.500±0.002 | 0.234±0.007 | 0.344±0.009 | 0.414±0.011 | 0.529±0.014 | 0.436±0.005 | 0.582±0.010 | 0.577±0.007 | 0.679±0.011 | 0.241±0.009 | 0.362±0.012 | 0.625±0.019 | 0.789±0.022 |
| ESearch-instruct | 0.448±0.004 | 0.589±0.004 | 0.416±0.002 | 0.499±0.003 | 0.233±0.006 | 0.336±0.009 | 0.442±0.009 | 0.585±0.012 | 0.516±0.006 | 0.614±0.009 | 0.522±0.004 | 0.641±0.007 | 0.213±0.008 | 0.313±0.013 | 0.671±0.014 | 0.807±0.018 |

Table 5: The main results. "†" indicates offline retrieval, and "∗" indicates online retrieval.

We have incorporated statistical analysis of confidence intervals into the main experimental results. Table 5 presents the experimental results with confidence intervals, helping to confirm that the performance gains we report are significant and go beyond random noise.

## D   RELATED WORK

Recent research has increasingly explored reinforcement learning (RL) as a means to improve the retrieval and reasoning capabilities of large language models (LLMs)(Jin et al., 2025b; Sun et al., 2025; Zheng et al., 2025a; Zhao et al., 2025a; Chen et al., 2025; Zhao et al., 2025b; Wu et al., 2025; Hao et al., 2025; Zhang et al., 2025a; Zheng et al., 2025b; Song et al., 2025; Ko et al., 2025). A key theme in this literature is the integration of retrieval into multi-step reasoning, often referred to as search–reinforcement learning. We summarize related work along three major dimensions: coupling retrieval with reasoning, reward design for retrieval optimization, and dynamic or structured retrieval strategies.

**Retrieval–Reasoning Coupling**   Several approaches train LLMs to seamlessly integrate retrieval into reasoning trajectories. Search-R1 (Jin et al., 2025b;a) applies RL to enable models to autonomously issue queries during multi-step reasoning, with iterative retrieval interactions guiding trajectory refinement. R1-Searcher (Song et al., 2025) introduces a two-stage training paradigm: a retrieve reward first encourages correct execution of retrieval operations independent of final answers, after which an answer reward incentivizes effective use of retrieved evidence to solve problems. ReSearch (Chen et al., 2025) explicitly regards search as part of the reasoning chain, training LLMs to perform retrieval whenever necessary and incorporate results into subsequent steps. DeepResearcher (Zheng et al., 2025b) pushes this line further by performing end-to-end training on real webpages, showcasing advanced behaviors such as planning, cross-source verification, and self-reflection.

**Reward Design and Training Paradigms**   Another line of work develops specialized environments and reward functions to guide retrieval. O2-Searcher constructs a localized search environment with carefully designed rewards to address both open-domain and closed-domain tasks. ZeroSearch (Sun et al., 2025) reduces training costs by simulating retrieval while maintaining comparable effectiveness to real search engines. StepSearch (Zheng et al., 2025a) introduces fine-grained step-level rewards, such as information gain and redundancy penalties, within PPO (Schulman et al., 2017) training to progressively refine search behaviors. MaskSearch (Wu et al., 2025) augments pretraining with retrieval-based masked prediction tasks, teaching models to leverage search tools to fill textual gaps and thereby improving multi-hop QA. EvolveSearch (Zhang et al., 2025a) integrates supervised

fine-tuning (SFT) with RL in an iterative self-evolution framework, continually improving multi-hop retrieval without requiring annotated reasoning data.

**Dynamic and Structured Retrieval Strategies**   Recent studies emphasize adaptive control over retrieval behaviors and the exploitation of structured query patterns. R-Search (Zhao et al., 2025a) employs multi-reward RL to dynamically decide when to retrieve versus when to reason, while integrating multi-turn results to enhance answers for knowledge- and logic-intensive tasks. DynaSearcher (Hao et al., 2025) leverages dynamic knowledge graphs and multi-reward RL to maintain consistency in retrieval and improve output quality. ParallelSearch (Zhao et al., 2025b) identifies decomposable query structures and executes multiple subqueries in parallel. HybridDeepSearcher (Ko et al., 2025) combines parallel and sequential retrieval modes, selecting the most suitable strategy based on problem characteristics. Finally, SSRL (Fan et al., 2025) explores retrieval grounded in a model's internal knowledge base, thereby reducing dependence on external search engines.

# E   DATASETS

We selected four benchmark datasets designed based on multi-hop questions: HotpotQA (Yang et al., 2018), 2WikiMultiHopQA (Ho et al., 2020), Musique(Trivedi et al., 2022), and Bamboogle (Press et al., 2023).

**HotpotQA**(Yang et al., 2018): HotpotQA was introduced to address the limitations of earlier QA datasets, which mostly focused on single-paragraph reasoning and lacked explicit supervision for multi-hop reasoning. HotpotQA aimed to build a large-scale dataset requiring reasoning across multiple documents, while also supporting explainable predictions. To achieve this, they crowd-sourced over 112k question–answer pairs based on Wikipedia, ensuring that questions required integrating information from more than one article. A key innovation was the collection of supporting facts—sentence-level evidence for answers—allowing models not only to find the correct response but also to explain it. Additionally, HotpotQA includes a novel class of comparison questions, which require systems to compare two entities on shared properties such as dates or numerical values. The dataset was split into train-easy (18,089), train-medium (56,814), train-hard (15,661), dev (7,405), and two test sets (7,405 each: distractor and fullwiki).

**2WikiMultiHopQA(2Wiki)**(Ho et al., 2020): The 2Wiki dataset is a large-scale multi-hop question answering benchmark created from Wikipedia and Wikidata. It aims to evaluate reasoning by requiring models to integrate information across multiple documents. Unlike earlier datasets, it provides explicit evidence paths in the form of triples, which both enhance interpretability and allow direct evaluation of reasoning skills. The construction process involved designing templates, applying logical rules, and filtering to guarantee multi-hop reasoning. Four question types are included: comparison, inference, compositional, and bridge-comparison, ensuring diversity and difficulty. In total, the dataset contains 192,606 examples, split into 167,454 for training, 12,576 for development, and 12,576 for testing. This scale makes it significantly larger than many prior multi-hop QA datasets. Human performance remains much higher than model baselines, showing the dataset's value as a challenging benchmark for machine reasoning.

**Musique**(Trivedi et al., 2022): The MuSiQue dataset was created to address the limitations of existing multi-hop question answering benchmarks. Musique proposed a bottom-up construction method: they carefully composed multi-hop questions from single-hop questions sourced from several Wikipedia-based datasets. The dataset consists of two main variants: MuSiQue-Ans, containing about 25,000 2–4 hop questions, and MuSiQue-Full, which doubles this size by adding contrastive unanswerable questions, resulting in 50,000 samples. Specifically, MuSiQue-Ans is split into 19,938 training, 2,417 development, and 2,459 test questions, with balanced distributions across different hop lengths. These features make MuSiQue a challenging and less "cheatable" benchmark, pushing research toward genuine multi-hop reasoning.

**Bamboogle**(Press et al., 2023): The Bamboogle dataset was introduced to address the limitations of existing question answering benchmarks, where many compositional questions cannot be answered with a single Google query because the necessary information is dispersed across multiple sources. Unlike prior datasets that often focus on single-hop fact retrieval, Bamboogle emphasizes multi-hop factual reasoning. It requires models to integrate multiple entities and relations to arrive at the correct answer. In terms of scale, the benchmark contains a test set of 125 questions, which are carefully

annotated to evaluate the model's ability to identify and use intermediate entities (bridging objects) during reasoning.

## F  EXPERIMENT SETUPS

Our implementation builds upon Search-R1 (Jin et al., 2025b) and StepSearch (Zheng et al., 2025a), with training performed using Verl (Sheng et al., 2025). We evaluate two model variants, Qwen-2.5-3B and Qwen-2.5-7B (Yang et al., 2024). We use the 2018 Wikipedia(Wiki-18) (Karpukhin et al., 2020) dump and E5 (Wang et al., 2022) as the knowledge base and retriever. For training, we utilize the MuSiQue dataset processed through our training, while evaluation is conducted on the full test or validation splits of 2Wiki, Bamboogle, HotpotQA, and MuSiQue. Both EM and F1 are reported as evaluation metrics. Training runs for 500 steps in total. The learning rates are set to $5 \times 10^{-7}$ for the policy model and $5 \times 10^{-6}$ for the value model, with warm-up ratios of 0.285 and 0.015, respectively. Experiments are executed across two nodes equipped with 16 H800 GPUs. We configure the total, mini-batch, and micro-batch sizes as 512, 64, and 16. To improve memory efficiency, we apply Fully Sharded Data Parallel (FSDP) with CPU offloading, fixing the GPU memory utilization ratio at 0.7.

For rollout sampling, we set both the temperature and top_p to 1.0. The KL-divergence regularization coefficient ($\beta$) and clipping ratio are set to $1 \times 10^{-3}$ and 0.2, respectively.

## G  PROMPT FOR RESEARCH PLAN ON QUESTION ANSWERING

> **Template for ESEARCH.**
>
> You are an expert AI assistant with search engine access. When answering complex questions, you need to decompose them into sub-questions and reason step by step. For each sub-question: provide concise search terms between <search> and </search>; the search results will be placed between <information> and </information>; conduct thorough analysis and reasoning in <observation> and </observation>; then output a concise conclusion in <sub_answer> and </sub_answer>. If you find that all sub-questions have been solved, you should directly provide the final answer inside <answer> and </answer> without detailed illustrations. For example, <answer> xxx </answer>.
>
> ---
>
> *Question:{question}*

Figure 10: LLM interacts with external search engines and provides answers to prompt templates. The *{question}* will be replaced with the actual question content.

## H  INCORRECT FORM

Esearch errors can be broadly categorized into three types. First, premature observations occur when the system concludes too quickly in the observation step without fully leveraging the available evidence, as shown in Table 13. Second, retrieval errors occur when the system fails to retrieve the correct documents, often due to imprecise or poorly formulated queries, as shown in Table 14. Third, entity alignment or localization errors arise when the correct document is retrieved. Still, the model fails to identify and ground the right entity within it, as shown in Table 15. These error types are the main impact of failures in retrieval, entity alignment, and observation, undermining multi-hop question answering. In the actual training process, we observed that observation errors decrease steadily with training steps, while retrieval errors also decline but at a much slower rate compared to observation errors. This further reveals that the training is hindered by the limited capabilities of the locally deployed search engine based on Wiki-18.

# I    COMPARE WITH TRADITIONAL METHODS

| Method | HotpotQA | | 2Wiki | | MuSiQue | | Bamboogle | |
|---|---|---|---|---|---|---|---|---|
| | EM | F1 | EM | F1 | EM | F1 | EM | F1 |
| **Qwen2.5-3b-Base/Instruct** | | | | | | | | |
| Direct Inference | 0.167 | 0.214 | 0.263 | 0.308 | 0.014 | 0.095 | 0.038 | 0.099 |
| CoT | 0.037 | 0.099 | 0.016 | 0.094 | 0.009 | 0.067 | 0.179 | 0.234 |
| IRCoT | 0.077 | 0.135 | 0.137 | 0.197 | 0.058 | 0.141 | 0.221 | 0.305 |
| Search-o1 | 0.204 | 0.287 | 0.230 | 0.293 | 0.047 | 0.126 | 0.336 | 0.397 |
| RAG | 0.285 | 0.366 | 0.192 | 0.271 | 0.089 | 0.148 | 0.303 | 0.364 |
| SFT | 0.197 | 0.252 | 0.158 | 0.243 | 0.077 | 0.139 | 0.100 | 0.181 |
| R1-base | 0.239 | 0.294 | 0.262 | 0.317 | 0.070 | 0.127 | 0.246 | 0.303 |
| R1-instruct | 0.194 | 0.279 | 0.239 | 0.327 | 0.085 | 0.151 | 0.204 | 0.297 |
| Esearch-base* | 0.415 | 0.548 | **0.428** | 0.499 | **0.236** | **0.345** | 0.414 | 0.529 |
| Esearch-instruct* | **0.447** | **0.587** | 0.415 | **0.500** | 0.232 | 0.339 | **0.446** | **0.587** |
| **Qwen2.5-7b-Base/Instruct** | | | | | | | | |
| Direct Inference | 0.201 | 0.248 | 0.238 | 0.319 | 0.019 | 0.106 | 0.107 | 0.191 |
| CoT | 0.079 | 0.165 | 0.127 | 0.184 | 0.035 | 0.094 | 0.214 | 0.299 |
| IRCoT | 0.121 | 0.206 | 0.133 | 0.218 | 0.055 | 0.143 | 0.237 | 0.296 |
| Search-o1 | 0.206 | 0.257 | 0.189 | 0.246 | 0.045 | 0.132 | 0.281 | 0.366 |
| RAG | 0.317 | 0.375 | 0.221 | 0.307 | 0.084 | 0.144 | 0.273 | 0.361 |
| SFT | 0.233 | 0.287 | 0.277 | 0.328 | 0.080 | 0.138 | 0.124 | 0.178 |
| R1-base | 0.212 | 0.301 | 0.229 | 0.315 | 0.066 | 0.157 | 0.277 | 0.335 |
| R1-instruct | 0.254 | 0.314 | 0.304 | 0.361 | 0.060 | 0.146 | 0.266 | 0.329 |
| Esearch-base* | 0.434 | 0.564 | **0.436** | **0.513** | **0.244** | **0.371** | **0.534** | **0.656** |
| Esearch-instruct* | **0.442** | **0.576** | 0.419 | 0.494 | 0.241 | 0.358 | 0.458 | 0.612 |

Table 6: Comparison of ESEARCH with traditional non-reinforcement learning methods on four multi-hop Q&A datasets, reported with Word-level F1 and Exact Match (EM) scores using Wiki-18 as search engine. The best results are highlighted in bold.

# J    NUMBER OF RETRIEVED K DOCUMENTS

Table 7 shows the effect of varying the number of top-K on the 3B model. A single document ($k = 1$) leads to the lowest performance across all datasets, indicating insufficient evidence for multi-hop reasoning. Three documents ($k = 3$) yield the most reliable improvements and deliver the strongest overall results. Increasing the retrieval to five ($k = 5$) produces different outcomes: in some cases, such as Bamboogle, performance is close to $k = 3$, while in others it is slightly degraded. Therefore, excessive retrieval may introduce irrelevant or distracting information, bring more confusion to the model. When $k = 3$ indicates the most robust choice, while $k = 5$ can offer marginal gains only in particular datasets.

| Top-K | HotpotQA | | 2Wiki | | MuSiQue | | Bamboogle | |
|---|---|---|---|---|---|---|---|---|
| | EM | F1 | EM | F1 | EM | F1 | EM | F1 |
| **Qwen2.5-3b-Base** | | | | | | | | |
| 1 | 0.379 | 0.508 | 0.415 | 0.481 | 0.206 | 0.332 | 0.329 | 0.447 |
| 3 | **0.415** | **0.548** | **0.428** | **0.499** | 0.236 | **0.345** | 0.414 | 0.529 |
| 5 | 0.389 | 0.518 | 0.406 | 0.491 | **0.245** | 0.343 | **0.424** | **0.573** |
| **Qwen2.5-3b-Instruct** | | | | | | | | |
| 1 | 0.409 | 0.546 | 0.364 | 0.457 | 0.192 | 0.316 | 0.373 | 0.496 |
| 3 | **0.447** | **0.587** | **0.415** | **0.500** | **0.232** | **0.339** | 0.446 | 0.587 |
| 5 | 0.455 | 0.603 | 0.374 | 0.469 | 0.208 | 0.304 | **0.448** | **0.603** |

Table 7: Impact of the Number of Retrieved Documents (Top-k) on Answer Accuracy. (ESEARCH on Qwen2.5-3B with updated results). Bold indicates the best performance.

# K   CASE STUDY

---

**Case study for bad planning recovery in 2-hops question during training**

**Question:** Who in the city where Chopin grew up has the power of legislative action?

**Ground Truth:** ['Warsaw City Council']
**Sub-answers:** ['Warsaw, Poland','Warsaw City Council']

---

**ERL:**
To answer this question, we need to break it down into sub-questions:
1. Identify the city where Chopin was born.
2. Determine which organization has legislative power in that city.
Let's address these sub-questions one by one.
<search> Chopin birthplace city </search>
<information> Doc 1<## Title: "Frédéric Chopin" ##> Frédéric François Chopin was born on 1 March 1810 in Żelazowa Wola, a village in the Duchy of Warsaw ... moved to Warsaw when he was seven months old ...
Doc 2<## Title: "Chopin family" ##> The Chopin family moved to Warsaw in October 1810, when Frédéric was seven months old ... grew up in Warsaw, attending school there ...
Doc 3<## Title: "Early life of Chopin" ##> Though born in Żelazowa Wola, Chopin was raised in Warsaw from infancy and considered Warsaw his hometown ... </information>
<observation> The city where Chopin was born is Zelazowa Wola, a village in the Duchy of Warsaw.</observation>
<sub_answer> Warsaw, Poland </sub_answer>
 Sub-answer is incorrect! It will trigger an erasure and regeneration of obervation! 
- - - - - - - - - - - - - - - - - - - - - - - - - - - - - - - - - - - - - - - - - - - - - - - - - - - -
**Erasure and Regeneration:**
<observation> Although Chopin was born in Żelazowa Wola, he grew up in Warsaw, which he considered his hometown. </observation>
<sub_answer> Zelazowa Wola </sub_answer>
- - - - - - - - - - - - - - - - - - - - - - - - - - - - - - - - - - - - - - - - - - - - - - - - - - - -
Now that we know Chopin grew up in Warsaw, we can move on to the second sub-question.
<search> legislative power Warsaw Poland government </search>
<information> Doc 1<## Title: "Warsaw City Council" ##> Warsaw City Council is a unicameral governing body of the city of Warsaw, the capital of Poland. It consists of 60 councilors elected in free elections for a four-year term and has legislative powers over local matters.
Doc 2<## Title: "Government of Warsaw" ##> Legislative power in Warsaw is vested in a unicameral Warsaw City Council ("Rada Miasta"), which comprises 60 members elected for four-year terms ...
Doc 3<## Title: "Warsaw" ##> The city council of Warsaw exercises legislative authority within the city limits, passing local ordinances and budget decisions ... </information>
<observation> The Warsaw City Council has legislative power in Warsaw. </observation>
<sub_answer> Warsaw City Council </sub_answer>
- - - - - - - - - - - - - - - - - - - - - - - - - - - - - - - - - - - - - - - - - - - - - - - - - - - -
The organization that has legislative power in the city where Chopin grew up (Warsaw, Poland) is <answer> Warsaw City Council </answer>

---

Figure 11: Case study demonstrating *error recovery* where initial planning focused on birthplace rather than where Chopin grew up, but observations corrected the understanding to identify Warsaw as his hometown.

---

**Case study for Esearch in 2-hops question**

**Question:** Who is the spouse of the person who won the popular vote in the 2000 election?

---

**Ground Truth:** ['Tipper Gore']
**Sub-answers:** ['Al Gore', 'Tipper Gore']

---

**ERL:**
To answer this question, we need to follow these steps:
1. Identify the person who won the electoral vote in the 2000 U.S. presidential election.
2. Then find out who that person's spouse is.
<search> Winner of electoral vote in the 2000 election </search>
<information>
 No useful information related are retrieved!
</information>
 Plan is incorrect! It will trigger an erasure and regeneration of plan!
- - - - - - - - - - - - - - - - - - - - - - - - - - - - - - - - - - - - - - - - -

**Erasure and Regeneration:**
To answer this question, we need to break it down into sub-questions:
1. Identify the person who won the popular vote in the 2000 election.
2. Determine the spouse of that person. Let's address these sub-questions one by one.
<search> popular vote in the 2000 election (Prussia)? </search>
<information> Doc 1<## Title: "2000 United States presidential election" ##>The 2000 United States presidential election was the 54th quadrennial presidential election...
Doc 2<## Title: "2000 United States presidential election" ##> slate of electors on December 12, should the dispu...
Doc 3<## Title: "2000 United States elections" ##> The 2000 United States elections were held on ...
</information>
<observation> The person who won the popular vote in the 2000 election was Al Gore.</observation>
<sub_answer>Al Gore</sub_answer>
- - - - - - - - - - - - - - - - - - - - - - - - - - - - - - - - - - - - - - - - -
Now that we know the person who won the popular vote in the 2000 election, we can move on to the second sub-question.
<search> spouse of Al Gore</search>
<information> Doc 1<## Title: "Pauline LaFon Gore" ##> Pauline LaFon Gore Pauline LaFon Gore (October 6, 1912 – December 15, 2004) was the mother of former United States Vice President Al Gore and the wife of ... Doc 2<## Title: "Tipper Gore" ##> Tipper Gore Mary Elizabeth ""Tipper"" Gore (""née"" Aitcheson; born August 19, 1948) is an American social issues advocate who was Second Lady of the United States from 1993 to 2001, and the wife of Al Gore, the 45th Vice ... Doc 3<## Title: "Tipper Gore" ##> former editor of ""National Geographic"". Tipper Gore is the author of a number of books including: She has also contributed to the following books: ...</information>
</information>
<observation> Doc 3 clearly mentioned that the spouse of Al Gore is Tipper Gore</observation>
<sub_answer>Tipper Gore</sub_answer>
- - - - - - - - - - - - - - - - - - - - - - - - - - - - - - - - - - - - - - - - -
Therefore, the spouse of the person who won the popular vote in the 2000 election is<answer> Tipper Gore </answer>

Figure 12: Step-by-step reasoning for a 2-hop question identifying the spouse of the winner of the popular vote in the 2000 U.S. presidential election. The example demonstrates how breaking the problem into sub-questions and using evidence from multiple sources leads to the correct answer

---

**Case study for Esearch in 2-hops question (with incorrect observation)**

**Question:** Who wrote "The Road to City A", with City A being the location where KFAB broadcasts?

---

**Ground Truth:** ['Robert Ludlum']
**Sub-answers:** ['Omaha', 'Robert Ludlum']

---

**Solution:**
To answer this question, we need to break it down into sub-questions:
1. Identify where City A is, where KFAB broadcasts.
2. Determine who wrote "The Road to City A" with City A being the location identified above.
Let's address these step by step:
<search> KFAB broadcasts </search>
<information> Doc 1<## Title: KFAB ##> KFAB (1110 AM) is a 50,000 watt clear channel news and talk radio station licensed to Omaha, Nebraska...
</information>
<observation> KFAB broadcasts in Omaha, Nebraska.</observation>
<sub_answer>Omaha, Nebraska</sub_answer>
- - - - - - - - - - - - - - - - - - - - - - - - - - - - - - - - - - - - - - - - - - - - - - - - - - - -
Now that we know City A, we move to the actor:
<search> The Road to Omaha actor </search>
<information> Doc 1<## Title: KFAB ##> ... (no relevant info about the book or author)
Doc 2<## Title: Paul Newman ##> Paul Newman was an American actor, film director, and philanthropist, known for his roles in classic films such as...
</information>
Search is incorrect! It will trigger an erasure and regeneration of Search!
- - - - - - - - - - - - - - - - - - - - - - - - - - - - - - - - - - - - - - - - - - - - - - - - - - - -
**Erasure and Regeneration:**
<search> The Road to Omaha author </search>
<information> Doc 1<## Title: KFAB ##> ... (no relevant info about the book or author)
Doc 2<## Title: Robert Ludlum ##> Robert Ludlum was an American author of 27 thriller novels, best known as the creator of Jason Bourne from the original The Bourne Trilogy...
</information>
<observation> Based on observation, Doc 2 has the correct answer, Robert Ludlum</observation>
<sub_answer>Robert Ludlum </sub_answer>
- - - - - - - - - - - - - - - - - - - - - - - - - - - - - - - - - - - - - - - - - - - - - - - - - - - -
Based on the analysis, the writer of "The Road to Omaha" is <answer> Robert Ludlum </answer>

---

Figure 13: Esearch makes an incorrect observation: although the first hop (KFAB → Omaha) is correct, the retrieval step for the author did not return documents mentioning *Robert Ludlum*. Lacking the correct evidence, the system hallucinates "Jack Elliott" as the author, yielding a wrong final answer.

Case study for Esearch in 2-hops question (with incorrect retrieval)

**Question:** Who released the Fantasy Land Tour 2004 album, performed live in the same city where Alchip is headquartered?

**Ground Truth:** ['S.H.E']
**Sub-answers:** ['Taipei', 'S.H.E']

**Solution:**
To answer this question, we need to break it down into sub-questions:
1. Identify the city where Alchip is headquartered.
2. Determine which performer released the Fantasy Land Tour 2004 album and performed it live in that city.
Let's address each sub-question:
<search> Alchip headquarters city </search>
<information> Doc 1<## Title: Alchip ##> Alchip is a fabless semiconductor company founded in 2003 and headquartered in Taipei, Taiwan...
</information>
<observation> Alchip is headquartered in Taipei, Taiwan.</observation>
<sub_answer>Taipei</sub_answer>
------------------------------------------------------------
Now that we know the city, we move on to the performer:
<search> performer who released the Fantasy Land Tour 2004 album and performed in Taipei </search>
<information> Doc 1<## Title: J1 World Tour ##> J1 World Tour was the debut concert tour by Taiwanese singer Jolin Tsai...
Doc 2<## Title: Play World Tour ##> Play World Tour was the fourth concert tour by Jolin Tsai...
Doc 3<## Title: Castle (Jolin Tsai album) ##> Castle is the sixth studio album by Jolin Tsai, released in 2004...
</information>
<observation> The performer is Jolin Tsai (INCORRECT).</observation>
<sub_answer>Jolin Tsai</sub_answer>
------------------------------------------------------------
Based on the analysis, the performer who released the Fantasy Land Tour 2004 album, recorded live in Taipei, is <answer> Jolin Tsai (INCORRECT) </answer>

Figure 14: Esearch makes an incorrect entity alignment: although the retrieval step surfaced the correct document mentioning *Fantasy Land Tour 2004* by S.H.E, the system failed to align the entity properly and instead associated the album with Jolin Tsai. This shows a case where the key evidence was present but reasoning and entity grounding went wrong, leading to a misleading yet internally consistent answer.

---

**Case study for Esearch in 2-hops question (with incorrect entity alignment)**

**Question:** Banir is located in which country, and in what year did the country achieve unification?

---

**Ground Truth:** ['1963']
**Sub-answers:** ['Malaysia', '1963']

---

**Solution:**
To answer this question, we need to break it down into sub-questions:
1. Identify the country Banir is located in.
2. Determine the year of the country's unification.
Let's address each sub-question:
<search> Banir location </search>
<information> Doc 1<## Title: Banior ##> Banior is a village and gram panchayat in Nalhati I Block in Rampurhat subdivision of Birbhum District in the Indian state of West Bengal...
Doc 2<## Title: Banagher ##> Banagher is a town in Republic of Ireland, located in the midlands, on the western edge of County Offaly in the province of Leinster, on the banks of the River Shannon...
Doc 3<## Title: Banhpur ##> Banhpur is a village in Bulandshahr District, Uttar Pradesh, India. It is situated near Kuchesar Fort...
</information>
<observation> Banir is located in Ireland (INCORRECT).</observation>
<sub_answer>Ireland</sub_answer>
------------------------------------------------------------
Now that we know Banir is in Ireland, we need to determine the year of Ireland's unification:
<search> Ireland unification year </search>
<information> Doc 1<## Title: "United Ireland" ##> United Ireland refers to the proposition that the whole of Ireland should be a single sovereign state... Partition since May 1921...
Doc 2<## Title: "Partition of Ireland" ##> The partition of Ireland divided the island into Northern Ireland and Southern Ireland on 3 May 1921...
Doc 3<## Title: "Irish Free State" ##> The Irish Free State was established in 1922 under the Anglo-Irish Treaty...
</information>
<observation> Ireland was united in 1922 (INCORRECT).</observation>
<sub_answer>1922</sub_answer>
------------------------------------------------------------
Based on the analysis, the year of the country Banir is located in (Ireland) happening unification is <answer> 1922 (INCORRECT) </answer>

---

Figure 15: Esearch also produces an erroneous observation reasoning chain: Banir was incorrectly matched to Banagher (Because the candidate document doesn't contain the target answer, the model makes a close spelling error and misinterprets "Banagher" as Banir) instead of Banir (Malaysia), leading to the wrong unification year (1922 instead of 1963).

---

**Case study for Esearch in 3-hops question**

**Question:** Who established the first committee of correspondence in 1772 in the capital of the state where Wellesley College is located, and why?

---

**Ground Truth:** ['Samuel Adams']
**Sub-answers:** ['Massachusetts', 'Boston', 'Samuel Adams']

---

**ERL:**
To answer this question, we need to break it down into sub-questions:
1. Identify the state where Wellesley College is located.
2. Determine the capital of that state.
3. Find out who established the first committee of correspondence in 1772 in that capital.
4. Understand the reason for establishing the committee.
Let's address each sub-question:

<search> Wellesley College location </search>
<information> Doc 1<## Title: "Wellesley, Massachusetts" ##> Wellesley is a town in Norfolk County, Massachusetts... It is best known as the home of Wellesley College...
Doc 2<## Title: "Wellesley College" ##> Wellesley College is a private women's liberal arts college located west of Boston in the town of Wellesley, Massachusetts...
</information>
<observation> Wellesley College is located in the state of Massachusetts.</observation>
<sub_answer> Massachusetts </sub_answer>

------------------------------------------------------------

Now we know the state, we can find the capital:
<search> capital of Massachusetts, United States </search>
<information> Doc 1<## Title: Massachusetts ##> Boston is the state capital and largest city in Massachusetts...
Doc 2<## Title: Massachusetts ##> The capital of Massachusetts is Boston...
</information>
<observation> The capital of Massachusetts is Boston.</observation>
<sub_answer> Boston </sub_answer>

------------------------------------------------------------

Next, we need to find out who established the first committee of correspondence in 1772 in Boston and why:
<search> first committee of correspondence Boston 1772 established by </search>
<information> Doc 3<## Title: "Committees of correspondence" ##> In Massachusetts, in November 1772, Samuel Adams and Dr. Joseph Warren formed a committee... in relation to the recent British decision to have the salaries of the royal governor and judges be paid by the Crown rather than the colonial assembly...
</information>
<observation> Samuel Adams established the first committee of correspondence in Boston in 1772.</observation>
<sub_answer> Samuel Adams </sub_answer>

------------------------------------------------------------

The reason for establishing the committee was to rally opposition and organize resistance against British reforms, particularly the Crown paying governors and judges directly, which undermined colonial self-governance.

Therefore, the person who established the first committee of correspondence in 1772 in Boston, the capital of Massachusetts, was <answer> Samuel Adams </answer>

Figure 16: Case study showing step-by-step reasoning with sub-questions leading to the identification of Samuel Adams as the founder of the first committee of correspondence in 1772 in Boston.

Case study for Esearch on 3-hops question

**Question:** What is the weekly publication in Jeanne Flanagan's birthplace that is issued by the school attended by the author of "America-Lite: How Imperial Academia Dismantled Our Culture"?

**Ground Truth:** ['Yale Herald']
**Sub-answers:** ['David Gelernter', 'Yale University', 'New Haven', 'Yale Herald']

**Solution:**
To answer this question, we need to break it down into sub-questions:
1. Identify the author of the book "America-Lite: How Imperial Academia Dismantled Our Culture".
2. Determine the school attended by this author.
3. Find the weekly publication in Jeanne Flanagan's birthplace that is issued by the school identified in step 2.
Let's address each sub-question:
<search> author of the book "America-Lite: How Imperial Academia Dismantled Our Culture" </search>
<information> Doc 1<## Title: America-Lite ##> America-Lite America-Lite: How Imperial Academia Dismantled Our Culture (and Ushered in the Obamacrats) is a 2012 book by David Gelernter, published by Encounter Books...
</information>
<observation> The author of the book "America-Lite: How Imperial Academia Dismantled Our Culture" is David Gelernter.</observation>
<sub_answer>David Gelernter</sub_answer>
------------------------------------------------------------
Now that we know the author, we need to find out which school he attended:
<search> school attended by David Gelernter </search>
<information> Doc 1<## Title: "David Gelernter" ##> patriotism and the traditional family, writing "Gelernter is Jewish, and it is not likely that a non-Jew would airily argue that obnoxious leftist Jews have taken over elite higher education..."
</information>
<observation> David Gelernter attended Yale University.</observation>
<sub_answer>Yale University</sub_answer>
------------------------------------------------------------
Finally, we need to find the weekly publication in Jeanne Flanagan's birthplace (which is Boston, Massachusetts) issued by Yale University:
<search> weekly publication in Boston, Massachusetts issued by Yale University </search>
<information> Doc 1<## Title: "The Daily Campus" ##> The Daily Campus The Daily Campus, founded in 1896, is a student-run newspaper at the University of Connecticut that has a circulation run of 10,000 copies weekdays during the school year...
</information>
<observation> The weekly publication in Boston, Massachusetts issued by Yale University is The Yale Herald.</observation>
<sub_answer>The Yale Herald</sub_answer>
------------------------------------------------------------
Based on the analysis, the weekly publication in Jeanne Flanagan's birthplace (Boston, Massachusetts) that is issued by the school attended by the author of "America-Lite: How Imperial Academia Dismantled Our Culture" (Yale University) is <answer>Yale Herald</answer>

Figure 17: Complex multi-step reasoning requiring identification of author, educational background, and geographic publication details.

---

**Case study for Esearch in 4-hops question**

**Question:** The director of "The Mystic Masseur" was born in a city that is in the same country as Kattalai, a city known for its famous temple. In which state of India is Kattalai located?

---

**Ground Truth:** ['Tamil Nadu']
**Sub-answers:** ['Ismail Merchant', 'Bombay', 'India', 'Tamil Nadu']

---

**Solution:**
To answer this question, we need to break it down into sub-questions:
1. Identify the director of The Mystic Masseur.
2. Determine the country where the director was born.
3. Find out in which state Kattalai, a city, is located in that country.
Let's address each sub-question:
<search> director of The Mystic Masseur </search>
<information> Doc 1<## Title: "The Mystic Masseur" ##> The Mystic Masseur The Mystic Masseur is a 2001 Merchant Ivory film based on the novel of the same name by V. S. Naipaul. It is one of relatively few films directed by Ismail Merchant, who is better known as the producer in the Merchant Ivory partnership...
</information>
<observation> The director of The Mystic Masseur is Ismail Merchant.</observation>
<sub_answer>Ismail Merchant</sub_answer>
------------------------------------------------------------
Now we know the director, we need to find out in which country he was born:
<search> Ismail Merchant birth country </search>
<information> Doc 1<## Title: "Ismail Merchant" ##> Ismail Merchant Ismail Merchant (25 December 1936 – 25 May 2005) was an Indian-born film producer and director. He worked for many years in collaboration with Merchant Ivory Productions... Born Ismail Noor Md. Abdul Rahman in Bombay (Mumbai) on Christmas Day 1936...
</information>
<observation> Ismail Merchant was born in India.</observation>
<sub_answer>India</sub_answer>
------------------------------------------------------------
Finally, we need to find out in which state Kattalai, a city, is located in India:
<search> Kattalai, India state </search>
<information> Doc 1<## Title: "Kattalai Amman Temple" ##> Kattalai Amman Temple Kattalai Amman Temple is a Hindu Temple situated in Kanyakumari District at the southern part of Tamil Nadu, India. The temple is located in the village, surrounded by mountains and canals...
</information>
<observation> Kattalai is in the state of Tamil Nadu, India.</observation>
<sub_answer>Tamil Nadu, India</sub_answer>
------------------------------------------------------------
Based on the analysis, the state in India where Kattalai, a city, is located is <answer> Tamil Nadu, India </answer>

Figure 18: Esearch can efficiently handle a 4-hops reasoning question: after gathering relevant information across multiple hops, the model completes the reasoning process in just three search queries.

