# OpenReview forum: "Erase to Improve: Erasable Reinforcement Learning for Search-Augmented LLMs"
_ICLR.cc/2026/Conference — ICLR 2026 Poster_

### Official Review · Reviewer_gvRH · 2025-10-27

**Soundness:** 3
**Presentation:** 3
**Contribution:** 3
**Rating:** 6
**Confidence:** 3

**Summary:**

This paper proposes Erasable Reinforcement Learning (ERL), a new framework for enhancing the robustness of search-augmented LLMs in multi-hop reasoning tasks. The key idea is to explicitly identify and erase faulty reasoning or retrieval steps within a reasoning trajectory, then regenerate subsequent steps from the last correct state. The method introduces three erasure mechanisms (plan, search, and sub-answer erasure) triggered by reward thresholds on intermediate steps. ERL is trained using Qwen2.5-3B/7B models, evaluated on different benchmarks. Experiments show consistent SOTA gains and improved stability over PPO/GRPO. Ablation studies confirm the complementary roles of the three erasure modules, with sub-answer erasure contributing most.

**Strengths:**

1. The “erasable” mechanism introduces a fresh perspective distinct from prior step-wise or reward-shaping RL methods. It explicitly operationalizes fault detection and local correction, which is both intuitively appealing and empirically validated.

2. The multi-component reward (for search, sub-answer, and final answer) effectively mitigates the sparsity of typical RL signals, and the formulation is clear and well-motivated.

3. The component-wise ablation and comparison with PPO/GRPO support the claim that ERL provides stable optimization and interpretable corrective behavior.

**Weaknesses:**

1. The iterative erase-and-regenerate cycle introduces non-trivial computational overhead, which the paper acknowledges but does not quantify.

2. The erasure triggers and dense-reward coefficients appear manually tuned; there’s no sensitivity analysis showing robustness to these hyperparameters.

3. The evaluation is limited to search QA-style benchmarks. It remains unclear whether ERL generalizes to other agentic reasoning tasks (e.g., math, tool use, code reasoning).

4. More clearer illustrative examples or reasoning traces would make the method more intuitive to follow.

**Questions:**

1. What is the training cost relative to PPO/GRPO baselines (e.g., GPU hours given the same steps)?

2. How sensitive is ERL to the choice of thresholds hyperparameters?

3. Have you tried applying ERL to other reasoning-centric tasks, such as math or coding?

4. Are there qualitative/quantitive analysis or visualizations showing where erasure occurs in the reasoning chain, and how it affects the final outcome?

---

> ### Author Response · Authors · 2025-11-25
> **Response to Reviewer gvRH (Part 1)**
>
> Dear Reviewer,
>
> Thank you for taking the time to provide a thorough, patient, and professional review of our work. Your feedback not only affirmed the strengths of our approach but also precisely identified areas for further improvement. In response to your comments, we have provided point-by-point clarifications below. We hope that the supplementary experiments and detailed explanations will offer you a clearer understanding of the innovation, practicality, and performance of our method in practical applications.
>
> > What is the training cost relative to PPO/GRPO baselines (e.g., GPU hours given the same steps)?
>
> To evaluate the additional overhead and training efficiency introduced by ERL, we conducted supplementary experiments by systematically varying the maximum number of retries for the three erasure strategies (sub-answer, search, and planning). Since the erasure mechanism is implemented sequentially, the actual runtime measured on the same hardware directly reflects GPU computational load and external query costs. We recorded the following metrics:
>
> - Per-step wall-clock time (s/step) — Directly reflects computational and query costs.
>
> - Accuracy-per-step curve — Tracks training progress relative to optimization steps.
>
> - Accuracy-per-time curve — Measures training progress normalized by cumulative runtime, indicating practical training efficiency.
>
> The experimental results reveal significant differences in the effects of applying erasure across different modules, primarily reflecting the varying importance of each module in the chain-of-thought reasoning process. Applying the erasure mechanism to the Sub-Answer module, which has the greatest impact on the final answer, yielded the most substantial improvement in training efficiency. Although the computational time per step increased due to repeated reasoning and regeneration, the rate of improvement in training metrics per unit of computation time became significantly higher compared to the original PPO.
>
> Another noteworthy finding is that while GRPO demonstrated considerably higher efficiency per unit time than both the original PPO and PPO enhanced with erasure mechanisms, it exhibited a clearly lower performance ceiling. For more detailed experiments and analysis, please refer to Appendix A of the latest version of the paper.
>
> |Step|1|16|31|46|61|76|91|106|121|136|151|166|181|196|211|226|241|256|271|286|301|316|
> |-|-|-|-|-|-|-|-|-|-|-|-|-|-|-|-|-|-|-|-|-|-|-|
> |ppo|183|187|191|196|193|188|177|182|181|181|183|181|194|186|188|180|189|176|181|176|178|178|
> |grpo|176|178|162|158|141|126|128|129|135|127|135|152|158|155|144|144|140|156|172|181|203|209|
> |ε-sub_answer1|240|245|245|235|218|219|203|216|218|218|205|216|218|206|218|206|230|227|242|261|255|257|
> |ε-sub_answer3|320|317|303|311|327|311|312|310|326|297|299|269|283|293|282|280|279|260|279|325|364|362|
> |ε-sub_answer5|400|374|388|400|412|414|419|377|427|406|402|359|365|328|315|330|384|339|372|380|399|371|
>
> *Table 1: Time(s) per step. Per-step wall-clock time for for PPO, GRPO and Sub-Answer Erase at different budgets.*
>
>
>
> |Step|1|16|31|46|61|76|91|106|121|136|151|166|181|196|211|226|241|256|271|286|301|316|
> |-|-|-|-|-|-|-|-|-|-|-|-|-|-|-|-|-|-|-|-|-|-|-|
> |ppo|5.18|5.51|5.84|6.25|6.82|7.75|9.26|11.69|14.88|18.23|20.97|23.04|24.78|26.39|27.73|28.65|29.42|30.16|30.97|31.82|32.82|33.83|
> |grpo|3.87|5.97|8.43|12.05|17.59|22.80|26.26|27.82|28.79|29.68|30.38|31.05|31.66|32.28|32.94|33.67|34.30|34.82|34.99|35.07|35.08|35.07|
> |ε-sub_answer1|4.84|6.01|7.35|9.36|12.93|17.46|21.94|25.44|27.77|29.34|30.32|31.00|31.58|32.08|32.68|33.36|34.15|35.10|35.96|36.77|37.55|38.31|
> |ε-sub_answer3|4.46|6.06|7.94|10.76|15.60|21.25|25.86|29.03|31.10|32.76|34.03|34.76|35.37|36.09|36.96|37.57|37.91|38.20|38.68|39.76|41.12|42.56|
> |ε-sub_answer5|4.43|6.23|8.32|11.42|16.69|22.68|27.54|30.48|32.26|33.75|35.11|36.32|37.32|38.32|39.56|40.86|41.95|42.74|43.52|44.46|45.44|46.41|
>
> *Table 2: Correct(F1 %) by Step.*
>
>
>
> |Time(s)|200|5090|9980|14870|19760|24650|29540|34430|39320|44210|49100|53990|58880|63770|68660|73550|78440|83330|88220|93110|98000|
> |-|-|-|-|-|-|-|-|-|-|-|-|-|-|-|-|-|-|-|-|-|-|
> |ppo|5.19|5.75|6.49|8.07|12.04|18.09|22.57|25.64|28.03|29.48|30.88|32.53|34.03|-|-|-|-|-|-|-|-|
> |grpo|0.53|8.65|16.83|25.01|29.32|31.19|32.66|33.92|34.62|35.04|35.54|36.21|36.86|37.42|37.83|38.08|38.24|38.41|38.64|38.89|39.12|
> |ε-sub_answer1|4.82|6.38|8.46|13.17|19.87|25.47|28.71|30.38|31.38|32.18|33.18|34.33|35.62|36.70|37.69|38.46|-|-|-|-|-|
> |ε-sub_answer3|4.32|5.93|7.81|10.61|15.47|21.14|25.78|29.02|31.15|32.92|34.21|34.98|35.71|36.65|37.46|37.90|38.25|38.85|40.11|41.51|43.00|
> |ε-sub_answer5|5.25|6.12|7.10|8.65|11.92|16.68|21.97|26.33|29.29|31.03|32.27|33.45|34.67|35.77|36.80|37.68|38.61|39.84|41.18|42.07|42.87|
>
> *Table 3: Correct(F1 %) by Accumulated Time(s).*

---

> ### Author Response · Authors · 2025-11-25
> **Response to Reviewer gvRH (Part 2)**
>
> |Step|1|16|31|46|61|76|91|106|121|136|151|166|181|196|211|226|241|256|271|286|301|316|
> |-|-|-|-|-|-|-|-|-|-|-|-|-|-|-|-|-|-|-|-|-|-|-|
> |ppo|183|187|191|196|193|188|177|182|181|181|183|181|194|186|188|180|189|176|181|176|178|178|
> |grpo|176|178|162|158|141|126|128|129|135|127|135|152|158|155|144|144|140|156|172|181|203|209|
> |ε-search1|264|264|260|228|201|217|213|251|250|260|292|281|297|273|248|260|264|251|240|236|219|232|
> |ε-search3|358|383|359|355|275|282|275|287|295|292|317|331|308|262|301|278|297|293|304|366|426|398|
> |ε-search5|372|384|380|476|591|526|391|388|373|341|351|308|285|270|275|308|374|356|371|526|535|541|
>
> *Table 4: Time(s) per step. Per-step wall-clock time for PPO, GRPO and Search Erase at different budgets.*
>
>
>
> |Step|1|16|31|46|61|76|91|106|121|136|151|166|181|196|211|226|241|256|271|286|301|316|
> |-|-|-|-|-|-|-|-|-|-|-|-|-|-|-|-|-|-|-|-|-|-|-|
> |ppo|5.18|5.51|5.84|6.25|6.82|7.75|9.26|11.69|14.88|18.23|20.97|23.04|24.78|26.39|27.73|28.65|29.42|30.16|30.97|31.82|32.82|33.83|
> |grpo|3.87|5.97|8.43|12.05|17.59|22.80|26.26|27.82|28.79|29.68|30.38|31.05|31.66|32.28|32.94|33.67|34.30|34.82|34.99|35.07|35.08|35.07|
> |ε-search1|5.64|6.07|6.53|7.22|8.83|11.57|15.16|18.79|21.57|23.44|24.67|25.83|27.05|28.26|29.40|30.37|31.35|32.26|33.14|33.79|34.41|35.02|
> |ε-search3|5.62|6.23|6.97|8.21|10.66|13.90|17.59|20.84|23.43|25.31|26.58|27.57|28.42|29.28|30.15|31.04|31.88|32.79|33.76|34.79|35.81|36.81|
> |ε-search5|5.00|6.35|7.81|9.62|12.20|15.32|18.73|21.78|24.29|26.11|27.63|28.92|30.01|31.05|31.91|32.79|33.59|34.43|35.11|35.90|36.84|37.82|
>
> *Table 5: Correct(F1 %) by Step.*
>
>
>
> |Time(s)|200|5090|9980|14870|19760|24650|29540|34430|39320|44210|49100|53990|58880|63770|68660|73550|78440|83330|88220|93110|98000|
> |-|-|-|-|-|-|-|-|-|-|-|-|-|-|-|-|-|-|-|-|-|-|
> |ppo|5.19|5.75|6.49|8.07|12.04|18.09|22.57|25.64|28.03|29.48|30.88|32.53|34.03|-|-|-|-|-|-|-|-|
> |grpo|0.53|8.65|16.83|25.01|29.32|31.19|32.66|33.92|34.62|35.04|35.54|36.21|36.86|37.42|37.83|38.08|38.24|38.41|38.64|38.89|39.12|
> |ε-search1|5.65|6.17|6.78|8.69|13.00|18.25|21.90|23.94|25.26|26.63|28.01|29.39|30.59|31.82|32.98|33.92|34.78|35.12|-|-|-|
> |ε-search3|5.78|6.18|6.68|7.52|9.61|13.01|17.38|21.17|23.99|25.83|27.07|28.04|28.99|29.97|30.98|31.89|32.85|33.87|34.94|36.10|37.17|
> |ε-search5|5.29|6.18|7.13|8.21|9.73|11.43|13.26|15.28|17.62|20.16|22.67|24.66|26.24|27.71|29.00|30.14|31.33|32.23|33.11|34.05|34.99|
>
> *Table 6: Correct(F1 %) by Accumulated Time(s).*

---

> > ### Author Response · Authors · 2025-11-25
> > **Response to Reviewer gvRH (Part 3)**
> >
> > |Step|1|16|31|46|61|76|91|106|121|136|151|166|181|196|211|226|241|256|271|286|301|316|
> > |-|-|-|-|-|-|-|-|-|-|-|-|-|-|-|-|-|-|-|-|-|-|-|
> > |ppo|183|187|191|196|193|188|177|182|181|181|183|181|194|186|188|180|189|176|181|176|178|178|
> > |grpo|176|178|162|158|141|126|128|129|135|127|135|152|158|155|144|144|140|156|172|181|203|209|
> > |ε-plan1|212|221|218|221|223|211|212|197|206|211|208|207|215|216|219|215|221|228|234|240|244|242|
> > |ε-plan3|278|276|291|281|255|267|287|274|272|265|269|266|262|265|284|276|278|271|278|344|297|301|
> > |ε-plan5|370|353|352|354|344|341|394|356|335|314|370|327|321|337|311|327|350|337|323|329|339|344|
> >
> > *Table 7: Time(s) per step.Per-step wall-clock time for PPO, GRPO and Plan Erase at different budgets.*
> >
> >
> >
> > |Step|1|16|31|46|61|76|91|106|121|136|151|166|181|196|211|226|241|256|271|286|301|316|
> > |-|-|-|-|-|-|-|-|-|-|-|-|-|-|-|-|-|-|-|-|-|-|-|
> > |ppo|5.18|5.51|5.84|6.25|6.82|7.75|9.26|11.69|14.88|18.23|20.97|23.04|24.78|26.39|27.73|28.65|29.42|30.16|30.97|31.82|32.82|33.83|
> > |grpo|3.87|5.97|8.43|12.05|17.59|22.80|26.26|27.82|28.79|29.68|30.38|31.05|31.66|32.28|32.94|33.67|34.30|34.82|34.99|35.07|35.08|35.07|
> > |ε-plan1|4.90|5.36|5.87|6.56|8.11|10.61|13.82|17.02|19.91|22.41|24.56|26.27|27.56|28.55|29.35|30.07|30.77|31.41|32.03|32.57|33.19|33.82|
> > |ε-plan3|5.24|5.93|6.67|7.72|9.73|12.73|16.34|19.58|22.11|23.92|25.44|26.92|28.38|29.52|30.24|30.83|31.42|32.13|32.73|33.27|33.81|34.34|
> > |ε-plan5|4.56|5.91|7.36|9.22|12.00|15.24|18.58|21.43|23.72|25.54|27.05|28.35|29.27|30.01|30.77|31.63|32.30|32.72|33.14|33.81|34.60|35.42|
> >
> > *Table 8: Correct(F1 %) by Step.*
> >
> >
> >
> > |Time(s)|200|5090|9980|14870|19760|24650|29540|34430|39320|44210|49100|53990|58880|63770|68660|73550|78440|83330|88220|93110|98000|
> > |-|-|-|-|-|-|-|-|-|-|-|-|-|-|-|-|-|-|-|-|-|-|
> > |ppo|5.19|5.75|6.49|8.07|12.04|18.09|22.57|25.64|28.03|29.48|30.88|32.53|34.03|-|-|-|-|-|-|-|-|
> > |grpo|0.53|8.65|16.83|25.01|29.32|31.19|32.66|33.92|34.62|35.04|35.54|36.21|36.86|37.42|37.83|38.08|38.24|38.41|38.64|38.89|39.12|
> > |ε-plan1|4.90|5.61|6.55|9.22|13.86|18.78|22.83|25.88|27.86|29.17|30.24|31.22|32.09|32.88|33.74|33.96|-|-|-|-|-|
> > |ε-plan3|5.24|6.02|6.88|8.47|11.44|15.59|19.56|22.55|24.60|26.42|28.20|29.58|30.42|31.11|31.89|32.60|33.19|33.76|34.33|34.45|-|
> > |ε-plan5|4.76|5.80|6.91|8.38|10.73|13.68|16.81|19.78|22.28|24.38|26.00|27.48|28.68|29.47|30.13|30.92|31.79|32.40|32.73|33.07|33.39|
> >
> > *Table 9: Correct(F1 %) by Accumulated Time(s).*

---

> > > ### Author Response · Authors · 2025-11-25
> > > **Response to Reviewer gvRH (Part 4)**
> > >
> > > > How sensitive is ERL to the choice of thresholds hyperparameters?
> > >
> > > We fully acknowledge the critical importance of threshold selection in the ERL algorithm, particularly for triggering the erasure mechanism. To evaluate ERL's sensitivity to these thresholds (α, β), we conducted multiple sets of experiments to analyze their impact on model performance under different configurations.
> > >
> > > By adjusting α and β independently, we observed changes in model performance and erasure behavior. Our findings indicate that while ERL does exhibit sensitivity to these parameters, such sensitivity is bounded and does not lead to significant performance fluctuations. In other words, while appropriate threshold selection contributes notably to performance improvement, ERL remains capable of performing stable and effective erasure operations even with adjustments within a reasonable range.
> > >
> > > Analysis of results under varying α and β values revealed that performance variations primarily occur under extreme settings (e.g., excessively high or low thresholds). In such cases, the model may either over-erase useful information or under-erase invalid content, leading to performance degradation. Therefore, we recommend using a set of reasonable default values (α, β) and suggest fine-tuning these thresholds within a narrow range for optimal performance in specific applications.
> > >
> > > In summary, although ERL demonstrates some sensitivity to threshold settings, adjustments within a reasonable range do not cause severe negative impacts, and our algorithm maintains strong robustness to these parameters. For more detailed experiments and analysis, please refer to Appendix C of the latest version of the paper.
> > >
> > >
> > > |Step|1|16|31|46|61|76|91|106|121|136|151|166|181|196|211|226|241|256|271|286|301|316|
> > > |-|-|-|-|-|-|-|-|-|-|-|-|-|-|-|-|-|-|-|-|-|-|-|
> > > |β-0.1|6.00|7.25|5.89|6.32|8.74|13.37|14.81|19.76|21.72|21.78|23.90|28.19|28.00|28.94|29.34|30.40|32.87|33.05|30.42|33.01|34.38|35.10|
> > > |β-0.2|6.13|5.56|7.34|7.06|8.03|10.47|13.15|17.07|21.54|24.11|26.83|24.42|30.63|30.22|29.77|32.11|32.18|33.37|31.89|33.16|32.07|35.36|
> > > |β-0.3|5.84|5.71|7.88|7.06|7.22|12.00|18.18|19.48|22.24|27.23|25.19|28.38|29.17|29.44|29.83|31.64|31.34|32.80|32.98|33.36|33.06|35.35|
> > >
> > > *Table 10: Answer Correct Score (F1) by Step. The thresholds β to trigger the Plan erasure are compared between {0.1, 0.2, 0.3}.*
> > >
> > > |Step|1|16|31|46|61|76|91|106|121|136|151|166|181|196|211|226|241|256|271|286|301|316|
> > > |-|-|-|-|-|-|-|-|-|-|-|-|-|-|-|-|-|-|-|-|-|-|-|
> > > |α-0.05|6.00|5.11|7.77|7.73|7.69|12.63|17.82|20.65|22.93|23.97|27.68|26.83|27.83|29.99|27.35|30.45|30.56|30.28|32.61|34.22|32.56|35.28|
> > > |α-0.1|6.04|6.11|6.21|7.94|11.08|12.44|16.69|22.88|25.75|25.82|26.35|27.72|27.28|30.76|29.86|30.60|33.13|32.11|31.58|34.38|36.43|36.33|
> > > |α-0.15|5.90|5.23|6.42|7.94|7.81|9.76|11.98|13.84|15.18|15.75|16.03|16.06|16.43|16.92|15.68|14.94|14.18|12.95|14.02|14.51|14.57|14.92|
> > > |α-0.2|5.94|5.63|7.04|6.33|6.09|7.43|6.99|7.53|6.44|5.14|4.58|3.54|2.97|2.80|2.74|3.62|5.47|4.68|5.22|5.24|5.32|6.10|
> > >
> > > *Table 11: Answer Correct Score (F1) by Step. The thresholds α to trigger the Search erasure are compared between {0.05, 0.10, 0.15, 0.20}.*
> > >
> > >
> > > |Step|1|16|31|46|61|76|91|106|121|136|151|166|181|196|211|226|241|256|271|286|301|316|
> > > |-|-|-|-|-|-|-|-|-|-|-|-|-|-|-|-|-|-|-|-|-|-|-|
> > > |α-0.1|5.52|5.76|7.23|9.26|16.35|25.17|27.99|27.75|33.54|35.34|35.39|35.46|34.78|34.90|38.44|37.09|37.52|39.02|39.03|39.02|42.19|43.43|
> > > |α-0.2|5.19|6.06|6.97|8.15|13.53|22.59|22.99|28.69|30.16|30.94|33.31|34.30|36.78|34.81|35.09|36.93|39.52|40.18|39.74|40.55|37.82|38.98|
> > > |α-0.3|5.65|4.73|7.26|8.45|9.94|18.91|24.11|26.38|29.14|29.26|31.69|30.89|31.21|31.27|34.77|35.11|33.90|35.78|35.71|37.08|39.25|38.26|
> > >
> > > *Table 12: Answer Correct Score (F1) by Step. The thresholds α to trigger the Sub-Answer erasure are compared between {0.1, 0.2, 0.3}.*

---

> > > > ### Author Response · Authors · 2025-11-25
> > > > **Response to Reviewer gvRH (Part 5)**
> > > >
> > > > > Have you tried applying ERL to other reasoning-centric tasks, such as math or coding?
> > > >
> > > > Thank you for raising this important and forward-looking question. We fully agree that a valuable RL framework should not be limited to QA but must be extendable to more general long-form reasoning scenarios.
> > > >
> > > > The core challenge in long-form reasoning—such as mathematical proofs, complex code generation, and multi-step arguments—lies in the propagation and amplification of errors throughout the reasoning chain. Traditional RL methods struggle with long-sequence generation: once an error occurs at an intermediate step, the reward signals for subsequent steps become sparse and noisy, leading to inefficient learning.
> > > >
> > > > The ERL mechanism is inherently suited to address this challenge. In long-form reasoning, the model can learn not only to generate tokens but also to identify and erase "weak links" or incorrect steps in the reasoning chain that may cause subsequent errors. While in QA tasks, erasure actions operate on individual factual segments, in long-form reasoning, the action space may need to be expanded to larger textual units—such as a reasoning step, a code block, or a sentence. The reward function must also be refined to assess the intermediate correctness of the reasoning process, rather than relying solely on the final answer.
> > > >
> > > > It is worth emphasizing that training models to perform spontaneous erasure in general tasks without procedural supervision represents a more universal, promising, and valuable research direction. We look forward to presenting our subsequent work in this area.
> > > >
> > > >
> > > > > Are there qualitative/quantitive analysis or visualizations showing where erasure occurs in the reasoning chain, and how it affects the final outcome?
> > > >
> > > > To further analyze the occurrence and impact of erasure within reasoning chains, we conducted additional experiments to examine the timing and frequency of erasure events in multi-hop reasoning trajectories. Specifically, for each training step, we recorded the average number of retries triggered by the sub-answer, search, and planning modules at different maximum reasoning depths.
> > > >
> > > > Our findings reveal that the average frequency of sub-answer erasures increases notably as reasoning depth grows. This indicates that later stages of the reasoning chain face greater difficulty in producing correct sub-answers, making erasure and retries more necessary. The increased retries for more challenging multi-hop problems further demonstrate that error accumulation in earlier steps complicates subsequent reasoning, underscoring the value of ERL's mechanism.
> > > >
> > > > A similar trend was observed in the rewriting rounds (rounds 2, 3, and 4) of the search process. In contrast, the planning module, which employs a unique erasure mechanism active only in the first round, exhibited stable erasure behavior. For more detailed experiments and analysis, please refer to Appendix B of the latest version of the paper.
> > > >
> > > > |Step|1|16|31|46|61|76|91|106|121|136|151|166|181|196|211|226|241|256|271|286|301|316|
> > > > |-|-|-|-|-|-|-|-|-|-|-|-|-|-|-|-|-|-|-|-|-|-|-|
> > > > |round-1|1.00|0.93|1.14|1.11|1.06|1.00|0.50|1.17|1.06|0.95|0.97|1.00|0.00|0.95|0.42|1.00|1.55|1.00|0.97|1.50|0.98|1.44|
> > > > |sub_answer-round-2|1.72|1.77|1.85|1.88|1.82|1.52|1.32|1.21|1.35|1.06|1.27|1.12|1.12|1.12|1.08|1.15|1.30|1.27|1.59|1.85|2.14|1.95|
> > > > |sub_answer-round-3|2.20|2.02|2.24|2.37|2.12|1.92|1.88|1.88|1.97|1.75|1.74|1.67|1.76|1.72|1.51|1.55|1.62|1.55|1.71|1.86|2.32|2.39|
> > > > |sub_answer-round-4|2.34|2.07|2.29|2.57|2.52|2.21|2.33|2.51|2.49|2.25|1.93|2.00|2.18|2.21|2.20|2.26|2.27|1.76|2.13|2.00|2.08|2.88|
> > > >
> > > > *Table 13: SubAnswer Retried times by Step.*
> > > >
> > > >
> > > > |Step|1|16|31|46|61|76|91|106|121|136|151|166|181|196|211|226|241|256|271|286|301|316|
> > > > |-|-|-|-|-|-|-|-|-|-|-|-|-|-|-|-|-|-|-|-|-|-|-|
> > > > |plan-round-1|2.53|2.47|2.60|2.69|2.65|2.74|2.70|2.67|2.67|2.69|2.71|2.71|2.70|2.65|2.66|2.71|2.69|2.69|2.62|2.64|2.70|2.71|
> > > > |search-round-2|1.61|1.80|1.69|1.76|1.00|0.90|0.85|0.88|0.83|0.82|0.86|0.81|0.76|0.78|0.77|0.76|0.79|0.77|0.83|0.82|0.93|0.87|
> > > > |search-round-3|2.28|2.41|2.38|2.32|2.06|1.40|1.14|1.44|1.15|1.33|1.00|0.95|0.99|0.97|1.01|1.00|0.96|1.01|1.06|1.16|1.11|1.18|
> > > > |search-round-4|2.33|2.31|2.34|2.36|2.44|3.00|2.00|2.33|1.00|1.00|1.00|1.00|2.00|0.67|1.83|1.89|0.99|1.17|1.00|1.80|1.61|1.85|
> > > >
> > > > *Table 14: Search&Plan Retried times by Step.*
> > > >
> > > > We sincerely appreciate your thoughtful, precise, and highly constructive feedback. Your comments have not only helped strengthen our paper but also significantly expanded the depth and scope of our research. We are truly grateful for every point you raised—each has contributed to meaningful improvements in the work.
> > > >
> > > > It is our genuine hope that the newly added experiments and analyses have fully addressed all of your concerns. Once again, thank you for the time, effort, and patience you have dedicated to reviewing our paper.

---

> > > > > ### Author Response · Authors · 2025-11-27
> > > > > **Ensuring Clarity: Follow-up on Our Rebuttal Responses**
> > > > >
> > > > > Dear Reviewer,
> > > > >
> > > > > Thank you once more for the valuable feedback you provided during the review process, as well as for giving us the opportunity to clarify and supplement our work through the rebuttal.
> > > > >
> > > > > In our responses above, we have conducted detailed additional experiments and explanations to address your concerns. We sincerely hope that our supplementary experiments and analyses have fully resolved the questions you raised.
> > > > >
> > > > > Therefore, we would like to take the liberty to follow up and ask whether you have any further questions regarding our previous responses or if there are any specific aspects for which you would like us to provide additional clarification. We are more than willing and fully prepared to offer any further information you may require.
> > > > >
> > > > > This research work represents a crucial part of our long-term exploration, and we greatly value this opportunity to exchange ideas at ICLR. We also hold your professional judgment in the highest regard. Any further feedback from you would be immensely important to us.
> > > > >
> > > > > Once again, thank you for the time and effort you have dedicated to this process!

---

### Official Review · Reviewer_BrTp · 2025-10-30

**Soundness:** 3
**Presentation:** 3
**Contribution:** 2
**Rating:** 4
**Confidence:** 3

**Summary:**

This paper proposes Erasable Reinforcement Learning (ERL), a novel framework designed to enhance the robustness of multi-hop reasoning in search-augmented LLMs. The key insight is that current search-augmented LLMs suffer from decomposition, retrieval, and reasoning errors, where a single failure can derail the entire reasoning chain. ERL addresses this by introducing an erasure mechanism that detects faulty reasoning steps, removes them, and regenerates new reasoning from the last correct state.

**Strengths:**

1. The idea of erasable reasoning is inspired by human-like self-correction, effectively addressing a key weakness in multi-hop reasoning systems.

2. ERL demonstrates consistent and notable improvements over strong baselines across multiple datasets.

3. The paper provides thorough ablation studies (plan/search/sub-answer erasure), comparisons with PPO and GRPO, and both offline and online evaluations.

4. The writing is clear and mathematically rigorous, with precise formulations of the MDP setup, reward decomposition, and erasure operators.

**Weaknesses:**

While novel in its integration, ERL primarily extends existing reinforcement learning frameworks with a modular erasure component. The conceptual depth may be viewed as incremental rather than fundamentally new.

**Questions:**

1. How does ERL’s computational cost compare to PPO or GRPO during training?
2. Can ERL generalize beyond QA tasks, for example, to long-form reasoning or tool-using agents? If so, what modifications would be required?
3. How sensitive is ERL to the thresholds ($\alpha$, $\beta$) used to trigger erasures?
4. How does ERL scale with reasoning depth (e.g., beyond five hops)?
5. In real-world settings without gold evidence, how could dense rewards such as $R_{search}$ and $R_{subanswer}$ be approximated?

---

> ### Author Response · Authors · 2025-11-25
> **Response to Reviewer BrTp (Part 1)**
>
> Dear Reviewer,
>
> We sincerely thank you for your thorough review of our manuscript amidst your busy schedule. Your feedback is highly constructive and has been crucial in helping us refine the paper and improve the overall quality of our work. We feel honored to have received your evaluation and deeply appreciate your positive recognition of our research.
>
> We have carefully studied all the questions you raised and conducted extensive supplementary experiments along with detailed analyses accordingly. We hope that our responses will further clarify and highlight the value of our work. We also look forward to enhancing the quality of the manuscript through your valuable suggestions.
>
> Below are the modifications and additions we have made based on your feedback:
>
> > While novel in its integration, ERL primarily extends existing reinforcement learning frameworks with a modular erasure component. The conceptual depth may be viewed as incremental rather than fundamentally new.
>
> Thank you for your insightful feedback and for recognizing the novelty of our integrated approach. However, we would like to respectfully argue that ERL represents a paradigm-level innovation rather than an incremental extension, for the following reasons:
>
> **A fundamental shift from static to dynamic reasoning within the reinforcement learning architecture.**
>
> Current RL frameworks—particularly in reasoning tasks—are inherently brittle and sequential by nature. A single error in the reasoning chain propagates irreversibly, leading to catastrophic failure. This mirrors the limitations of early deep networks without residual connections, where gradient flow and error correction were severely constrained.
>
> ERL introduces the concept of "reasoning-layer erasure," enabling the agent to:
>
> - Dynamically detect and locate errors during the reasoning process, rather than in post-hoc analysis,
>
> - Excise flawed reasoning steps without discarding the entire reasoning chain,
>
> - Regenerate corrected reasoning in place while preserving valid context.
>
> This bears analogy to the introduction of residual connections in deep learning: while residual blocks were initially presented as modular components, they fundamentally redefined network architecture by enabling gradient flow across layers. Similarly, ERL’s erasure mechanism redefines how RL agents manage and recover from internal state errors—transitioning from a rigid sequential process to a dynamic, self-correcting one.
>
> **Going beyond "modular stacking": a new abstraction introduced for reasoning systems.**
>
> Although ERL is implemented modularly for ease of integration, its contribution is fundamentally conceptual. It introduces a novel abstraction: "reasoning is an editable process." This stands in sharp contrast to post-hoc reflection methods. ERL enables fine-grained, in-process editing of reasoning trajectories—a capability previously absent in RL. This is not merely an "extension" but a new primitive for building robust reasoning systems.
>
> Furthermore, training models to perform spontaneous erasure behavior on general tasks lacking procedural evidence presents a more universal, promising, and valuable research direction. We invite anticipation for our subsequent series of works in this area.
>
> **Real-Time Correction: Advantages in Efficiency and Scalability**
>
> ERL can be integrated with existing RL algorithms as a general-purpose mechanism to universally enhance robustness and sample efficiency.
>
> Critically, ERL operates in real time, unlike reflection-based methods that require complete regeneration of reasoning chains and incur significant latency. This makes it particularly suitable for:
>
> - Long-horizon tasks where error accumulation is a critical issue,
>
> - Resource-constrained environments where repeated full-chain reasoning is computationally expensive.
>
> **A New Perspective on Reasoning Resilience**
>
> ERL draws inspiration from human reasoning: we rarely discard an entire line of thought when making an error. We pause, erase, and rewrite. ERL is the first computational framework to embed this capability into RL, moving beyond the mainstream "generate-and-verify" paradigm toward a "generate, edit, and continue" approach.
>
> **Conclusion**
>
> We contend that ERL is not an incremental improvement, but a fundamental step toward editable and fault-tolerant reasoning in RL. Its implications may extend to any sequential decision-making system requiring robustness against intermediate errors.
>
> We hope this clarifies the conceptual depth and transformative potential of ERL.

---

> > ### Author Response · Authors · 2025-11-25
> > **Response to Reviewer BrTp (Part 2)**
> >
> > > How does ERL’s computational cost compare to PPO or GRPO during training?
> >
> > To evaluate the additional overhead and training efficiency introduced by ERL, we conducted supplementary experiments by systematically varying the maximum retry limits for the three erasure strategies (sub-answer, search, plan). Since the erasure mechanism is implemented sequentially, the actual runtime measured on identical hardware directly reflects GPU computational load and external query volume. We recorded the following metrics:
> >
> > - Wall-clock time per step (s/step) — Directly reflects the combined cost of computation and external queries.
> >
> > - Accuracy-per-step curve — Tracks training progress relative to the number of optimization steps.
> >
> > - Accuracy-per-time curve — Measures training progress normalized by cumulative runtime, representing practical training efficiency.
> >
> > Experimental results indicate significant variation in the effects of applying erasure across different reasoning stages, primarily reflecting the relative importance of each stage within the chain-of-thought process. Implementing the erasure mechanism for the Sub-Answer stage—which exerts the greatest influence on the final answer—yields the most substantial improvement in training efficiency. Although the computational time per step increases due to repeated reasoning, the rate of improvement in training metrics per unit of computation time becomes higher compared to the original PPO.
> >
> > Another noteworthy observation is that while GRPO demonstrates significantly higher efficiency per unit time than both the original PPO and PPO enhanced with erasure mechanisms, it exhibits a clearly lower performance ceiling. For more detailed experiments and analysis, please refer to Appendix A of the latest version of the paper.
> >
> > |Step|1|16|31|46|61|76|91|106|121|136|151|166|181|196|211|226|241|256|271|286|301|316|
> > |-|-|-|-|-|-|-|-|-|-|-|-|-|-|-|-|-|-|-|-|-|-|-|
> > |ppo|183|187|191|196|193|188|177|182|181|181|183|181|194|186|188|180|189|176|181|176|178|178|
> > |grpo|176|178|162|158|141|126|128|129|135|127|135|152|158|155|144|144|140|156|172|181|203|209|
> > |ε-sub_answer1|240|245|245|235|218|219|203|216|218|218|205|216|218|206|218|206|230|227|242|261|255|257|
> > |ε-sub_answer3|320|317|303|311|327|311|312|310|326|297|299|269|283|293|282|280|279|260|279|325|364|362|
> > |ε-sub_answer5|400|374|388|400|412|414|419|377|427|406|402|359|365|328|315|330|384|339|372|380|399|371|
> >
> > *Table 1: Time(s) per step. Per-step wall-clock time for for PPO, GRPO and Sub-Answer Erase at different budgets.*
> >
> >
> >
> > |Step|1|16|31|46|61|76|91|106|121|136|151|166|181|196|211|226|241|256|271|286|301|316|
> > |-|-|-|-|-|-|-|-|-|-|-|-|-|-|-|-|-|-|-|-|-|-|-|
> > |ppo|5.18|5.51|5.84|6.25|6.82|7.75|9.26|11.69|14.88|18.23|20.97|23.04|24.78|26.39|27.73|28.65|29.42|30.16|30.97|31.82|32.82|33.83|
> > |grpo|3.87|5.97|8.43|12.05|17.59|22.80|26.26|27.82|28.79|29.68|30.38|31.05|31.66|32.28|32.94|33.67|34.30|34.82|34.99|35.07|35.08|35.07|
> > |ε-sub_answer1|4.84|6.01|7.35|9.36|12.93|17.46|21.94|25.44|27.77|29.34|30.32|31.00|31.58|32.08|32.68|33.36|34.15|35.10|35.96|36.77|37.55|38.31|
> > |ε-sub_answer3|4.46|6.06|7.94|10.76|15.60|21.25|25.86|29.03|31.10|32.76|34.03|34.76|35.37|36.09|36.96|37.57|37.91|38.20|38.68|39.76|41.12|42.56|
> > |ε-sub_answer5|4.43|6.23|8.32|11.42|16.69|22.68|27.54|30.48|32.26|33.75|35.11|36.32|37.32|38.32|39.56|40.86|41.95|42.74|43.52|44.46|45.44|46.41|
> >
> > *Table 2: Correct(F1 %) by Step.*
> >
> >
> >
> > |Time(s)|200|5090|9980|14870|19760|24650|29540|34430|39320|44210|49100|53990|58880|63770|68660|73550|78440|83330|88220|93110|98000|
> > |-|-|-|-|-|-|-|-|-|-|-|-|-|-|-|-|-|-|-|-|-|-|
> > |ppo|5.19|5.75|6.49|8.07|12.04|18.09|22.57|25.64|28.03|29.48|30.88|32.53|34.03|-|-|-|-|-|-|-|-|
> > |grpo|0.53|8.65|16.83|25.01|29.32|31.19|32.66|33.92|34.62|35.04|35.54|36.21|36.86|37.42|37.83|38.08|38.24|38.41|38.64|38.89|39.12|
> > |ε-sub_answer1|4.82|6.38|8.46|13.17|19.87|25.47|28.71|30.38|31.38|32.18|33.18|34.33|35.62|36.70|37.69|38.46|-|-|-|-|-|
> > |ε-sub_answer3|4.32|5.93|7.81|10.61|15.47|21.14|25.78|29.02|31.15|32.92|34.21|34.98|35.71|36.65|37.46|37.90|38.25|38.85|40.11|41.51|43.00|
> > |ε-sub_answer5|5.25|6.12|7.10|8.65|11.92|16.68|21.97|26.33|29.29|31.03|32.27|33.45|34.67|35.77|36.80|37.68|38.61|39.84|41.18|42.07|42.87|
> >
> > *Table 3: Correct(F1 %) by Accumulated Time(s).*

---

> ### Author Response · Authors · 2025-11-25
> **Response to Reviewer BrTp (Part 3)**
>
> |Step|1|16|31|46|61|76|91|106|121|136|151|166|181|196|211|226|241|256|271|286|301|316|
> |-|-|-|-|-|-|-|-|-|-|-|-|-|-|-|-|-|-|-|-|-|-|-|
> |ppo|183|187|191|196|193|188|177|182|181|181|183|181|194|186|188|180|189|176|181|176|178|178|
> |grpo|176|178|162|158|141|126|128|129|135|127|135|152|158|155|144|144|140|156|172|181|203|209|
> |ε-search1|264|264|260|228|201|217|213|251|250|260|292|281|297|273|248|260|264|251|240|236|219|232|
> |ε-search3|358|383|359|355|275|282|275|287|295|292|317|331|308|262|301|278|297|293|304|366|426|398|
> |ε-search5|372|384|380|476|591|526|391|388|373|341|351|308|285|270|275|308|374|356|371|526|535|541|
>
> *Table 4: Time(s) per step. Per-step wall-clock time for PPO, GRPO and Search Erase at different budgets.*
>
>
>
> |Step|1|16|31|46|61|76|91|106|121|136|151|166|181|196|211|226|241|256|271|286|301|316|
> |-|-|-|-|-|-|-|-|-|-|-|-|-|-|-|-|-|-|-|-|-|-|-|
> |ppo|5.18|5.51|5.84|6.25|6.82|7.75|9.26|11.69|14.88|18.23|20.97|23.04|24.78|26.39|27.73|28.65|29.42|30.16|30.97|31.82|32.82|33.83|
> |grpo|3.87|5.97|8.43|12.05|17.59|22.80|26.26|27.82|28.79|29.68|30.38|31.05|31.66|32.28|32.94|33.67|34.30|34.82|34.99|35.07|35.08|35.07|
> |ε-search1|5.64|6.07|6.53|7.22|8.83|11.57|15.16|18.79|21.57|23.44|24.67|25.83|27.05|28.26|29.40|30.37|31.35|32.26|33.14|33.79|34.41|35.02|
> |ε-search3|5.62|6.23|6.97|8.21|10.66|13.90|17.59|20.84|23.43|25.31|26.58|27.57|28.42|29.28|30.15|31.04|31.88|32.79|33.76|34.79|35.81|36.81|
> |ε-search5|5.00|6.35|7.81|9.62|12.20|15.32|18.73|21.78|24.29|26.11|27.63|28.92|30.01|31.05|31.91|32.79|33.59|34.43|35.11|35.90|36.84|37.82|
>
> *Table 5: Correct(F1 %) by Step.*
>
>
>
> |Time(s)|200|5090|9980|14870|19760|24650|29540|34430|39320|44210|49100|53990|58880|63770|68660|73550|78440|83330|88220|93110|98000|
> |-|-|-|-|-|-|-|-|-|-|-|-|-|-|-|-|-|-|-|-|-|-|
> |ppo|5.19|5.75|6.49|8.07|12.04|18.09|22.57|25.64|28.03|29.48|30.88|32.53|34.03|-|-|-|-|-|-|-|-|
> |grpo|0.53|8.65|16.83|25.01|29.32|31.19|32.66|33.92|34.62|35.04|35.54|36.21|36.86|37.42|37.83|38.08|38.24|38.41|38.64|38.89|39.12|
> |ε-search1|5.65|6.17|6.78|8.69|13.00|18.25|21.90|23.94|25.26|26.63|28.01|29.39|30.59|31.82|32.98|33.92|34.78|35.12|-|-|-|
> |ε-search3|5.78|6.18|6.68|7.52|9.61|13.01|17.38|21.17|23.99|25.83|27.07|28.04|28.99|29.97|30.98|31.89|32.85|33.87|34.94|36.10|37.17|
> |ε-search5|5.29|6.18|7.13|8.21|9.73|11.43|13.26|15.28|17.62|20.16|22.67|24.66|26.24|27.71|29.00|30.14|31.33|32.23|33.11|34.05|34.99|
>
> *Table 6: Correct(F1 %) by Accumulated Time(s).*

---

> ### Author Response · Authors · 2025-11-25
> **Response to Reviewer BrTp (Part 4)**
>
> |Step|1|16|31|46|61|76|91|106|121|136|151|166|181|196|211|226|241|256|271|286|301|316|
> |-|-|-|-|-|-|-|-|-|-|-|-|-|-|-|-|-|-|-|-|-|-|-|
> |ppo|183|187|191|196|193|188|177|182|181|181|183|181|194|186|188|180|189|176|181|176|178|178|
> |grpo|176|178|162|158|141|126|128|129|135|127|135|152|158|155|144|144|140|156|172|181|203|209|
> |ε-plan1|212|221|218|221|223|211|212|197|206|211|208|207|215|216|219|215|221|228|234|240|244|242|
> |ε-plan3|278|276|291|281|255|267|287|274|272|265|269|266|262|265|284|276|278|271|278|344|297|301|
> |ε-plan5|370|353|352|354|344|341|394|356|335|314|370|327|321|337|311|327|350|337|323|329|339|344|
>
> *Table 7: Time(s) per step.Per-step wall-clock time for PPO, GRPO and Plan Erase at different budgets.*
>
>
>
> |Step|1|16|31|46|61|76|91|106|121|136|151|166|181|196|211|226|241|256|271|286|301|316|
> |-|-|-|-|-|-|-|-|-|-|-|-|-|-|-|-|-|-|-|-|-|-|-|
> |ppo|5.18|5.51|5.84|6.25|6.82|7.75|9.26|11.69|14.88|18.23|20.97|23.04|24.78|26.39|27.73|28.65|29.42|30.16|30.97|31.82|32.82|33.83|
> |grpo|3.87|5.97|8.43|12.05|17.59|22.80|26.26|27.82|28.79|29.68|30.38|31.05|31.66|32.28|32.94|33.67|34.30|34.82|34.99|35.07|35.08|35.07|
> |ε-plan1|4.90|5.36|5.87|6.56|8.11|10.61|13.82|17.02|19.91|22.41|24.56|26.27|27.56|28.55|29.35|30.07|30.77|31.41|32.03|32.57|33.19|33.82|
> |ε-plan3|5.24|5.93|6.67|7.72|9.73|12.73|16.34|19.58|22.11|23.92|25.44|26.92|28.38|29.52|30.24|30.83|31.42|32.13|32.73|33.27|33.81|34.34|
> |ε-plan5|4.56|5.91|7.36|9.22|12.00|15.24|18.58|21.43|23.72|25.54|27.05|28.35|29.27|30.01|30.77|31.63|32.30|32.72|33.14|33.81|34.60|35.42|
>
> *Table 8: Correct(F1 %) by Step.*
>
>
>
> |Time(s)|200|5090|9980|14870|19760|24650|29540|34430|39320|44210|49100|53990|58880|63770|68660|73550|78440|83330|88220|93110|98000|
> |-|-|-|-|-|-|-|-|-|-|-|-|-|-|-|-|-|-|-|-|-|-|
> |ppo|5.19|5.75|6.49|8.07|12.04|18.09|22.57|25.64|28.03|29.48|30.88|32.53|34.03|-|-|-|-|-|-|-|-|
> |grpo|0.53|8.65|16.83|25.01|29.32|31.19|32.66|33.92|34.62|35.04|35.54|36.21|36.86|37.42|37.83|38.08|38.24|38.41|38.64|38.89|39.12|
> |ε-plan1|4.90|5.61|6.55|9.22|13.86|18.78|22.83|25.88|27.86|29.17|30.24|31.22|32.09|32.88|33.74|33.96|-|-|-|-|-|
> |ε-plan3|5.24|6.02|6.88|8.47|11.44|15.59|19.56|22.55|24.60|26.42|28.20|29.58|30.42|31.11|31.89|32.60|33.19|33.76|34.33|34.45|-|
> |ε-plan5|4.76|5.80|6.91|8.38|10.73|13.68|16.81|19.78|22.28|24.38|26.00|27.48|28.68|29.47|30.13|30.92|31.79|32.40|32.73|33.07|33.39|
>
> *Table 9: Correct(F1 %) by Accumulated Time(s).*

---

> > ### Author Response · Authors · 2025-11-25
> > **Response to Reviewer BrTp (Part 5)**
> >
> > > Can ERL generalize beyond QA tasks, for example, to long-form reasoning or tool-using agents? If so, what modifications would be required?
> >
> > Thank you for raising this important and forward-looking question. We fully agree that a valuable RL framework should not be limited to QA but should be extendable to more general long-form reasoning and tool-using agent scenarios. Below, we will elaborate in detail on the applicability, potential advantages, and necessary modifications of the ERL framework in these two cutting-edge scenarios.
> >
> > **Extending to Long-form Reasoning**
> >
> > The core challenge in long-form reasoning—such as mathematical proofs, complex code generation, and multi-step arguments—lies in the propagation and amplification of errors throughout the reasoning chain. Traditional RL methods struggle with generating long sequences: once an error occurs at an intermediate step, the reward signals for all subsequent steps become sparse and noisy, leading to inefficient learning.
> >
> > The ERL mechanism is inherently well-suited to address this challenge. In long-form reasoning, the model can learn not only to generate tokens but also to identify and erase "weak links" or erroneous steps in the reasoning chain that may lead to subsequent mistakes. While in QA tasks, the erasure action operates on individual factual segments, in long-form reasoning, the action space may need to be expanded to larger textual units—such as a reasoning step, a code block, or a sentence. Additionally, the reward function must be refined to assess the intermediate correctness of the reasoning process, rather than solely relying on the final answer.
> >
> > **Extending to Tool-using Agents**
> >
> > In tool-using scenarios, an agent must plan and execute a sequence of actions, such as API calls, database queries, or code execution. Failures often stem from an incorrect action choice during the planning phase or a tool invocation error due to formatting mistakes during execution. ERL can serve as a powerful mechanism for planning reflection and correction. When a tool-using agent's execution trajectory fails—for instance, if a Python interpreter returns a syntax error or a search engine yields irrelevant results—ERL's eraser can be trained to roll back to the critical decision point in the trajectory that caused the failure.
> >
> > The state space must encompass the complete interaction history, including tool invocations, their returned results, and environmental feedback. The erasure action needs to be defined as revoking a specific tool call or modifying an erroneous query. Rewards will be linked to the outcomes of tool usage. The framework requires tighter integration with external tools and environments.
> >
> > **Summary**
> >
> > In conclusion, ERL serves as a versatile paradigm whose applicability extends far beyond open-domain QA. It offers a novel and promising solution to address error propagation in long-form reasoning and planning fragility in tool-using scenarios.

---

> > > ### Author Response · Authors · 2025-11-25
> > > **Response to Reviewer BrTp (Part 6)**
> > >
> > > > How sensitive is ERL to the thresholds (α，β) used to trigger erasures?
> > >
> > > We fully acknowledge that the selection of thresholds is crucial for the ERL algorithm, particularly in triggering the erasure mechanism. To evaluate ERL's sensitivity to these thresholds (α, β), we conducted multiple sets of experiments to analyze the impact of different threshold settings on model performance.
> > >
> > > By adjusting α and β independently, we observed how model performance and erasure behavior changed under different configurations. Our findings indicate that while ERL does exhibit sensitivity to these two parameters, such sensitivity is bounded and does not lead to significant performance fluctuations. In other words, appropriate threshold selection contributes notably to performance improvement, but even with moderate adjustments within a certain range, ERL consistently performs effective erasure operations to enhance outcomes.
> > >
> > > Analysis of the experimental results under varying α and β values revealed that performance variations primarily occurred under extreme settings (e.g., excessively high or low thresholds). In such cases, the model might either over-erase useful information or under-erase invalid information, leading to performance degradation. Therefore, we recommend using a set of reasonable default values (α, β) and suggest users fine-tune these thresholds within a narrow range for optimal performance in specific applications.
> > >
> > > In summary, although ERL exhibits some sensitivity to threshold settings, adjustments within a reasonable range do not cause severe negative impacts, and our algorithm demonstrates strong robustness to these parameters. For more detailed experiments and analysis, please refer to Appendix C of the latest version of the paper.
> > >
> > >
> > > |Step|1|16|31|46|61|76|91|106|121|136|151|166|181|196|211|226|241|256|271|286|301|316|
> > > |-|-|-|-|-|-|-|-|-|-|-|-|-|-|-|-|-|-|-|-|-|-|-|
> > > |β-0.1|6.00|7.25|5.89|6.32|8.74|13.37|14.81|19.76|21.72|21.78|23.90|28.19|28.00|28.94|29.34|30.40|32.87|33.05|30.42|33.01|34.38|35.10|
> > > |β-0.2|6.13|5.56|7.34|7.06|8.03|10.47|13.15|17.07|21.54|24.11|26.83|24.42|30.63|30.22|29.77|32.11|32.18|33.37|31.89|33.16|32.07|35.36|
> > > |β-0.3|5.84|5.71|7.88|7.06|7.22|12.00|18.18|19.48|22.24|27.23|25.19|28.38|29.17|29.44|29.83|31.64|31.34|32.80|32.98|33.36|33.06|35.35|
> > >
> > > *Table 10: Answer Correct Score (F1) by Step. The thresholds β to trigger the Plan erasure are compared between {0.1, 0.2, 0.3}.*
> > >
> > > |Step|1|16|31|46|61|76|91|106|121|136|151|166|181|196|211|226|241|256|271|286|301|316|
> > > |-|-|-|-|-|-|-|-|-|-|-|-|-|-|-|-|-|-|-|-|-|-|-|
> > > |α-0.05|6.00|5.11|7.77|7.73|7.69|12.63|17.82|20.65|22.93|23.97|27.68|26.83|27.83|29.99|27.35|30.45|30.56|30.28|32.61|34.22|32.56|35.28|
> > > |α-0.1|6.04|6.11|6.21|7.94|11.08|12.44|16.69|22.88|25.75|25.82|26.35|27.72|27.28|30.76|29.86|30.60|33.13|32.11|31.58|34.38|36.43|36.33|
> > > |α-0.15|5.90|5.23|6.42|7.94|7.81|9.76|11.98|13.84|15.18|15.75|16.03|16.06|16.43|16.92|15.68|14.94|14.18|12.95|14.02|14.51|14.57|14.92|
> > > |α-0.2|5.94|5.63|7.04|6.33|6.09|7.43|6.99|7.53|6.44|5.14|4.58|3.54|2.97|2.80|2.74|3.62|5.47|4.68|5.22|5.24|5.32|6.10|
> > >
> > > *Table 11: Answer Correct Score (F1) by Step. The thresholds α to trigger the Search erasure are compared between {0.05, 0.10, 0.15, 0.20}.*
> > >
> > >
> > > |Step|1|16|31|46|61|76|91|106|121|136|151|166|181|196|211|226|241|256|271|286|301|316|
> > > |-|-|-|-|-|-|-|-|-|-|-|-|-|-|-|-|-|-|-|-|-|-|-|
> > > |α-0.1|5.52|5.76|7.23|9.26|16.35|25.17|27.99|27.75|33.54|35.34|35.39|35.46|34.78|34.90|38.44|37.09|37.52|39.02|39.03|39.02|42.19|43.43|
> > > |α-0.2|5.19|6.06|6.97|8.15|13.53|22.59|22.99|28.69|30.16|30.94|33.31|34.30|36.78|34.81|35.09|36.93|39.52|40.18|39.74|40.55|37.82|38.98|
> > > |α-0.3|5.65|4.73|7.26|8.45|9.94|18.91|24.11|26.38|29.14|29.26|31.69|30.89|31.21|31.27|34.77|35.11|33.90|35.78|35.71|37.08|39.25|38.26|
> > >
> > > *Table 12: Answer Correct Score (F1) by Step. The thresholds α to trigger the Sub-Answer erasure are compared between {0.1, 0.2, 0.3}.*

---

> > > > ### Author Response · Authors · 2025-11-25
> > > > **Response to Reviewer BrTp (Part 7)**
> > > >
> > > > > How does ERL scale with reasoning depth (e.g., beyond five hops)?
> > > >
> > > > To analyze ERL's performance in deeper reasoning tasks, we conducted additional experiments to examine the timing and frequency of erasure events during multi-hop reasoning trajectories. Specifically, for each training step, we recorded the average number of retries triggered by the sub-answer, search, and planning modules at different maximum reasoning depths.
> > > >
> > > > Our findings indicate that the average frequency of sub-answer erasures increases noticeably as reasoning depth grows. This suggests that later stages of the reasoning chain face greater difficulty in generating correct sub-answers, making erasure and retries more necessary. The increased retries for more challenging multi-hop problems further demonstrate that error accumulation in earlier steps complicates subsequent reasoning, highlighting the value of ERL's mechanism.
> > > >
> > > > A similar trend was observed in the rewriting rounds (round 2, 3, 4) of the search process. In contrast, the planning module, which employs a unique erasure mechanism only in the first round, exhibited stable erasure behavior. For more detailed experiments and analysis, please refer to Appendix B of the latest version of the paper.
> > > >
> > > > |Step|1|16|31|46|61|76|91|106|121|136|151|166|181|196|211|226|241|256|271|286|301|316|
> > > > |-|-|-|-|-|-|-|-|-|-|-|-|-|-|-|-|-|-|-|-|-|-|-|
> > > > |round-1|1.00|0.93|1.14|1.11|1.06|1.00|0.50|1.17|1.06|0.95|0.97|1.00|0.00|0.95|0.42|1.00|1.55|1.00|0.97|1.50|0.98|1.44|
> > > > |sub_answer-round-2|1.72|1.77|1.85|1.88|1.82|1.52|1.32|1.21|1.35|1.06|1.27|1.12|1.12|1.12|1.08|1.15|1.30|1.27|1.59|1.85|2.14|1.95|
> > > > |sub_answer-round-3|2.20|2.02|2.24|2.37|2.12|1.92|1.88|1.88|1.97|1.75|1.74|1.67|1.76|1.72|1.51|1.55|1.62|1.55|1.71|1.86|2.32|2.39|
> > > > |sub_answer-round-4|2.34|2.07|2.29|2.57|2.52|2.21|2.33|2.51|2.49|2.25|1.93|2.00|2.18|2.21|2.20|2.26|2.27|1.76|2.13|2.00|2.08|2.88|
> > > >
> > > > *Table 13: SubAnswer Retried times by Step.*
> > > >
> > > >
> > > > |Step|1|16|31|46|61|76|91|106|121|136|151|166|181|196|211|226|241|256|271|286|301|316|
> > > > |-|-|-|-|-|-|-|-|-|-|-|-|-|-|-|-|-|-|-|-|-|-|-|
> > > > |plan-round-1|2.53|2.47|2.60|2.69|2.65|2.74|2.70|2.67|2.67|2.69|2.71|2.71|2.70|2.65|2.66|2.71|2.69|2.69|2.62|2.64|2.70|2.71|
> > > > |search-round-2|1.61|1.80|1.69|1.76|1.00|0.90|0.85|0.88|0.83|0.82|0.86|0.81|0.76|0.78|0.77|0.76|0.79|0.77|0.83|0.82|0.93|0.87|
> > > > |search-round-3|2.28|2.41|2.38|2.32|2.06|1.40|1.14|1.44|1.15|1.33|1.00|0.95|0.99|0.97|1.01|1.00|0.96|1.01|1.06|1.16|1.11|1.18|
> > > > |search-round-4|2.33|2.31|2.34|2.36|2.44|3.00|2.00|2.33|1.00|1.00|1.00|1.00|2.00|0.67|1.83|1.89|0.99|1.17|1.00|1.80|1.61|1.85|
> > > >
> > > > *Table 14: Search&Plan Retried times by Step.*
> > > >
> > > >
> > > > > In real-world settings without gold evidence, how could dense rewards such as $R_{search}$ and $R_{subanswer}$ be approximated?
> > > >
> > > > For more general tasks, shifting from "direct supervision" to "integrating verifiable weak signals" can reduce reliance on gold evidence. In the absence of gold evidence, verifiable weak signals—such as retrieval relevance, regeneration consistency, self-consistency variance, counterfactual gain, and causal contribution to answers—can be leveraged to approximate dense reward signals.
> > > >
> > > > These proxy signals are incorporated into policy learning via confidence-weighted updates and conservative adjustments. Once refuted by counterevidence, they can be "erased" to prevent noise propagation. Moreover, training models to perform spontaneous erasure in tasks lacking gold evidence represents a more general, promising, and valuable research direction. We invite attention to our subsequent work in this area.
> > > >
> > > > Thank you once again for all your valuable comments and suggestions. We have carefully reflected on them and made detailed additions to the paper. We sincerely hope that these improvements and efforts are evident to you and that we may receive your continued encouragement and recognition.
> > > >
> > > > We deeply appreciate your support for our research and look forward to your further evaluation. It is our earnest hope that our work will earn your approval.

---

> > > > > ### Author Response · Authors · 2025-11-27
> > > > > **Ensuring Clarity: Follow-up on Our Rebuttal Responses**
> > > > >
> > > > > Dear Reviewer,
> > > > >
> > > > > Thank you once more for the valuable feedback you provided during the review process, as well as for giving us the opportunity to clarify and supplement our work through the rebuttal.
> > > > >
> > > > > In our responses above, we have conducted detailed additional experiments and explanations to address your concerns. We sincerely hope that our supplementary experiments and analyses have fully resolved the questions you raised.
> > > > >
> > > > > Therefore, we would like to take the liberty to follow up and ask whether you have any further questions regarding our previous responses or if there are any specific aspects for which you would like us to provide additional clarification. We are more than willing and fully prepared to offer any further information you may require.
> > > > >
> > > > > This research work represents a crucial part of our long-term exploration, and we greatly value this opportunity to exchange ideas at ICLR. We also hold your professional judgment in the highest regard. Any further feedback from you would be immensely important to us.
> > > > >
> > > > > Once again, thank you for the time and effort you have dedicated to this process!

---

### Official Review · Reviewer_jujf · 2025-10-31

**Soundness:** 3
**Presentation:** 1
**Contribution:** 3
**Rating:** 6
**Confidence:** 2

**Summary:**

This paper proposes Erasable Reinforcement Learning (ERL) for search-augmented LLM agents: the agent detects faulty steps in multi-hop reasoning (decomposition, retrieval, or reasoning), reverts those segments, and regenerates them in-place with dense stepwise rewards, reducing error propagation. On HotpotQA, MuSiQue, 2Wiki, and Bamboogle, ERL-trained models beat strong baselines as new SOTA in both offline and online settings

**Strengths:**

* Novel training strategy: Introduces *Erasable RL* with three erasure signals—plan/init search, subsequent search, and sub-answer—to cut off faulty steps and retry, directly targeting error propagation.
* Dense, well-aligned rewards: Designs stepwise rewards for retrieval and intermediate reasoning, with token-level attribution, improving credit assignment beyond sparse final answers.
* Strong empirical results & diagnostics: The experiments shows the proposed approach achieves SOTA across four multi-hop QA datasets in both offline and online settings, with consistent gains at 3B/7B and ablations showing each erasure component is complementary.

**Weaknesses:**

- Unclear implementation details. Which dataset is used for training? And also, it is unclear on how the grounding gold evidence and gold sub-answers set is constructed. Furthermore, the dependence on gold set may be infessible for many real-world applications, unless a general and lightweight approach can be provided.
- Concerns on efficiency and scalability : The erase–regenerate loop adds computational overhead and may require repeated passes when multiple heterogeneous errors occur; training notes also show retrieval quality is bottlenecked by a local Wiki-18 setup, suggesting a practical bottleneck beyond the algorithm itself.  Also for real-world challenge setup, it is likely the search cannot provide answer, which means the model will struck in the loop, unless the training data and search resource ensure valid information can be provided.
- Writing issue, e.g. L160 Jin et al. (2025b) > (Jin et al., 2025b). These issue occurs very frequent spanning intro, sec 4.1, and related work.

**Questions:**

- I would expect longer rollout time and more research queries thus make it both cost more and takes a longer time for each unit update/iteration, but I am not sure about the overall effect.

---

> ### Author Response · Authors · 2025-11-25
> **Response to Reviewer jujf (Part 1)**
>
> Dear Reviewer,
>
> Thank you very much for taking the time to carefully review our paper and providing such detailed and constructive feedback. Your comments are highly valuable to us. We have thoroughly considered them and have supplemented the manuscript with additional experiments to better address your concerns.
>
> > Unclear implementation details. Which dataset is used for training? And also, it is unclear on how the grounding gold evidence and gold sub-answers set is constructed. Furthermore, the dependence on gold set may be infessible for many real-world applications, unless a general and lightweight approach can be provided.
>
> During training, we utilize the MuSiQue dataset. Each sample in this dataset contains: a main question, its decomposed sub-questions, corresponding sub-answers, the final answer, and supporting documents. For detailed information on the MuSiQue dataset construction process, please refer to the corresponding paper [1].
>
> Therefore, instead of manually re-annotating a new gold-standard dataset, we directly leverage the inherent structure of the MuSiQue dataset, which includes supporting paragraphs, question decomposition, and answers.
>
> For more general tasks, to reduce reliance on gold evidence, the paradigm can shift from direct supervision toward integrating verifiable weak signals. In the absence of gold evidence, dense rewards can be approximated using verifiable weak signals—such as retrieval relevance, regeneration consistency, self-consistency variance, counterfactual gain, and causal contribution to answers.
>
> These proxy signals are incorporated into policy learning through confidence weighting and conservative updating. Once refuted, they can be "erased" to prevent noise amplification.
>
> Furthermore, training models to perform spontaneous erasure without gold evidence represents a more general, promising, and valuable research direction. We look forward to presenting our follow-up work on this topic.
>
> [1] Musique: Multihop questions via single-hop question composition.

---

> ### Author Response · Authors · 2025-11-25
> **Response to Reviewer jujf (Part 2)**
>
> To evaluate the computational overhead and training efficiency introduced by ERL, we conducted additional experiments by systematically varying the maximum retry limits for the three erasure strategies (sub-answer, search, and plan). Since the erasure mechanism is implemented sequentially, the actual runtime measured on identical hardware directly reflects GPU computational load and external query volume. We recorded the following metrics:
> - Per-step wall-clock time (s/step) — directly reflecting computation and querying costs.
> - Accuracy-per-step curve — training progress relative to optimization steps.
> - Accuracy-per-time curve — training progress normalized by cumulative runtime, measuring practical training efficiency.
>
> The experimental results reveal significant differences in the effects of applying erasure at different stages, primarily reflecting the varying importance of each stage within the chain-of-thought reasoning. Applying the erasure mechanism to the sub-answer stage—which has the greatest impact on the final answer—yields the most substantial improvement in training efficiency. Although the computational time per step increases due to the rethinking process, the rate of improvement in training metrics per unit of computational time becomes even higher compared to the original PPO.
>
> Another noteworthy finding is that while GRPO demonstrates significantly higher efficiency per unit time than both the original PPO and PPO enhanced with erasure mechanisms, it exhibits a clearly lower performance ceiling. For more detailed experiments and analysis, please refer to Appendix A of the latest version of the paper.
>
> |Step|1|16|31|46|61|76|91|106|121|136|151|166|181|196|211|226|241|256|271|286|301|316|
> |-|-|-|-|-|-|-|-|-|-|-|-|-|-|-|-|-|-|-|-|-|-|-|
> |ppo|183|187|191|196|193|188|177|182|181|181|183|181|194|186|188|180|189|176|181|176|178|178|
> |grpo|176|178|162|158|141|126|128|129|135|127|135|152|158|155|144|144|140|156|172|181|203|209|
> |ε-sub_answer1|240|245|245|235|218|219|203|216|218|218|205|216|218|206|218|206|230|227|242|261|255|257|
> |ε-sub_answer3|320|317|303|311|327|311|312|310|326|297|299|269|283|293|282|280|279|260|279|325|364|362|
> |ε-sub_answer5|400|374|388|400|412|414|419|377|427|406|402|359|365|328|315|330|384|339|372|380|399|371|
>
> *Table 1: Time(s) per step. Per-step wall-clock time for for PPO, GRPO and Sub-Answer Erase at different budgets.*
>
>
>
> |Step|1|16|31|46|61|76|91|106|121|136|151|166|181|196|211|226|241|256|271|286|301|316|
> |-|-|-|-|-|-|-|-|-|-|-|-|-|-|-|-|-|-|-|-|-|-|-|
> |ppo|5.18|5.51|5.84|6.25|6.82|7.75|9.26|11.69|14.88|18.23|20.97|23.04|24.78|26.39|27.73|28.65|29.42|30.16|30.97|31.82|32.82|33.83|
> |grpo|3.87|5.97|8.43|12.05|17.59|22.80|26.26|27.82|28.79|29.68|30.38|31.05|31.66|32.28|32.94|33.67|34.30|34.82|34.99|35.07|35.08|35.07|
> |ε-sub_answer1|4.84|6.01|7.35|9.36|12.93|17.46|21.94|25.44|27.77|29.34|30.32|31.00|31.58|32.08|32.68|33.36|34.15|35.10|35.96|36.77|37.55|38.31|
> |ε-sub_answer3|4.46|6.06|7.94|10.76|15.60|21.25|25.86|29.03|31.10|32.76|34.03|34.76|35.37|36.09|36.96|37.57|37.91|38.20|38.68|39.76|41.12|42.56|
> |ε-sub_answer5|4.43|6.23|8.32|11.42|16.69|22.68|27.54|30.48|32.26|33.75|35.11|36.32|37.32|38.32|39.56|40.86|41.95|42.74|43.52|44.46|45.44|46.41|
>
> *Table 2: Correct(F1 %) by Step.*
>
>
>
> |Time(s)|200|5090|9980|14870|19760|24650|29540|34430|39320|44210|49100|53990|58880|63770|68660|73550|78440|83330|88220|93110|98000|
> |-|-|-|-|-|-|-|-|-|-|-|-|-|-|-|-|-|-|-|-|-|-|
> |ppo|5.19|5.75|6.49|8.07|12.04|18.09|22.57|25.64|28.03|29.48|30.88|32.53|34.03|-|-|-|-|-|-|-|-|
> |grpo|0.53|8.65|16.83|25.01|29.32|31.19|32.66|33.92|34.62|35.04|35.54|36.21|36.86|37.42|37.83|38.08|38.24|38.41|38.64|38.89|39.12|
> |ε-sub_answer1|4.82|6.38|8.46|13.17|19.87|25.47|28.71|30.38|31.38|32.18|33.18|34.33|35.62|36.70|37.69|38.46|-|-|-|-|-|
> |ε-sub_answer3|4.32|5.93|7.81|10.61|15.47|21.14|25.78|29.02|31.15|32.92|34.21|34.98|35.71|36.65|37.46|37.90|38.25|38.85|40.11|41.51|43.00|
> |ε-sub_answer5|5.25|6.12|7.10|8.65|11.92|16.68|21.97|26.33|29.29|31.03|32.27|33.45|34.67|35.77|36.80|37.68|38.61|39.84|41.18|42.07|42.87|
>
> *Table 3: Correct(F1 %) by Accumulated Time(s).*

---

> ### Author Response · Authors · 2025-11-25
> **Response to Reviewer jujf (Part 3)**
>
> |Step|1|16|31|46|61|76|91|106|121|136|151|166|181|196|211|226|241|256|271|286|301|316|
> |-|-|-|-|-|-|-|-|-|-|-|-|-|-|-|-|-|-|-|-|-|-|-|
> |ppo|183|187|191|196|193|188|177|182|181|181|183|181|194|186|188|180|189|176|181|176|178|178|
> |grpo|176|178|162|158|141|126|128|129|135|127|135|152|158|155|144|144|140|156|172|181|203|209|
> |ε-search1|264|264|260|228|201|217|213|251|250|260|292|281|297|273|248|260|264|251|240|236|219|232|
> |ε-search3|358|383|359|355|275|282|275|287|295|292|317|331|308|262|301|278|297|293|304|366|426|398|
> |ε-search5|372|384|380|476|591|526|391|388|373|341|351|308|285|270|275|308|374|356|371|526|535|541|
>
> *Table 4: Time(s) per step. Per-step wall-clock time for PPO, GRPO and Search Erase at different budgets.*
>
>
>
> |Step|1|16|31|46|61|76|91|106|121|136|151|166|181|196|211|226|241|256|271|286|301|316|
> |-|-|-|-|-|-|-|-|-|-|-|-|-|-|-|-|-|-|-|-|-|-|-|
> |ppo|5.18|5.51|5.84|6.25|6.82|7.75|9.26|11.69|14.88|18.23|20.97|23.04|24.78|26.39|27.73|28.65|29.42|30.16|30.97|31.82|32.82|33.83|
> |grpo|3.87|5.97|8.43|12.05|17.59|22.80|26.26|27.82|28.79|29.68|30.38|31.05|31.66|32.28|32.94|33.67|34.30|34.82|34.99|35.07|35.08|35.07|
> |ε-search1|5.64|6.07|6.53|7.22|8.83|11.57|15.16|18.79|21.57|23.44|24.67|25.83|27.05|28.26|29.40|30.37|31.35|32.26|33.14|33.79|34.41|35.02|
> |ε-search3|5.62|6.23|6.97|8.21|10.66|13.90|17.59|20.84|23.43|25.31|26.58|27.57|28.42|29.28|30.15|31.04|31.88|32.79|33.76|34.79|35.81|36.81|
> |ε-search5|5.00|6.35|7.81|9.62|12.20|15.32|18.73|21.78|24.29|26.11|27.63|28.92|30.01|31.05|31.91|32.79|33.59|34.43|35.11|35.90|36.84|37.82|
>
> *Table 5: Correct(F1 %) by Step.*
>
>
>
> |Time(s)|200|5090|9980|14870|19760|24650|29540|34430|39320|44210|49100|53990|58880|63770|68660|73550|78440|83330|88220|93110|98000|
> |-|-|-|-|-|-|-|-|-|-|-|-|-|-|-|-|-|-|-|-|-|-|
> |ppo|5.19|5.75|6.49|8.07|12.04|18.09|22.57|25.64|28.03|29.48|30.88|32.53|34.03|-|-|-|-|-|-|-|-|
> |grpo|0.53|8.65|16.83|25.01|29.32|31.19|32.66|33.92|34.62|35.04|35.54|36.21|36.86|37.42|37.83|38.08|38.24|38.41|38.64|38.89|39.12|
> |ε-search1|5.65|6.17|6.78|8.69|13.00|18.25|21.90|23.94|25.26|26.63|28.01|29.39|30.59|31.82|32.98|33.92|34.78|35.12|-|-|-|
> |ε-search3|5.78|6.18|6.68|7.52|9.61|13.01|17.38|21.17|23.99|25.83|27.07|28.04|28.99|29.97|30.98|31.89|32.85|33.87|34.94|36.10|37.17|
> |ε-search5|5.29|6.18|7.13|8.21|9.73|11.43|13.26|15.28|17.62|20.16|22.67|24.66|26.24|27.71|29.00|30.14|31.33|32.23|33.11|34.05|34.99|
>
> *Table 6: Correct(F1 %) by Accumulated Time(s).*

---

> > ### Author Response · Authors · 2025-11-25
> > **Response to Reviewer jujf (Part 4)**
> >
> > |Step|1|16|31|46|61|76|91|106|121|136|151|166|181|196|211|226|241|256|271|286|301|316|
> > |-|-|-|-|-|-|-|-|-|-|-|-|-|-|-|-|-|-|-|-|-|-|-|
> > |ppo|183|187|191|196|193|188|177|182|181|181|183|181|194|186|188|180|189|176|181|176|178|178|
> > |grpo|176|178|162|158|141|126|128|129|135|127|135|152|158|155|144|144|140|156|172|181|203|209|
> > |ε-plan1|212|221|218|221|223|211|212|197|206|211|208|207|215|216|219|215|221|228|234|240|244|242|
> > |ε-plan3|278|276|291|281|255|267|287|274|272|265|269|266|262|265|284|276|278|271|278|344|297|301|
> > |ε-plan5|370|353|352|354|344|341|394|356|335|314|370|327|321|337|311|327|350|337|323|329|339|344|
> >
> > *Table 7: Time(s) per step.Per-step wall-clock time for PPO, GRPO and Plan Erase at different budgets.*
> >
> >
> >
> > |Step|1|16|31|46|61|76|91|106|121|136|151|166|181|196|211|226|241|256|271|286|301|316|
> > |-|-|-|-|-|-|-|-|-|-|-|-|-|-|-|-|-|-|-|-|-|-|-|
> > |ppo|5.18|5.51|5.84|6.25|6.82|7.75|9.26|11.69|14.88|18.23|20.97|23.04|24.78|26.39|27.73|28.65|29.42|30.16|30.97|31.82|32.82|33.83|
> > |grpo|3.87|5.97|8.43|12.05|17.59|22.80|26.26|27.82|28.79|29.68|30.38|31.05|31.66|32.28|32.94|33.67|34.30|34.82|34.99|35.07|35.08|35.07|
> > |ε-plan1|4.90|5.36|5.87|6.56|8.11|10.61|13.82|17.02|19.91|22.41|24.56|26.27|27.56|28.55|29.35|30.07|30.77|31.41|32.03|32.57|33.19|33.82|
> > |ε-plan3|5.24|5.93|6.67|7.72|9.73|12.73|16.34|19.58|22.11|23.92|25.44|26.92|28.38|29.52|30.24|30.83|31.42|32.13|32.73|33.27|33.81|34.34|
> > |ε-plan5|4.56|5.91|7.36|9.22|12.00|15.24|18.58|21.43|23.72|25.54|27.05|28.35|29.27|30.01|30.77|31.63|32.30|32.72|33.14|33.81|34.60|35.42|
> >
> > *Table 8: Correct(F1 %) by Step.*
> >
> >
> >
> > |Time(s)|200|5090|9980|14870|19760|24650|29540|34430|39320|44210|49100|53990|58880|63770|68660|73550|78440|83330|88220|93110|98000|
> > |-|-|-|-|-|-|-|-|-|-|-|-|-|-|-|-|-|-|-|-|-|-|
> > |ppo|5.19|5.75|6.49|8.07|12.04|18.09|22.57|25.64|28.03|29.48|30.88|32.53|34.03|-|-|-|-|-|-|-|-|
> > |grpo|0.53|8.65|16.83|25.01|29.32|31.19|32.66|33.92|34.62|35.04|35.54|36.21|36.86|37.42|37.83|38.08|38.24|38.41|38.64|38.89|39.12|
> > |ε-plan1|4.90|5.61|6.55|9.22|13.86|18.78|22.83|25.88|27.86|29.17|30.24|31.22|32.09|32.88|33.74|33.96|-|-|-|-|-|
> > |ε-plan3|5.24|6.02|6.88|8.47|11.44|15.59|19.56|22.55|24.60|26.42|28.20|29.58|30.42|31.11|31.89|32.60|33.19|33.76|34.33|34.45|-|
> > |ε-plan5|4.76|5.80|6.91|8.38|10.73|13.68|16.81|19.78|22.28|24.38|26.00|27.48|28.68|29.47|30.13|30.92|31.79|32.40|32.73|33.07|33.39|
> >
> > *Table 9: Correct(F1 %) by Accumulated Time(s).*

---

> ### Author Response · Authors · 2025-11-25
> **Response to Reviewer jujf (Part 5)**
>
> > Writing issue, e.g. L160 Jin et al. (2025b) > (Jin et al., 2025b). These issue occurs very frequent spanning intro, sec 4.1, and related work.
>
> Regarding the writing issue you pointed out concerning inconsistent citation formatting, we apologize for the lack of standardization. We have carefully reviewed the entire paper and corrected all citation formats to ensure they comply with the required guidelines. Thank you for your attention to this detail. We believe these revisions will enhance the overall readability of the paper.
>
> > I would expect longer rollout time and more research queries thus make it both cost more and takes a longer time for each unit update/iteration, but I am not sure about the overall effect.
>
> To analyze the performance of ERL in deeper reasoning tasks, we conducted additional experiments to track the timing and frequency of erasure events in multi-hop reasoning trajectories. Specifically, for each training step, we recorded the average number of retries triggered by the sub-answer, search, and planning modules under varying maximum reasoning depths.
>
> Our findings indicate that the average number of erasures in the Sub-Answer module increases notably as reasoning depth grows. This suggests that later stages of the reasoning chain face greater difficulty in generating correct sub-answers, thus requiring more frequent erasure and retry. It also indirectly demonstrates that errors accumulated in earlier steps make subsequent reasoning more challenging, particularly for complex, multi-hop problems. The higher retry rate in these cases underscores the value of the ERL mechanism.
>
> A similar trend was observed in the Search module during later rounds (e.g., rounds 2–4) of regeneration. In contrast, the Planning module, which only allows erasure in the first round, exhibited a stable pattern of erasure events. For more detailed experiments and analysis, please refer to Appendix B of the latest version of the paper.
>
> |Step|1|16|31|46|61|76|91|106|121|136|151|166|181|196|211|226|241|256|271|286|301|316|
> |-|-|-|-|-|-|-|-|-|-|-|-|-|-|-|-|-|-|-|-|-|-|-|
> |round-1|1.00|0.93|1.14|1.11|1.06|1.00|0.50|1.17|1.06|0.95|0.97|1.00|0.00|0.95|0.42|1.00|1.55|1.00|0.97|1.50|0.98|1.44|
> |sub_answer-round-2|1.72|1.77|1.85|1.88|1.82|1.52|1.32|1.21|1.35|1.06|1.27|1.12|1.12|1.12|1.08|1.15|1.30|1.27|1.59|1.85|2.14|1.95|
> |sub_answer-round-3|2.20|2.02|2.24|2.37|2.12|1.92|1.88|1.88|1.97|1.75|1.74|1.67|1.76|1.72|1.51|1.55|1.62|1.55|1.71|1.86|2.32|2.39|
> |sub_answer-round-4|2.34|2.07|2.29|2.57|2.52|2.21|2.33|2.51|2.49|2.25|1.93|2.00|2.18|2.21|2.20|2.26|2.27|1.76|2.13|2.00|2.08|2.88|
>
> *Table 10: SubAnswer Retried times by Step.*
>
>
> |Step|1|16|31|46|61|76|91|106|121|136|151|166|181|196|211|226|241|256|271|286|301|316|
> |-|-|-|-|-|-|-|-|-|-|-|-|-|-|-|-|-|-|-|-|-|-|-|
> |plan-round-1|2.53|2.47|2.60|2.69|2.65|2.74|2.70|2.67|2.67|2.69|2.71|2.71|2.70|2.65|2.66|2.71|2.69|2.69|2.62|2.64|2.70|2.71|
> |search-round-2|1.61|1.80|1.69|1.76|1.00|0.90|0.85|0.88|0.83|0.82|0.86|0.81|0.76|0.78|0.77|0.76|0.79|0.77|0.83|0.82|0.93|0.87|
> |search-round-3|2.28|2.41|2.38|2.32|2.06|1.40|1.14|1.44|1.15|1.33|1.00|0.95|0.99|0.97|1.01|1.00|0.96|1.01|1.06|1.16|1.11|1.18|
> |search-round-4|2.33|2.31|2.34|2.36|2.44|3.00|2.00|2.33|1.00|1.00|1.00|1.00|2.00|0.67|1.83|1.89|0.99|1.17|1.00|1.80|1.61|1.85|
>
> *Table 11: Search&Plan Retried times by Step.*
>
> We extend our sincere gratitude for your thorough review and valuable suggestions on our paper. Through supplementary experiments and detailed explanations, we have addressed all raised questions and concerns while further enhancing the practical applicability and innovativeness of our work. We hope these revisions provide deeper insight into our proposed methodology, and we welcome any additional feedback you may have.

---

> > ### Author Response · Authors · 2025-11-27
> > **Ensuring Clarity: Follow-up on Our Rebuttal Responses**
> >
> > Dear Reviewer,
> >
> > Thank you once more for the valuable feedback you provided during the review process, as well as for giving us the opportunity to clarify and supplement our work through the rebuttal.
> >
> > In our responses above, we have conducted detailed additional experiments and explanations to address your concerns. We sincerely hope that our supplementary experiments and analyses have fully resolved the questions you raised.
> >
> > Therefore, we would like to take the liberty to follow up and ask whether you have any further questions regarding our previous responses or if there are any specific aspects for which you would like us to provide additional clarification. We are more than willing and fully prepared to offer any further information you may require.
> >
> > This research work represents a crucial part of our long-term exploration, and we greatly value this opportunity to exchange ideas at ICLR. We also hold your professional judgment in the highest regard. Any further feedback from you would be immensely important to us.
> >
> > Once again, thank you for the time and effort you have dedicated to this process!

---

### Official Review · Reviewer_knbt · 2025-11-01

**Soundness:** 3
**Presentation:** 3
**Contribution:** 3
**Rating:** 6
**Confidence:** 2

**Summary:**

This paper addresses critical reliability issues faced by search-augmented LLMs when performing complex multi-hop reasoning tasks. It identifies three primary failure modes: decomposition errors, retrieval errors, and reasoning errors. To mitigate these issues, the authors propose ERL, a framework that identifies, erases, and regenerates faulty reasoning steps, thereby preventing error propagation. Extensive experiments on multi-benchmarks demonstrate substantial performance improvements with ERL, achieving new state-of-the-art results.

**Strengths:**

- The paper introduces ERL, a novel approach to addressing the inherent brittleness in search-augmented LLMs. The key innovation lies in explicitly identifying, erasing, and regenerating faulty reasoning steps, a method previously unexplored in multi-hop reasoning contexts. This creative combination of reinforcement learning with targeted error correction significantly advances existing frameworks.

- Validates the framework across multiple challenging multi-hop reasoning datasets, HotpotQA, MuSiQue, 2Wiki, and Bamboogle, demonstrating generalizability.

- This framework enhances the robustness and real-world applicability of search-augmented language models, delivering state-of-the-art results on multiple benchmarks. Its dynamic error-correction capability ensures improved reliability and adaptability in practical scenarios.

**Weaknesses:**

- The main results are point estimates without confidence intervals, so it’s unclear whether reported gains exceed noise.

- The manuscript does not present empirical measurements or benchmarks of runtime, memory use, or computational resource demands. The lack of quantitative efficiency analysis weakens the practical argument for real-world applications.

- Certain equations contain notation ambiguities or inconsistencies, such as unclear definitions or insufficient explanations for symbols.

**Questions:**

Please check the weaknesses.

---

> ### Author Response · Authors · 2025-11-25
> **Response to Reviewer knbt (Part 1)**
>
> Dear Reviewer,
>
> Thank you very much for your thorough review and valuable feedback on our paper. We are truly honored to receive such professional and insightful comments. Your assessment has not only helped us identify areas for improvement but has also provided new perspectives for our research.
>
> We have carefully considered the weaknesses you pointed out and have made the following additions:
>
> >The main results are point estimates without confidence intervals, so it’s unclear whether reported gains exceed noise.
>
> We appreciate your comment regarding the absence of confidence intervals. We have now incorporated statistical analysis with confidence intervals into the main experimental results. Table 1 presents the experimental results with confidence intervals, which helps confirm that the reported performance gains are statistically significant and not due to random variations.
>
> | Method | HotpotQA† | HotpotQA† | 2Wiki† | 2Wiki† | MuSiQue† | MuSiQue† | Bamboogle† | Bamboogle† | HotpotQA* | HotpotQA* | 2Wiki* | 2Wiki* | MuSiQue* | MuSiQue* | Bamboogle* | Bamboogle* |
> |------------------------|-----------|-----------|--------|--------|----------|----------|------------|------------|-----------|-----------|--------|--------|----------|----------|------------|------------|
> | | EM↑ | F1↑ | EM↑ | F1↑ | EM↑ | F1↑ | EM↑ | F1↑ | EM↑ | F1↑ | EM↑ | F1↑ | EM↑ | F1↑ | EM↑ | F1↑ |
> | **Qwen2.5-3b-Base/Instruct** | | | | | | | | | | | | | | | | |
> | Search-R1-base | 0.268±0.004 | 0.355±0.005 | 0.250±0.002 | 0.298±0.002 | 0.078±0.002 | 0.138±0.006 | 0.188±0.011 | 0.281±0.012 | 0.255±0.005 | 0.429±0.007 | 0.377±0.003 | 0.447±0.002 | 0.118±0.006 | 0.186±0.006 | 0.274±0.011 | 0.397±0.013 |
> | Search-R1-instruct | 0.304±0.001 | 0.402±0.001 | 0.291±0.002 | 0.351±0.001 | 0.116±0.004 | 0.184±0.005 | 0.240±0.004 | 0.347±0.003 | 0.354±0.002 | 0.436±0.004 | 0.370±0.002 | 0.449±0.002 | 0.129±0.005 | 0.197±0.004 | 0.392±0.007 | 0.513±0.008 |
> | ZeroSearch-base | 0.276±0.002 | 0.374±0.003 | 0.255±0.002 | 0.311±0.002 | 0.091±0.005 | 0.166±0.004 | 0.168±0.009 | 0.252±0.010 | 0.320±0.007 | 0.410±0.006 | 0.381±0.009 | 0.465±0.011 | 0.131±0.014 | 0.231±0.015 | 0.355±0.014 | 0.513±0.016 |
> | ZeroSearch-instruct | 0.275±0.003 | 0.363±0.001 | 0.250±0.003 | 0.291±0.001 | 0.089±0.003 | 0.146±0.002 | 0.190±0.006 | 0.303±0.005 | 0.357±0.004 | 0.453±0.006 | 0.351±0.007 | 0.446±0.009 | 0.111±0.004 | 0.167±0.003 | 0.427±0.008 | 0.532±0.011 |
> | R-Search-instruct-GRPO | 0.313±0.006 | 0.413±0.004 | 0.324±0.017 | 0.366±0.015 | 0.125±0.007 | 0.190±0.011 | 0.246±0.013 | 0.354±0.015 | 0.371±0.008 | 0.465±0.011 | 0.460±0.011 | 0.521±0.014 | 0.139±0.008 | 0.230±0.010 | 0.497±0.015 | 0.627±0.018 |
> | R-Search-instruct-PPO | 0.273±0.014 | 0.362±0.015 | 0.278±0.002 | 0.329±0.002 | 0.119±0.006 | 0.177±0.007 | 0.252±0.008 | 0.351±0.011 | 0.392±0.006 | 0.492±0.008 | 0.498±0.006 | 0.560±0.009 | 0.142±0.006 | 0.227±0.007 | 0.493±0.012 | 0.652±0.016 |
> | SSRL-instruct | 0.321±0.009 | 0.419±0.010 | 0.298±0.008 | 0.358±0.009 | 0.109±0.009 | 0.171±0.013 | 0.247±0.026 | 0.333±0.040 | 0.366±0.017 | 0.444±0.019 | 0.394±0.019 | 0.496±0.023 | 0.122±0.014 | 0.204±0.016 | 0.389±0.033 | 0.503±0.041 |
> | StepSearch-base | 0.327±0.003 | 0.436±0.002 | 0.341±0.004 | 0.393±0.002 | 0.182±0.003 | 0.271±0.004 | 0.322±0.006 | 0.417±0.007 | 0.351±0.006 | 0.473±0.008 | 0.441±0.008 | 0.564±0.010 | 0.201±0.004 | 0.299±0.007 | 0.512±0.011 | 0.652±0.017 |
> | StepSearch-instruct | 0.346±0.002 | 0.451±0.001 | 0.320±0.003 | 0.386±0.001 | 0.180±0.003 | 0.260±0.005 | 0.341±0.009 | 0.449±0.010 | 0.396±0.007 | 0.478±0.007 | 0.406±0.010 | 0.501±0.009 | 0.155±0.006 | 0.243±0.009 | 0.520±0.012 | 0.644±0.011 |
> | ESearch-base | 0.412±0.005 | 0.546±0.007 | 0.428±0.002 | 0.500±0.002 | 0.234±0.007 | 0.344±0.009 | 0.414±0.011 | 0.529±0.014 | 0.436±0.005 | 0.582±0.010 | 0.577±0.007 | 0.679±0.011 | 0.241±0.009 | 0.362±0.012 | 0.625±0.019 | 0.789±0.022 |
> | ESearch-instruct | 0.448±0.004 | 0.589±0.004 | 0.416±0.002 | 0.499±0.003 | 0.233±0.006 | 0.336±0.009 | 0.442±0.009 | 0.585±0.012 | 0.516±0.006 | 0.614±0.009 | 0.522±0.004 | 0.641±0.007 | 0.213±0.008 | 0.313±0.013 | 0.671±0.014 | 0.807±0.018 |
>
> *Table 1: Main result. " † " indicates offline retrieval, and " * " indicates online retrieval.*

---

> > ### Author Response · Authors · 2025-11-25
> > **Response to Reviewer knbt (Part 2)**
> >
> > > The manuscript does not present empirical measurements or benchmarks of runtime, memory use, or computational resource demands. The lack of quantitative efficiency analysis weakens the practical argument for real-world applications.
> >
> > To evaluate the additional overhead and training efficiency introduced by ERL, we conducted additional experiments by systematically varying the maximum retry limits for the three erasure strategies (sub-answer, search, and plan). Since the erasure mechanism is implemented sequentially, the actual runtime measured on identical hardware directly reflects the GPU computational load and the volume of external queries. We recorded the following metrics:
> >
> > - Per-step wall-clock time (s/step) — directly reflecting the combined cost of computation and querying.
> >
> > - Accuracy-per-step curve — illustrating the training progress relative to the number of optimization steps.
> >
> > - Accuracy-per-time curve — illustrating the training progress normalized by the cumulative runtime, which measures the practical training efficiency.
> >
> > The experimental results reveal significant differences in the effects of applying erasure to different stages, primarily reflecting the varying importance of each stage within the chain-of-thought reasoning. Applying the erasure mechanism to the sub-answer stage—which has the greatest impact on the final answer—yields the most substantial improvement in training efficiency. Although the computational time per step increases due to the rethinking process, the rate of improvement in training metrics per unit of computational time becomes higher compared to the original PPO.
> >
> > Another noteworthy finding is that while GRPO demonstrates significantly higher efficiency per unit time than both the original PPO and the PPO variants with erasure mechanisms, it exhibits a clearly lower performance ceiling. For more detailed experiments and analysis, please refer to Appendix A in the latest version of the paper.
> >
> > |Step|1|16|31|46|61|76|91|106|121|136|151|166|181|196|211|226|241|256|271|286|301|316|
> > |-|-|-|-|-|-|-|-|-|-|-|-|-|-|-|-|-|-|-|-|-|-|-|
> > |ppo|183|187|191|196|193|188|177|182|181|181|183|181|194|186|188|180|189|176|181|176|178|178|
> > |grpo|176|178|162|158|141|126|128|129|135|127|135|152|158|155|144|144|140|156|172|181|203|209|
> > |ε-sub_answer1|240|245|245|235|218|219|203|216|218|218|205|216|218|206|218|206|230|227|242|261|255|257|
> > |ε-sub_answer3|320|317|303|311|327|311|312|310|326|297|299|269|283|293|282|280|279|260|279|325|364|362|
> > |ε-sub_answer5|400|374|388|400|412|414|419|377|427|406|402|359|365|328|315|330|384|339|372|380|399|371|
> >
> > *Table 2: Time(s) per step. Per-step wall-clock time for for PPO, GRPO and Sub-Answer Erase at different budgets.*
> >
> >
> >
> > |Step|1|16|31|46|61|76|91|106|121|136|151|166|181|196|211|226|241|256|271|286|301|316|
> > |-|-|-|-|-|-|-|-|-|-|-|-|-|-|-|-|-|-|-|-|-|-|-|
> > |ppo|5.18|5.51|5.84|6.25|6.82|7.75|9.26|11.69|14.88|18.23|20.97|23.04|24.78|26.39|27.73|28.65|29.42|30.16|30.97|31.82|32.82|33.83|
> > |grpo|3.87|5.97|8.43|12.05|17.59|22.80|26.26|27.82|28.79|29.68|30.38|31.05|31.66|32.28|32.94|33.67|34.30|34.82|34.99|35.07|35.08|35.07|
> > |ε-sub_answer1|4.84|6.01|7.35|9.36|12.93|17.46|21.94|25.44|27.77|29.34|30.32|31.00|31.58|32.08|32.68|33.36|34.15|35.10|35.96|36.77|37.55|38.31|
> > |ε-sub_answer3|4.46|6.06|7.94|10.76|15.60|21.25|25.86|29.03|31.10|32.76|34.03|34.76|35.37|36.09|36.96|37.57|37.91|38.20|38.68|39.76|41.12|42.56|
> > |ε-sub_answer5|4.43|6.23|8.32|11.42|16.69|22.68|27.54|30.48|32.26|33.75|35.11|36.32|37.32|38.32|39.56|40.86|41.95|42.74|43.52|44.46|45.44|46.41|
> >
> > *Table 3: Correct(F1 %) by Step.*
> >
> >
> >
> > |Time(s)|200|5090|9980|14870|19760|24650|29540|34430|39320|44210|49100|53990|58880|63770|68660|73550|78440|83330|88220|93110|98000|
> > |-|-|-|-|-|-|-|-|-|-|-|-|-|-|-|-|-|-|-|-|-|-|
> > |ppo|5.19|5.75|6.49|8.07|12.04|18.09|22.57|25.64|28.03|29.48|30.88|32.53|34.03|-|-|-|-|-|-|-|-|
> > |grpo|0.53|8.65|16.83|25.01|29.32|31.19|32.66|33.92|34.62|35.04|35.54|36.21|36.86|37.42|37.83|38.08|38.24|38.41|38.64|38.89|39.12|
> > |ε-sub_answer1|4.82|6.38|8.46|13.17|19.87|25.47|28.71|30.38|31.38|32.18|33.18|34.33|35.62|36.70|37.69|38.46|-|-|-|-|-|
> > |ε-sub_answer3|4.32|5.93|7.81|10.61|15.47|21.14|25.78|29.02|31.15|32.92|34.21|34.98|35.71|36.65|37.46|37.90|38.25|38.85|40.11|41.51|43.00|
> > |ε-sub_answer5|5.25|6.12|7.10|8.65|11.92|16.68|21.97|26.33|29.29|31.03|32.27|33.45|34.67|35.77|36.80|37.68|38.61|39.84|41.18|42.07|42.87|
> >
> > *Table 4: Correct(F1 %) by Accumulated Time(s).*

---

> ### Author Response · Authors · 2025-11-25
> **Response to Reviewer knbt (Part 3)**
>
> |Step|1|16|31|46|61|76|91|106|121|136|151|166|181|196|211|226|241|256|271|286|301|316|
> |-|-|-|-|-|-|-|-|-|-|-|-|-|-|-|-|-|-|-|-|-|-|-|
> |ppo|183|187|191|196|193|188|177|182|181|181|183|181|194|186|188|180|189|176|181|176|178|178|
> |grpo|176|178|162|158|141|126|128|129|135|127|135|152|158|155|144|144|140|156|172|181|203|209|
> |ε-search1|264|264|260|228|201|217|213|251|250|260|292|281|297|273|248|260|264|251|240|236|219|232|
> |ε-search3|358|383|359|355|275|282|275|287|295|292|317|331|308|262|301|278|297|293|304|366|426|398|
> |ε-search5|372|384|380|476|591|526|391|388|373|341|351|308|285|270|275|308|374|356|371|526|535|541|
>
> *Table 5: Time(s) per step. Per-step wall-clock time for PPO, GRPO and Search Erase at different budgets.*
>
>
>
> |Step|1|16|31|46|61|76|91|106|121|136|151|166|181|196|211|226|241|256|271|286|301|316|
> |-|-|-|-|-|-|-|-|-|-|-|-|-|-|-|-|-|-|-|-|-|-|-|
> |ppo|5.18|5.51|5.84|6.25|6.82|7.75|9.26|11.69|14.88|18.23|20.97|23.04|24.78|26.39|27.73|28.65|29.42|30.16|30.97|31.82|32.82|33.83|
> |grpo|3.87|5.97|8.43|12.05|17.59|22.80|26.26|27.82|28.79|29.68|30.38|31.05|31.66|32.28|32.94|33.67|34.30|34.82|34.99|35.07|35.08|35.07|
> |ε-search1|5.64|6.07|6.53|7.22|8.83|11.57|15.16|18.79|21.57|23.44|24.67|25.83|27.05|28.26|29.40|30.37|31.35|32.26|33.14|33.79|34.41|35.02|
> |ε-search3|5.62|6.23|6.97|8.21|10.66|13.90|17.59|20.84|23.43|25.31|26.58|27.57|28.42|29.28|30.15|31.04|31.88|32.79|33.76|34.79|35.81|36.81|
> |ε-search5|5.00|6.35|7.81|9.62|12.20|15.32|18.73|21.78|24.29|26.11|27.63|28.92|30.01|31.05|31.91|32.79|33.59|34.43|35.11|35.90|36.84|37.82|
>
> *Table 6: Correct(F1 %) by Step.*
>
>
>
> |Time(s)|200|5090|9980|14870|19760|24650|29540|34430|39320|44210|49100|53990|58880|63770|68660|73550|78440|83330|88220|93110|98000|
> |-|-|-|-|-|-|-|-|-|-|-|-|-|-|-|-|-|-|-|-|-|-|
> |ppo|5.19|5.75|6.49|8.07|12.04|18.09|22.57|25.64|28.03|29.48|30.88|32.53|34.03|-|-|-|-|-|-|-|-|
> |grpo|0.53|8.65|16.83|25.01|29.32|31.19|32.66|33.92|34.62|35.04|35.54|36.21|36.86|37.42|37.83|38.08|38.24|38.41|38.64|38.89|39.12|
> |ε-search1|5.65|6.17|6.78|8.69|13.00|18.25|21.90|23.94|25.26|26.63|28.01|29.39|30.59|31.82|32.98|33.92|34.78|35.12|-|-|-|
> |ε-search3|5.78|6.18|6.68|7.52|9.61|13.01|17.38|21.17|23.99|25.83|27.07|28.04|28.99|29.97|30.98|31.89|32.85|33.87|34.94|36.10|37.17|
> |ε-search5|5.29|6.18|7.13|8.21|9.73|11.43|13.26|15.28|17.62|20.16|22.67|24.66|26.24|27.71|29.00|30.14|31.33|32.23|33.11|34.05|34.99|
>
> *Table 7: Correct(F1 %) by Accumulated Time(s).*

---

> > ### Author Response · Authors · 2025-11-25
> > **Response to Reviewer knbt (Part 4)**
> >
> > |Step|1|16|31|46|61|76|91|106|121|136|151|166|181|196|211|226|241|256|271|286|301|316|
> > |-|-|-|-|-|-|-|-|-|-|-|-|-|-|-|-|-|-|-|-|-|-|-|
> > |ppo|183|187|191|196|193|188|177|182|181|181|183|181|194|186|188|180|189|176|181|176|178|178|
> > |grpo|176|178|162|158|141|126|128|129|135|127|135|152|158|155|144|144|140|156|172|181|203|209|
> > |ε-plan1|212|221|218|221|223|211|212|197|206|211|208|207|215|216|219|215|221|228|234|240|244|242|
> > |ε-plan3|278|276|291|281|255|267|287|274|272|265|269|266|262|265|284|276|278|271|278|344|297|301|
> > |ε-plan5|370|353|352|354|344|341|394|356|335|314|370|327|321|337|311|327|350|337|323|329|339|344|
> >
> > *Table 8: Time(s) per step.Per-step wall-clock time for PPO, GRPO and Plan Erase at different budgets.*
> >
> >
> >
> > |Step|1|16|31|46|61|76|91|106|121|136|151|166|181|196|211|226|241|256|271|286|301|316|
> > |-|-|-|-|-|-|-|-|-|-|-|-|-|-|-|-|-|-|-|-|-|-|-|
> > |ppo|5.18|5.51|5.84|6.25|6.82|7.75|9.26|11.69|14.88|18.23|20.97|23.04|24.78|26.39|27.73|28.65|29.42|30.16|30.97|31.82|32.82|33.83|
> > |grpo|3.87|5.97|8.43|12.05|17.59|22.80|26.26|27.82|28.79|29.68|30.38|31.05|31.66|32.28|32.94|33.67|34.30|34.82|34.99|35.07|35.08|35.07|
> > |ε-plan1|4.90|5.36|5.87|6.56|8.11|10.61|13.82|17.02|19.91|22.41|24.56|26.27|27.56|28.55|29.35|30.07|30.77|31.41|32.03|32.57|33.19|33.82|
> > |ε-plan3|5.24|5.93|6.67|7.72|9.73|12.73|16.34|19.58|22.11|23.92|25.44|26.92|28.38|29.52|30.24|30.83|31.42|32.13|32.73|33.27|33.81|34.34|
> > |ε-plan5|4.56|5.91|7.36|9.22|12.00|15.24|18.58|21.43|23.72|25.54|27.05|28.35|29.27|30.01|30.77|31.63|32.30|32.72|33.14|33.81|34.60|35.42|
> >
> > *Table 9: Correct(F1 %) by Step.*
> >
> >
> >
> > |Time(s)|200|5090|9980|14870|19760|24650|29540|34430|39320|44210|49100|53990|58880|63770|68660|73550|78440|83330|88220|93110|98000|
> > |-|-|-|-|-|-|-|-|-|-|-|-|-|-|-|-|-|-|-|-|-|-|
> > |ppo|5.19|5.75|6.49|8.07|12.04|18.09|22.57|25.64|28.03|29.48|30.88|32.53|34.03|-|-|-|-|-|-|-|-|
> > |grpo|0.53|8.65|16.83|25.01|29.32|31.19|32.66|33.92|34.62|35.04|35.54|36.21|36.86|37.42|37.83|38.08|38.24|38.41|38.64|38.89|39.12|
> > |ε-plan1|4.90|5.61|6.55|9.22|13.86|18.78|22.83|25.88|27.86|29.17|30.24|31.22|32.09|32.88|33.74|33.96|-|-|-|-|-|
> > |ε-plan3|5.24|6.02|6.88|8.47|11.44|15.59|19.56|22.55|24.60|26.42|28.20|29.58|30.42|31.11|31.89|32.60|33.19|33.76|34.33|34.45|-|
> > |ε-plan5|4.76|5.80|6.91|8.38|10.73|13.68|16.81|19.78|22.28|24.38|26.00|27.48|28.68|29.47|30.13|30.92|31.79|32.40|32.73|33.07|33.39|
> >
> > *Table 10: Correct(F1 %) by Accumulated Time(s).*
> >
> >
> > > Certain equations contain notation ambiguities or inconsistencies, such as unclear definitions or insufficient explanations for symbols.
> >
> > Thank you very much for your valuable feedback regarding the "ambiguity and inconsistency in the mathematical notation." We fully agree that a clear and rigorous notation system is essential for understanding the methodology. Following your suggestion, we have systematically reviewed the relevant formulas and improved the explanations of the corresponding notations to ensure readers can clearly comprehend the meaning and role of each symbol within the method.
> >
> > We sincerely appreciate your meticulous work throughout the review process. We hope these additions have addressed your concerns and further validate the reliability and practical utility of our method in multi-step reasoning. Should you have any further questions or suggestions, we would be very pleased to discuss them.

---

> > > ### Author Response · Authors · 2025-11-27
> > > **Ensuring Clarity: Follow-up on Our Rebuttal Responses**
> > >
> > > Dear Reviewer,
> > >
> > > Thank you once more for the valuable feedback you provided during the review process, as well as for giving us the opportunity to clarify and supplement our work through the rebuttal.
> > >
> > > In our responses above, we have conducted detailed additional experiments and explanations to address your concerns. We sincerely hope that our supplementary experiments and analyses have fully resolved the questions you raised.
> > >
> > > Therefore, we would like to take the liberty to follow up and ask whether you have any further questions regarding our previous responses or if there are any specific aspects for which you would like us to provide additional clarification. We are more than willing and fully prepared to offer any further information you may require.
> > >
> > > This research work represents a crucial part of our long-term exploration, and we greatly value this opportunity to exchange ideas at ICLR. We also hold your professional judgment in the highest regard. Any further feedback from you would be immensely important to us.
> > >
> > > Once again, thank you for the time and effort you have dedicated to this process!

---

### Author Response · Authors · 2025-12-01
**Rebuttal Summary: Core Contributions, Reviewer Feedback, and Our Responses (Part 1)**

Dear ACs, SACs, and PCs,

We extend our deepest gratitude to you for organizing this esteemed conference and for providing us with the valuable opportunity to submit our work. We are profoundly aware of the immense dedication, time, and scholarly expertise required to orchestrate such an event and to review each submission meticulously. We sincerely thank you for your exceptional efforts and unwavering commitment.

We are also immensely grateful to the reviewers for their insightful and constructive feedback. Their discerning comments were instrumental in helping us refine our research and enhance the clarity and rigor of the manuscript. Below, we summarize the core contribution of our work and provide an overview of our responses to the reviewers' comments and the corresponding enhancements made to the paper.
***
***

## Core Contribution:
This work addresses the reliability challenges—such as decomposition errors, retrieval misses, and reasoning errors—faced by search-augmented large language models in complex multi-step reasoning. We propose a novel Erasable Reinforcement Learning (ERL) framework.
- Its core inspiration is the human cognitive process of "identify error → erase → rewrite" for self-correction. During RL training, ERL dynamically identifies faulty steps within a reasoning chain, "erases" them, and restarts generation from the last correct state.

- This mechanism effectively halts error propagation, transforming a fragile reasoning process into a robust and reliable one, thereby substantially improving sample efficiency in reinforcement learning and enabling more stable training with a higher performance ceiling.

- Experimental results on multiple multi-hop QA benchmarks demonstrate that our ERL-trained model (ESearch) achieves significant performance improvements, establishing new state-of-the-art results.
***
***

## Response to Reviewer Comments & Paper Enhancements:

>  Statistical Rigor (Addressing Reviewer knbt):

We have added confidence intervals to results in the main results table, providing statistical evidence for the significance of the reported performance gains. Specific details can be found in **Appendix D**.
***

>  Computational Efficiency & Overhead Analysis (Addressing Reviewer knbt、jujf、BrTp、gvRH):

We have added a comprehensive efficiency analysis in **Appendix A**, including:

- Per-step Latency: Quantified and compared the computation and query time per step under different erasure strategies and retry limits.

- Convergence Efficiency Curves: Plotted "Accuracy vs. Training Steps" and "Accuracy vs. Cumulative Training Time" curves. The results show that while ERL incurs slightly higher per-step cost, it achieves higher training efficiency per unit of time and converges to a significantly higher performance ceiling compared to baselines like PPO and GRPO.
***

>  Method Generalizability & Extensibility (Addressing Reviewer jujf、BrTp、gvRH):

- Depth Scalability Analysis: In **Appendix B**, we analyze the average number of erasures per module across different reasoning depths. The results show more frequent "sub-answer" erasures in later stages, vividly illustrating the challenge of error propagation and the value of ERL in mitigating it.

- Task Generalization Potential: In our response, we elaborated on how the ERL framework is principally generalizable to broader scenarios such as long-text reasoning and tool use. We also discussed potential adaptations required for these new settings.
***

>  Clarification on Implementation Details & Dependencies (Addressing Reviewer jujf、BrTp):

- Training Data: We have explicitly stated the use of the MuSiQue dataset for training, whose inherent structure of sub-questions, evidence, and answers provides direct supervision signals.

- Universal Setting without Gold Evidence: We discuss how to leverage verifiable weak signals (such as retrieval relevance, self-consistency scores, etc.) to approximate dense rewards in more general task settings, and point out that achieving model autonomous erasure is an important direction for future exploration.
***

>  Hyperparameter Sensitivity Analysis (Addressing Reviewer BrTp、gvRH):

We have added a systematic hyperparameter sensitivity analysis in **Appendix C**, focusing on the erasure-triggering thresholds α and β. Results indicate that ERL remains robust within a reasonable parameter range, with limited performance fluctuation. Significant degradation only occurs with extreme values.
***

>  Improved Writing and Notational Clarity (Addressing Reviewer knbt、jujf、gvRH):

- Mathematical Notation: We have thoroughly reviewed and clarified all symbolic definitions in the formulas throughout the paper to ensure accuracy and consistency.

- Citation Format: We have uniformly revised the citation format throughout the manuscript to comply with the conference requirements.

---

> ### Author Response · Authors · 2025-12-01
> **Rebuttal Summary: Core Contributions, Reviewer Feedback, and Our Responses (Part 2)**
>
> >  Elaboration on Contribution Depth (Addressing Reviewer BrTp):
>
> We further clarify that ERL introduces a new paradigm of "reasoning as a dynamically editable process." It fundamentally alters the way reinforcement learning agents handle errors in their internal states. This idea can be analogized to the revolutionary significance of residual connections in deep neural network architectures, demonstrating potential paradigm-shifting value.
> ***
> ***
> ## Summary
> Throughout this
>  rebuttal and revision process, we have systematically incorporated key experiments and in-depth analyses that directly address all valuable reviewer concerns. The revised manuscript now offers strengthened:
>
> - Statistical rigor (CI analysis)
> - Quantitative efficiency evaluation
> - Robustness and sensitivity analyses
> - Clarification of methodological generalizability
>
>
> These enhancements collectively reinforce the paper’s position as a work that is conceptually novel, empirically solid, and broadly inspiring for future research on reliable reasoning in large language models.
>
> We once again extend our sincere appreciation for the tremendous efforts and invaluable guidance provided by the reviewers, ACs, SACs, and PCs.

---

### Meta-Review · Area_Chair_xFCk · 2026-01-07

**Summary:**

This paper proposes Erasable Reinforcement Learning (ERL), a framework designed to improve the robustness of search-augmented large language models in multi-hop reasoning. Reviewers broadly agree that the work tackles an important and timely problem—error propagation in multi-step reasoning—and introduces a novel mechanism that explicitly detects, erases, and regenerates faulty reasoning steps. The empirical results are strong, with consistent state-of-the-art performance across several multi-hop QA benchmarks and both offline and online retrieval settings.

At the same time, reviewers initially raised concerns about (i) missing statistical rigor (confidence intervals), (ii) unclear computational efficiency and scalability, (iii) reliance on gold supervision and limited discussion of generalization beyond QA, and (iv) clarity and presentation issues.

**Reviewer Concerns:**

**Concerns largely addressed by the rebuttal:**

Statistical significance of results: The authors added confidence intervals to main results, alleviating concerns about variance and noise.

Computational cost and efficiency: Extensive new experiments reporting wall-clock time, accuracy-per-step, and accuracy-per-time curves provide a much clearer picture of the overhead and efficiency trade-offs of ERL relative to PPO and GRPO.

Hyperparameter sensitivity (erasure thresholds): The rebuttal includes detailed sensitivity analyses showing bounded and interpretable effects, addressing robustness concerns.

Notation clarity and writing issues: Reviewers’ comments on ambiguous notation and inconsistent citations appear to have been carefully addressed through revisions.

Qualitative understanding of erasure behavior: Added analyses on erasure frequency across reasoning depth help clarify when and why erasure occurs.

**Concerns that remain partially outstanding**

Generality beyond QA: The rebuttal provides convincing conceptual arguments and discussion, but no concrete empirical evidence beyond QA-style benchmarks.

Practical deployment assumptions: Reliance on gold sub-answers or structured datasets during training remains a concern for some reviewers, even though the authors outline promising future directions using weak or proxy signals.

**Reviewer Scores:**

Reviewer knbt (initial: 6, marginally above threshold): Likely to maintain a 6, given that all stated weaknesses were directly addressed with substantial new experiments.

Reviewer jujf (initial: 6): Likely to remain at 6, with improved confidence due to added efficiency analyses and clarification of training details, though generalization concerns persist.

Reviewer BrTp (initial: 4): Likely to increase to 6, as questions on efficiency, scalability, and sensitivity were comprehensively answered.

Reviewer gvRH (initial: 6): Likely to remain at 6, with stronger justification after rebuttal, though some scope and generalization limitations remain.

---

### Decision · Program_Chairs · 2026-01-26

Accept (Poster)